

# Multi-decadal (1953 – 2017) rock glacier morphodynamics analysed by high-resolution topographic data in the Upper Kauner Valley, Austria

Fabian Fleischer[1], Florian Haas[1], Livia Piermattei[2], Madlene Pfeiffer[3], Tobias Heckmann[1], Moritz Altmann[1], Jakob Rom[1], Manuel Stark[1], Michael H. Wimmer[4], Norbert Pfeifer[4] and Michael Becht[1]

[1] Chair of Physical Geography, Catholic University of Eichstätt-Ingolstadt, 95072 Eichstätt, Germany
[2] Department of Geosciences, University of Oslo, 0316 Oslo, Norway
[3] Institute of Geography, University of Bremen, 28359 Bremen, Germany
[4] Department of Geodesy and Geoinformation, TU Wien, 1040 Vienna, Austria

*Correspondence to*: Fabian Fleischer (Fabian.fleischer@ku.de)

**Abstract.** Permafrost is being degraded worldwide due to the change in external forcing caused by climate change. This has also been shown to affect the morphodynamics of active rock glaciers. We studied these changes, depending on the analysis, on nine or eight active rock glaciers with different characteristics in multiple epochs between 1953 and 2017 in Kauner Valley, Austria. A combination of historical aerial photographs and airborne laser scanning data and their derivatives are used to analyse surface movement and 3D displacements. In general, the studied landforms show a significant acceleration of varying magnitude in the epoch 1997–2006 and a volume loss to varying degrees throughout the investigation period. Besides, we detect rock glaciers that show indication of inactivation. By analysing meteorological data (temperature, precipitation and snow cover onset and duration), we are able to identify possible links to these external forcing parameters. The combined investigation of horizontal and vertical 3D displacements shows that these are temporally decoupled on some rock glaciers. The catchment-wide survey further reveals that, despite the general trend, timing, magnitude and temporal peaks of morphodynamic changes indicate a slightly different sensitivity, response or response time of individual rock glaciers to fluctuations and changes in external forcing parameters.



## 1 Introduction

Rock glaciers are a downslope creep phenomenon of mountain permafrost and responsible for extensive mass transport in alpine environments (Barsch, 1996). Active rock glaciers consist of a coarse debris layer (active layer), covering ice supersaturated debris or pure ice and form lobate or tongue shaped land forms (Haeberli et al., 2006). They can be found in most cold mountain regions of the earth (Jones et al., 2019). It has long been debated whether rock glaciers are of glacial (e.g. Whalley, 1974; Humlum, 1982) or periglacial (e.g. Barsch, 1977; Haeberli, 1985) origin. A review of this topic is presented in Berthling (2011). In his definition, permafrost is the primary condition for rock glaciers to form, but the deforming ice and debris may be of both glacial and periglacial origin, stressing the equifinality of these landforms. Besides, the incorporated ice might, in part, also develop from freezing liquid precipitation or melt water (Humlum, 1998). Borehole cores and inclinometer measurements from different rock glaciers reveal the internal structure and deformation of these landforms (Arenson et al., 2002; Krainer et al., 2015; Buchli et al., 2018). They suggest common layers, although structure, composition and thickness of these layers differ to a certain extent also within individual rock glaciers. The few meters thick topmost active layer consists of unconsolidated boulders and isolates the underlying ice-rich frozen body. Because pores in the active layer are mostly air filled, the thermal regime is governed by freeze-thaw cycles and air convection. These are in turn controlled by ground permeability and the thermal gradient and have an influence on the ground thermal regime of the underlying permafrost (Wicky and Hauck, 2020). The ice-rich permafrost body constitutes the main body of the rock glacier. The volumetric ice content varies between 40% - 90% (Arenson et al., 2002; Haeberli et al., 2006; Hausmann et al., 2012) and the debris size is smaller than in the active layer, which is explained by fall sorting, washing away and kinetic sieving (Haeberli et al., 2006). The thermal regime of the permafrost layer is mainly controlled by heat conduction, therefore temperature signal from the surface is linearly delayed and its amplitude exponentially decreases with depth (Haeberli et al., 2006). This leads to the conclusion that seasonal temperature signals can just penetrate to the permafrost body to the depth of zero amplitude. Changes below this depth require long-term changes of the thermal forcing. In the permafrost layer, internal plastic deformation is the main component of deformation, which is governed by temperature and the structure of the debris-ice mixture. A large part of the horizontal deformation (50% - 97%) takes place in one or more shear zones at the base of the ice-rich permafrost body, which are a few meters thick (Arenson et al., 2002). These shear zones can be active at the same time (Krainer et al., 2015) or if inactive give information on past creep behaviour and failure mechanisms (Buchli et al., 2018). The dynamics of rock glaciers vary at different temporal scales: decadal, inter-annual and intra-annual (Delaloye et al., 2008; Delaloye et al., 2010; Wirz et al., 2016b; Kenner et al., 2017). Studies investigating the decadal-scale variability of rock glacier morphodynamics reveal a significant increase in flow velocities starting in the 1990s, while some studies also report phases of stable flow velocities and velocity decrease since then (Roer, 2005; Kellerer-Pirklbauer and Kaufmann, 2012; Scapozza et al., 2014; Hartl et al., 2016; Kellerer-Pirklbauer and Kaufmann, 2018; Kenner et al., 2020). The increase in flow velocities has been explained by rising mean annual air and ground temperatures and mechanisms of heat conduction and melt water advection (Roer, 2005; Kääb et al., 2007; Ikeda et al., 2008; Delaloye et al., 2010). Kääb et al. (2007) show that in general rock glaciers with ground



temperatures close to 0 °C move faster and are more sensitive to thermal forcing than colder ones. More recent studies
highlighted the role of liquid water, especially in the shear horizon, and attribute little or no significance to the change in
permafrost temperature to explain the deformation variations on a multi-annual, inter-annual, seasonal and shot-term scale.
(Wirz et al., 2016a; Kenner et al., 2017; Buchli et al., 2018; Cicoira et al., 2019a). Kenner et al. (2020) synthesise these findings
by showing that water availability in the rock glacier is governed by ground temperature which is a function of mean annual
air temperature and onset as well as duration of snow cover and thus correlate with rock glacier deformation as well. Although
rock glaciers normally move at rates ranging from a few cm/yr to a few m/yr, some studies show destabilization of rock glaciers
leading to rates of movement of several tens of meters per year (Roer et al., 2008; Scotti et al., 2017). Besides, rock glacier
dynamics can also be influenced by other factors like topography, temporal and vertical variations in ice content, rheology of
the ice-debris mixture, thickness, and input of ice and debris to the system.

The present and former response of rock glacier morphodynamics to atmospheric warming and climate change observed in
many high mountain regions (Hock et al., 2019) is of large scientific interest for climate change projections and landscape
evolution models. But an understanding of these landforms has also implications for natural hazard protection (Schoeneich et
al., 2015) or future water availability (Jones et al., 2019). Although there are several studies investigating rock glacier
morphodynamics on different time scales, the number of studies is low compared to ice glaciers. Groh and Blöthe (2019)
investigated the recent flow velocities of Kauner Valley rock glaciers and ascertained slightly lower flow velocities between
2001/2003-2009 than between 2010-2015. They also noted that the velocity of rock glaciers in the study area mainly depends
on parameters describing the general inclination and that their activity status is controlled by their size and the topoclimate.

Apart from Roer et al. (2005), who investigated multi-decadal catchment wide rock glacier morphodynamics in Turtman
Valley, Swiss Alps, most studies investigating rock glacier morphodynamics on a decadal time scale investigate just one or
two large and prominent rock glaciers (Scapozza et al., 2014; Scotti et al., 2017; Kellerer-Pirklbauer et al., 2018; Kellerer-
Pirklbauer and Kaufmann, 2018; Kaufmann et al., 2019; Kenner et al., 2020).

In this study, we focus on long-term (1953-2017) morphodynamic investigations rock glaciers, located in the Upper Kauner
Valley, Ötztal Alps, Austria, displaying different characteristics in order to understand their reaction to climate change under
similar climatic forcing. We do this by analysing surface movement (flow velocity) of eight rock glaciers by means of image-
correlation techniques on the base of orthophotos and hillshades. In addition, multitemporal 3D displacements are derived for
nine rock glaciers by a 3D distance analysis using photogrammetric as well as Airborne Laser Sanning (ALS) point clouds.
The identified changes in rock glacier morphodynamics will be discussed with regard to rock glacier characteristics, elevation
classes and changes in the meteorological forcing by investigating different climate parameters recorded directly in the
catchment and nearby meteorological stations.



## 2 Study Area and investigated rock glaciers

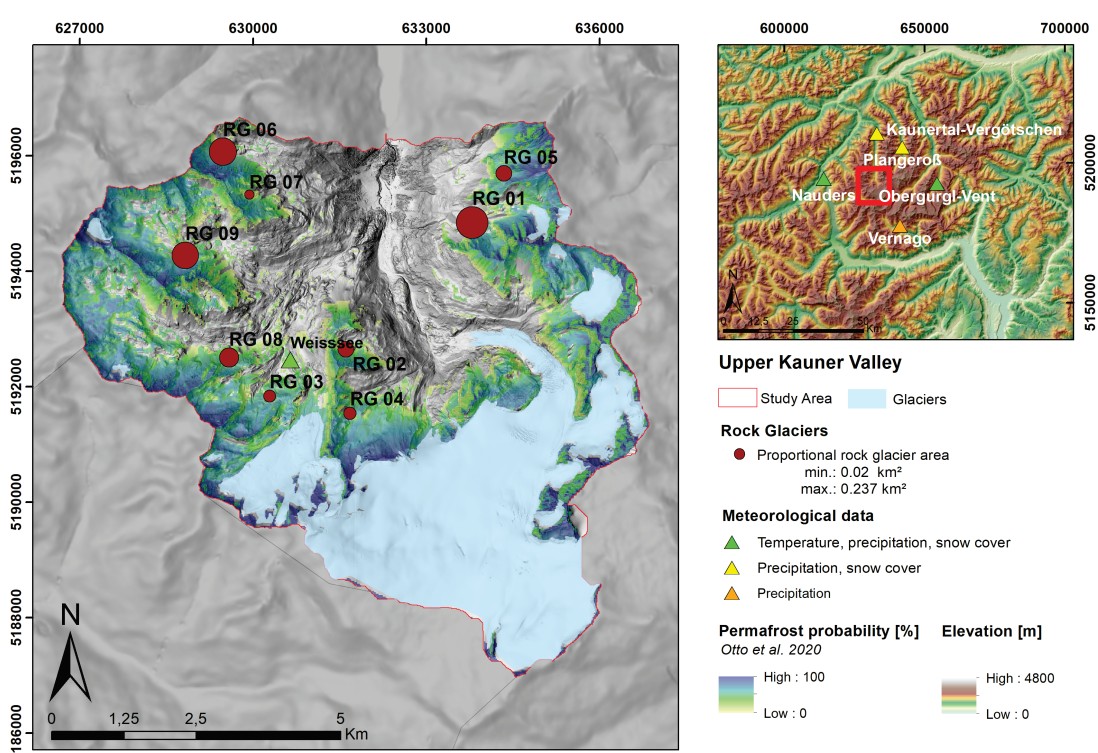

**Figure 1. Location and relative size of the investigated rock glaciers in the study area and location of the analysed meteorological stations. The stations Obergurgl and Obergurgl-Vent are marked as one station due to their proximity of 230 m. The permafrost probability (Otto et al., 2020) and glacier extents of 2015 (Buckel and Otto, 2018) are shown. The overview map and the background of the main map were produced using Copernicus data and information funded by the European Union - EU-DEM layers.**

The investigated rock glaciers are located in the catchment area of the Upper Kauner Valley within the Ötztal Alps, Austria (Fig. 1). The study area is more precisely defined as the hydrological contributing area at the inlet of the Fagge river into the Gepatsch reservoir. It has an area of ~62 km² and a relief of 1773 m ranging from 1810 m at the inlet to 3583 m at the summit of Hochvernagtspitze. Geologically, the study area belongs to the Eastern Alps crystalline zone and the polymetamorphic Stubai complex. Paragneiss and Orthogneiss are dominant, but Amphibolit and Mica-schist occur subordinately (Hoinkes and

Thöni, 1993). The study area is climatically characterized by the central-alpine dry region (Fliri, 1975). At Weißsee meteorological station (2470 m), a mean annual air temperature (MAAT) of −0.11 °C (2007-2019) and annual precipitation ranging from 731 mm to 1118 mm are recorded (data source: Tiroler Wasserkraft AG (TIWAG)). Detailed analysis of climate parameters of the study area is provided in Sect. 4.2 of this paper. According to a permafrost pseudo-probability map (Otto et al., 2020), 38% of the study area could be underlain by permafrost and 30% was covered by glaciers in 2015 (Buckel and Otto,

2018). The most prominent, Gepatschferner and Weißseeferner are located in the southern part of the catchment. Due to the glacier road, which makes the valley accessible by car, many studies of various disciplines of geo- and bioscience have been



carried out in this area (Dusik et al., 2015; Groh and Blöthe, 2019; Altmann et al., 2020). But the road and the associated ski area also cause anthropogenic influences on natural systems, which have to be considered.

A rock glacier inventory was compiled for the study area (c.f. Sect.3.1). Within the catchment, 54 rock glaciers can be found,
which were classified as active (20), inactive (16) and fossil (19). Due to poor image quality or snow cover (in one or more epochs of the historic data) and the activity status, the vast majority of these rock glaciers had to be excluded from the following analyses. However, eight active rock glaciers representing different characteristics and conditions were investigated in detail regarding flow velocities and nine rock glaciers regarding 3D displacements. Detailed characteristics of these rock glaciers can be found in Table A1. The most prominent of those is the well-studied and largest (0.237 km²) rock glacier Innere Ölgrube
(RG 01) (Fig. 2).

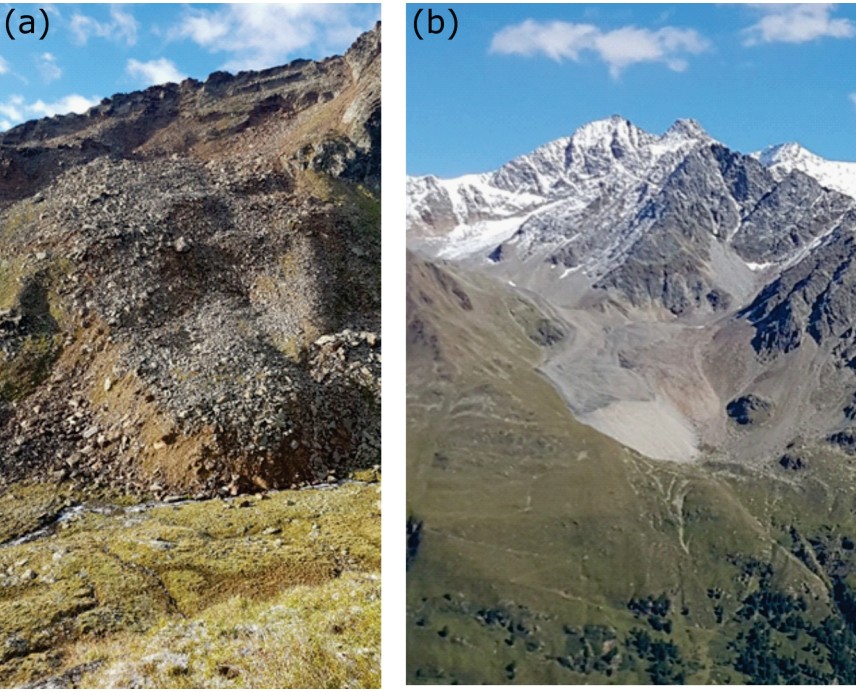

**Figure 2. Images of two of the studied rock glaciers. (a) the north-exposed rock glacier RG 08, located at the entrance to the Krummgampen Valley and covering an area of 0.088 km². (b) the well-studied, west-exposed Ölgruben rock glacier RG 01 covering an area of 0.237 km².**

On the Ölgrube, the first velocity studies were already carried out by Finsterwalder (1928) and Pillewizer (1957) and more recent studies continue their research and contribute additional information about its hydrology, internal and external structure and morphodynamics (Berger et al., 2004; Krainer and Mostler, 2006; Hausmann et al., 2012; Groh and Blöthe, 2019). The
area of the investigated rock glaciers ranges from 0.02 km² (RG 07) to 0.237 km² (RG 01). They show expositions of N, NE, E and W, with minimum elevation ranging from 2446 m to 2727 m. The elongated rock glacier RG 06 spans heights from 2702 m to 3093 m and thus reaches the highest elevation. Without geophysical, geochemical or petrographic information, interpretations about the genesis and internal structure are difficult (Berthling, 2011; Clark et al., 1998). In the case of the rock





glaciers RG 03, RG 04 and RG 09 a complete or partial covering of the rock glaciers by the LIA glacial extent (Fischer et al.,
2015) suggests a glacial genesis after 1850 or at least a glacial influence during and after this time as described by Dusik et al.
(2015) for RG 09. Some detailed studies about Innere Ölgrube (RG 01) exist, revealing information on internal structure
(Hausmann et al., 2012) and genesis (Berger et al., 2004). Berger et al. (2004) show that both lobes of the rock glacier
developed from debris covered glaciers after the peak of the Little Ice Age (LIA) (~1850). The rock glacier is composed of a
4 m to 6 m active layer, a 20 m to 30 m thick ice rich permafrost body and a underlying 10 m to 15 m ice free sediment layer
and has an ice content of 40% to 60% (Hausmann et al., 2012).

## 3 Materials and Methods

### 3.1 Rock glacier inventory

Although manual mapping of rock glacier landforms is shown to be highly subjective (Brardinoni et al., 2019), we tried to
minimize the heterogeneity in the inventory by incorporating the guide lines for inventorying rock glaciers (IPA Action Group
Rock glacier inventories and kinematics 2020) and only mandate one operator to compile the inventory on the basis of Krainer
& Ribis (2012).  Rock glacier outlines were corrected and additional landforms were mapped on the basis of the most recent
hillshade derived from the 2017 ALS campaign of the DFG founded project PROSA and an orthoimage of 2015 (data source:
Land Tirol - data.tirol.gv.at). Activity status was assigned according to morphological characteristics in combination with a
DEM of difference (DoD) of the 2012 and 2017 ALS campaigns and image correlation analysis on the derived hillshades
without local alignment of the data. In the case of active rock glacier complexes, which are overflowing fossil ones, they were
considered as two separate units.

### 3.2 Meteorological data

For the analysis of rock glacier morphodynamics over a decadal time period, a reference to climatic conditions that influence
such systems in various ways is indispensable. However, long time series data in the high alpine areas are only very
sporadically available, as early climate monitoring stations tended to be located in population centres. This also applies to our
catchment area, where the meteorological station Weißsee (2540 m) (data source: TIWAG) is recording data since 2006. For
this reason, we use additional data from nearby meteorological stations, which have longer time series available, to provide
information on the approximate climatic development in the catchment area. The locations of these stations are shown in
Figure 1, while an overview of the stations and the used data is given in Table 1.
In earlier studies, it is shown that changes in rock glacier morphodynamics can be related to the warming of permafrost through
heat conduction caused by an increase in air temperature (Roer, 2005; Kääb et al., 2007; Delaloye et al., 2010; Kellerer-
Pirklbauer and Kaufmann, 2012; Kellerer-Pirklbauer et al., 2018; Cicoira et al., 2019b). But especially on a shorter timescale,
it is very likely that rock glacier morphodynamics are also controlled by liquid water availability and snow cover parameters
(Ikeda et al., 2008; Wirz et al., 2016a; Kenner et al., 2017; Buchli et al., 2018; Cicoira et al., 2019a; Kenner et al., 2020). We






**Table 1. Overview of the meteorological stations used. Distance gives the distance to the center of the study area. T - Temperature; Pr - Precipitation, SC - Snow cover. The data were provided by the "Federal Misistry of Agriculture, Regions and Tourism" (BMLRT), the "Central Institute for Meteorology and Geodynamics" (ZAMG), "Historical Instrumental Climatological Surface Time Series of the Greater Alpine Region" (HISTALP), "Autonomous Province of Bozen/Bolzano " (Province BZ) and TIWAG**

| Station Name | Recording started | Distance [km] | Temporal resolution | Elevation [m] | Parameters analysed | Provider |
|---|---|---|---|---|---|---|
| Kaunertal-Vergötschen | 1895 | 18.1 | monthly, daily | 1269 | Pr, SC | BMLRT |
| Nauders | 1895 | 19 | daily | 1330 | T, Pr, SC | ZAMG |
| Obergurgl | 1953 | 21 | daily | 1942 | T, Pr, SC | ZAMG |
| Obergurgl-Vent | 1851 | 21.4 | monthly | 1938 | T | HISTALP |
| Plangeroß | 1895 | 16 | daily | 1605 | Pr, SC | BMLRT |
| Vernago | 1953 | 17 | daily | 1700 | Pr | Province BZ |
| Weißsee | 2006 | 2.4 | 15-min | 2540 | T, Pr, SC | TIWAG |


therefore analyse changes, anomalies and trends in air temperature, precipitation and snow cover for the whole period of investigation and between the individual epochs. In addition, we analyse temperature and precipitation on a seasonal basis. Snow cover onset and duration were determined according to Peng et al. (2013).

**3.3 Airborne Laser Scanning (ALS) data**

To analyse rock glacier flow velocities on hillshades (see Sect. 3.50) and 3D displacements in point clouds (see Sect. 3.6) in the two most recent epochs 2006-2012 and 2012-2017, we used data from different ALS campaigns (see Table 2). The most recent one was acquired on the 5th of June 2017 by a helicopter and a mounted mobile laser scanning system VuxSys-LR from Riegl (www.riegl.com). This ALS flight mission was carried out by the Chair of Physical Geography at the Catholic University of Eichstätt-Ingolstadt during the DFG-funded project (PROSA), achieving a mean point density of 20.0 pts/m² on the studied

rock glaciers. Due to weather conditions and organizational problems on the part of the contracted company, which made an area-wide data acquisition on one day impossible, the 2012 ALS data were recorded (also during the PROSA project) on the 4th and 18th of July. An LMS Q680i-S laser scanner from Riegl mounted on a helicopter was used for data recording. Depending on the date of recording, the average point density ranges between 12.3 pts/m² and 12.7 pts/m². Furthermore, an additional ALS data set from the 5th of September 2006 with an average point density of 5.0 pts/m² was provided by the

TIWAG. All datasets were georeferenced with parameters optimised by an automatic strip adjustment (Glira et al., 2015). To account for possible variable errors throughout the catchment, we locally fine registered the 2006 and 2012 point clouds to the 2017 data set for every single rock glacier by using an iterative closest point algorithm (ICP) (Besl and McKay, 1992) on mapped stable areas around each rock glacier.




**Table 2. Overview of ALS data used to derive DEMs and hillshades. These data were used for the 3D displacement analysis as well**

| Flight date | Scanner model | Wave length [nm] | Average point density* [pts/m²] | Operator |
|---|---|---|---|---|
| Sep. 5th 2006 | unknown | 999 | 4.95 | TIWAG |
| Jul. 4th 2012 | Riegl LMS Q680i-S | 1064 | 12.65 | Milan Geoservice GmbH (commissioned) |
| Jul. 18th 2012 | Riegl LMS Q680i-S | 1064 | 12.34 | Milan Geoservice GmbH (commissioned) |
| Jun. 5th 2017 | Riegl VuxSys-LR | 1550 | 19.97 | Chair of Physical Geography University of Eichstätt-Ingolstadt |

* Mean point density on studied rock glaciers with snow patches excluded

### 3.4 Generation of point clouds and orthoimages from historical aerial images

In order to extend the time series of ALS data (2006-2017) and to quantify the morphodynamics of the rock glaciers that
occurred in the previous century, we used aerial photos acquired at irregular intervals over the entire Austrian territory.
The useable aerial photos for the Kauner Valley catchment were collected at six separate epochs between 1953 and 1997. The
epochs were chosen based on data availability, similarity in acquisition date (i.e. late summer), image quality, and sufficient
image overlap. Note that the photos of 1953 were collected in three different flights on different days. However, considering
these three datasets separately was not possible for the 3D reconstruction due to lacking image overlap. Therefore, all the
images were processed together to generate one 1953 DEM.
The historical aerial photos used in this study were scanned and provided in tiff-format by "Office of the Tyrolean Government-
Department of Geoinformation" (https://www.tirol.gv.at/en/) and "Austrian Federal Office of Surveying and Metrology"
(BEV) (https: //www.bev.gv.at) along with the camera calibration protocols if available (Table 3).
Using advances in digital photogrammetry, particularly Structure from Motion (SfM) with Multi-View Stereo (MVS), the
reconstruction of 3D information in form of point cloud from scanned historical photos does not require specialized knowledge
(Bakker and Lane, 2017; Fawcett et al., 2019). The aerial images were processed in Agisoft Methashape (v.1.6.1) using the
film camera tool, which estimates the camera calibration parameters based on the fiducial marks. The software automatically
derived the locations of the fiducial mark in the images. Their distance in mm and the focal length were available from the
calibration protocol. Having defined the camera interior orientation, the camera exterior orientation, the 3D point cloud
reconstruction and the orthophoto generation follow the standard SfM-MVS workflow. This includes ground control points
(GCPs) measurement for georeferencing and dense image matching. The 3D coordinates of the GCPs were chosen from the
ALS 2017 point cloud on stable terrain and were evenly distributed throughout the catchment. The resulting average point
density on the studied rock glaciers varies from 1.2 pts/m² to 11.9 pts/m². The ground resolution of the orthoimages varies
between 0.2 m and 0.5 m.





The photogrammetric point clouds were co-registered with the reference 2017 ALS data in order to minimize inherent systematic errors (Bakker and Lane, 2017) and by using again an iterative closest point (ICP) algorithm (Besl and McKay, 1992) on mapped stable areas around the rock glaciers. Fine registration was performed for all individual rock glaciers and epochs separately to account for any variable errors throughout the catchment.

**Table 3. Overview of the acquired historical aerial image flights used to generate point clouds and orthophotos in Agisoft**
**Methashape Professional (v.1.6.1)**

| Flight date | Number of images* | Purpose | Source | Camera | Focal length [mm] | Scanning resolution [μm] | Flight altitude [m asl] | Resolution orthophoto [m] | Point density*** [pts/m²] |
|---|---|---|---|---|---|---|---|---|---|
| Jun. 5th Aug. 31st Sep. 8th 1953** | 124 | Forest condition estimation; Flight C | BEV | unknown | 210.11 | 15 | 4700 - 8300 | 0.225 | 6.6 |
| Sep. 1st 1954 | 36 | Forest condition estimation; Flight D | BEV | unknown | 210.11 | 15 | 10130 | 0.225 | 7.0 |
| Sep. 29th 1970 | 32 | Tyrolean state surveying flight | Land Tirol | Wild RC5/RC8 | 210.43 | 12 | 8665 | 0.2 | 10.1 |
| Aug. 18th 1971 | 91 | Tyrolean state surveying flight | Land Tirol | Wild RC5/RC8 | 209.48 | 12 | 5025 | 0.2 | 12.0 |
| Sep. 13th 1982 | 34 | Tyrolean state high altitude surveying flight | BEV | Wild RC10 | 152.58 | 15 | N/A | 0.5 | 1.2 |
| Sep. 11th 1997 | 25 | KF 173 | BEV | Wild RC10 | 152.70 | 15 | N/A | 0.5 | 1.2 |

\* Number of images used to reconstruct the whole catchment

\*\* Three Dates were processed as one dataset

\*\*\* Mean point density on studied rock glaciers with snow patches excluded



To account for possible shifts in the orthoimages, we resampled them at a resolution of 0.5 m and we locally co-registered all
individual rock glaciers for each epoch to the 1953/54 orthoimage. We used 9 to 29 co-registration points equally distributed
around the rock glaciers. We obtained co-registration root mean squared errors (RSMEs) between 0.225 m and 0.549 m with
an average of 0.316 m.

## 3.5 Horizontal flow velocities

### 3.5.1 Calculation of horizontal flow velocities

Horizontal flow velocities of the rock glaciers were calculated for the six processed time steps between 1953 and 2017. For
this purpose, an image correlation approach was chosen, which is a common method to derive glacier and rock glacier velocity
from orthoimage, hillshades and satellite images (Scambos et al., 1992; Kääb and Vollmer, 2000; Heid and Kääb, 2012;
Monnier and Kinnard, 2017; Kellerer-Pirklbauer and Kaufmann, 2018). In this study, orthoimage and hillshade image pairs
were utilized. For the time step 1997 – 2006, where orthoimage and hillshade were used in combination, illumination was
chosen for the 2006 DEM according to the exact position of the sun in the 1997 orthoimage. To calculate flow velocity vectors,
the image correlation algorithm IMCORR (Scambos et al., 1992) within SAGA-GIS software was applied. The algorithm
attempts to match small sub-scenes from two images by applying a fast Fourier transform-based version of a cross- correlation.
It can locally adjust the intensity values between two image pairs and therefore compensate for differences in illumination.
Using this algorithm, sub-pixel precision of displacement vectors can be achieved. We used search and reference chip size
combinations of 64/32, 128/64 and 256/128 with a fixed spacing of 5 m. The combinations were calculated for all image pairs
and the most reasonable was chosen for further analysis. This was done by visually analysing the resulting displacement vectors
in combination with the input data. In general, larger chip sizes were chosen for faster moving rock glaciers and/or long time
spans between the image pairs. The resulting raw vector maps can contain erroneous displacement measurements or
decorrelation, where no measurement is possible, due to snow, strong shading effects, areas where displacements are
dominated by rock fall and large displacements, which cause a change of texture. These vectors were excluded manually for
all time steps with the help of the matching orthoimages or hillshades. Subsequently, a mask was created for the areas where
measurements were possible in all time steps and just measurements in these areas were used for further analysis to make the
individual time steps comparable.

The combination of orthoimages and hillshades has to be chosen because low point densities in some of the aerial images
derived point clouds resulted in low details in the resulting DEMs. Tests regarding image correlation on these DEMs showed
very poor results. We are aware that the low point densities also affect the accuracy of the resulting orthoimages and outline
the variable errors in Sect. 4.1. On the other hand, we decided not to use orthoimages for the more recent epochs from 2006 to
2017, available from "Office of the Tyrolean Government-Department of Geoinformation" (https://www.tirol.gv.at/en/) for
the reason that they are orthorectified utilizing the most up to date DEM with a resolution of 5 m, which could result in




erroneous displacement measurements. If a non-matching DEM is used, it would lead to orthorectification errors particularly on moving landforms, like rock glaciers (Kaufmann and Kellerer-Pirklbauer, 2015).

The measurement of horizontal flow velocities of rock glaciers on remote sensing data, especially when using historical aerial images and their derivatives, is prone to errors. As described by Kääb et al. (2020), the error budget is composed of the

following components: 1) overall shifts between the orthorectified data 2) lateral shifts in the orthoimages due to errors in the DEM used for orthorectification 3) distortions in the aerial images or in the sensor model that propagate into the orthoimages 4) image matching uncertainties and errors. We minimized the shifts between the orthoimages by local co-registration of the orthoimages. By using the matching DEMs of the individual years for orthorectification, we addressed error type 2). However, quality of the DEMs varies locally in a single epoch and more crucially between the epochs and therefore are still a source of

error. The DEMs with the lowest quality were the epochs 1982 and 1997. These were also the years with the worst quality of the raw aerial images (error type 3). Another source of error when working with historical aerial images are scratches and alterations on the original image film caused by storage and age. These can lead to problems in the processing and thus were masked out before processing. Errors of type 4) contain errors caused by the image correlation method itself. The measurement errors as consequence of image correlation vary with the image quality like resolution, shadow, contrast and noise of the image

pairs (Kääb et al., 2020). We removed both directional and magnitudinal gross outliers manually by counterchecking the resulting displacement vectors with the corresponding orthoimage and hillshade pairs.

### 3.5.2 Error assessment for horizontal flow velocities

To quantify the overall error budget for horizontal flow velocities, we mapped close stable areas of similar texture/roughness and exposition on the single rock glaciers for all time steps. Due to snow and shading effects, these stable areas had to be

adjusted slightly for some time steps. Subsequently, displacement vectors in these areas were analysed for all individual epochs and rock glaciers. As no gross outliers were found in these areas, we used the mean value ($disp\overline{x}$) added by two times the standard deviation ($disp\sigma$) as measure for error budget ($error_{disp}$) of flow velocity measurements.

$$error_{disp} = disp\overline{x}_{\Delta epoch1-epoch2} + 2disp\sigma_{\Delta epoch1-epoch2} \qquad (1)$$


This measure was also applied by Fey and Krainer (2020) to determine a level of detection (LoD) for rock glacier flow velocity and recommended as a statistical measure of flow velocity error by Paul et al. (2017). We have decided not to use a LoD for calculating rock glacier flow velocity statistics for the fact that even in areas below the LoD there might be actual displacement (Anderson, 2019). We therefore rather illustrate the errors as red bars in Figure to give an assessment of the uncertainties and

data quality of the individual time steps and discuss possible implications in Sect. 4.1.



### 3.6 3D displacements on rock glaciers

We used both photogrammetric and ALS point clouds to measure the 3D displacements on the rock glaciers, which represent the surface change normal to the surface. The method described in the following can be used as a simple and robust alternative to a DEM of difference analysis in the 2D case, but offers some advantages in complex 3D cases, particularly on vertical to
near vertical and rough surfaces or if point densities are variable (c.f. Lague et al. 2013). The datasets for the 3D reconstruction differ slightly from the datasets used for the velocity analysis on the rock glaciers as the processing of the aerial photographs did not lead to sufficient point cloud resolutions for all of them. Thus the 1982 and 1997 epochs had to be excluded and the analysis could only be performed for the 1953-1970/71, 1970/71-2006, 2006-2012, and 2012-2017 epochs, where the 2006, 2012 and 2017 datasets represent ALS data. In 1970 and 1971, only a portion of the valley was covered in each case, so this
epoch had to be composed of two partial data sets.

As already described, the point clouds were locally fine-registered (c.f. Sect. 3.4) and then thinned (0.5 m) during the import process into LIS SAGA in order to produce homogeneous point densities for all epochs. To account for the sometimes very long time intervals between individual data sets and the expected high 3D displacements, we used the 3D distance between points approach by Fey & Wichmann (2017), which is based on Lague et al. (2013) and is recommended as a robust distance
measurement for geomorphological change detection in a complex terrain (Fig. A1). For each point, the normal vector for a best fitting plane (including the point neighborhood) with a radius of 5 m around the point was calculated in a first step (module "point cloud features" in LIS SAGA). Using this normal information of individual points, the approach (implemented in LIS SAGA) determines corresponding points in two point clouds of different epochs along the normal or flipped normal vector and derives the 3D distance (distance perpendicular to surface) and thus the 3D displacement between these corresponding
point pairs by using a distance threshold (10 m) and a threshold for the maximum normal difference (15°). The relatively high threshold for the radius was chosen to account for the expected high 3D displacements over the long investigation periods. In addition, we decided to use the same limit for all data sets to ensure comparability of the data. A detailed description of the workflow can be found in Fey & Wichmann (2017). Analogous to the rock glaciers, the calculations were also performed on the stable areas around the single rock glaciers for all epochs in order to calculate the measurement error. At the end of this
workflow, the 3D displacements for the rock glacier areas as well as for the stable areas were assigned as an attribute to the respective point clouds and were used for the following analyses. As for the flow velocities, we have omitted an LoD and have included the error values as additional information in the corresponding figures. The volume calculations were also performed based on the point cloud information. For this purpose, raster data sets (1m) were aggregated from the mean annual 3D displacment point cloud attribute and the volume was calculated from this raster dataset.



## 4 Results and Discussion

### 4.1 Errors and uncertainties

Despite efforts to minimise errors, in particular by image co-registration in the case of flow velocities and ICP in the case of 3D displacements, they are still present in the data. The errors are indicated in the corresponding Fig. 6, 7, 8 and 9. For flow velocity measurements, error values range between 0.013 m/yr and 0.088 m/yr. For the 3D displacements, the values range on all stable areas for all epochs from -1.38 m to 1.4 m with a mean value of -0.0007 m, a $\sigma$ of 0.014 and the 95% quantile of 0.02. The distribution of error values from individual rock glaciers per image pair combination are displayed in Fig. 3. In general, best results were obtained for epochs where hillshade image pairs were utilized. The largest errors were determined for the epoch 1997-2006, in which a combination of orthoimages and hillshade was used. We attribute this to the poor image quality of the 1997 dataset and the use of orthoimages in combination with hillshades. Although the illumination of the hillshades was adapted to the orthoimages, only contrasts and patterns, which are caused by the illumination, can be used to determine the flow velocity by image correlation. On the contrary, contrasts and patterns that are caused by different colours in the orthoimage are not taken into account or can even lead to incorrect measurements. In other cases, the variability of the errors may be related to the accuracy of the co-registration, but also to differences in the quality of the image in terms of contrast, illumination, and resolution.

To assert the validity of our results we performed a qualitive comparison with dGPS measurements, which were taken by Krainer and Mostler (2006) between 2002 and 2004 for Ölgruben rock glacier (RG 01), confirm the magnitude, flow direction and pattern that were determined by our method. In addition, a comparison of flow velocities derived by Groh and Blöthe (2019) by feature tracking for Kauner Valley rock glaciers between 2001 and 2015, showed good agreement. Although other time-steps were used, span, mean and median values show similar and plausible results when comparing them to the two most recent time steps of our study.

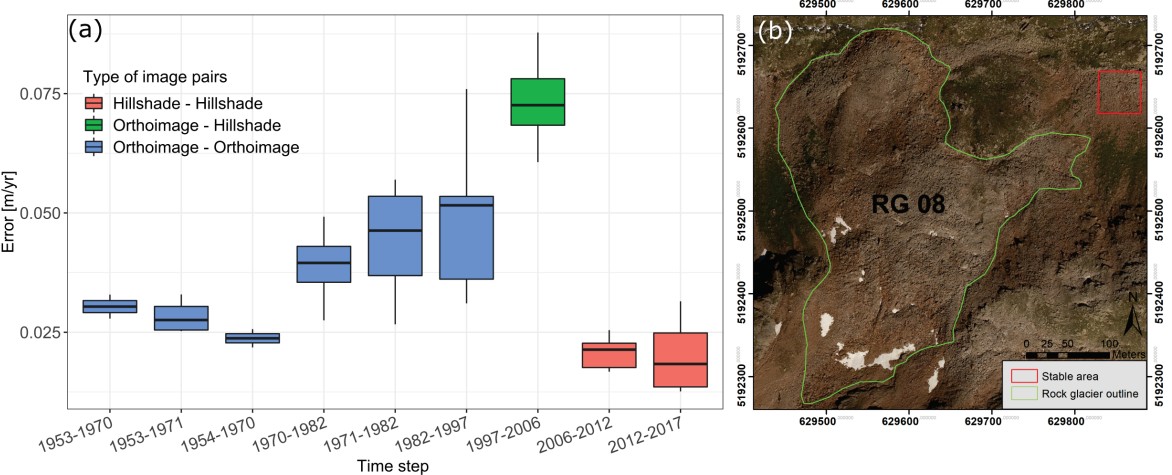

**Figure 3. (a) errors derived from stable areas in proximity to the individual rock glaciers for all image pair combinations. (b) example of a stable area displaying similar exposition, texture/roughness used to analyse error for RG 08.**



Due to the generation of masks with the aim of only taking into account areas in which measurements were possible in all
epochs in order to ensure better comparability, some areas of the rock glaciers had to be excluded (Table 4). In the case of the
flow velocity analysis, between 27.39% and 80.00% of the rock glacier area could be taken into account. In the case of the 3D
distance analysis, it was 50.50% to 95.67%.

**Table 4. Percentage of the rock glacier area in which measurements were possible in all time steps and which were therefore considered for the analysis of the flow velocity and 3D displacement. Reasons for exclusion were snow cover, shading effects or**
**decorrelation in the image correlation.**

| | Measurement area [%] | |
|---|---|---|
| Rock glacier | Flow velocity | 3D-distances |
| RG 01 | 41.97 | 92.09 |
| RG 02 | 75.94 | 87.72 |
| RG 03 | 45.42 | 70.93 |
| RG 04 | 65.72 | 90.71 |
| RG 05 | 27.39 | 81.10 |
| RG 06 | 39.12 | 64.40 |
| RG 07 | 70.62 | 92.90 |
| RG 08 | 80.00 | 95.67 |
| RG 09 | - | 50.50 |

## 4.2 Changes in meteorological forcing

As it is known from many studies that the changes in the external (meteorological) forcing (temperature, precipitation, snow
cover) play an important role for changes in the periglacial system and thus rock glaciers, we analysed climate data from
weather stations within and close to our catchments, which was challenging due to the large temporal scale (1953-2017). All
stations show similar patterns, even if the manifestation of the anomalies are slightly different in some cases. We note that the
positive trend of temperature increase is slightly higher for stations of higher elevation in the study period. In the case of
temperature and snow cover, we mainly present data from the stations Obergurgl-Vent (1938 m a.s.l.) and Obergurgl (1942 m
a.s.l.), as these are located at the highest elevation and only about 21 km away from the centre of our study area. In the case of
precipitation, we mainly present data from the Plangeroß station (1605 m a.s.l.), because although it is located at a significantly
lower elevation than the Weißsee station (2540 m a.s.l.) and the studied rock glaciers, it offers the best agreement with the
Weißsee station data in terms of monthly precipitation (r= 0.892, p<0.001). Wherever possible, we try to confirm the
observations with the limited time series (2007-2017) of the Weißsee station. In the following sections, we describe the general
trend of meteorological forcing which is further discussed with regard to changes in flow velocity in Sect. 4.4.



### 4.2.1 Temperature

During the period of investigation (1953-2017), the temperature trend shows an increase of 1.92 °C in 65 years at the Obergurgl-Vent (1938 m a.s.l.) station (Fig. A2). This is a stronger increase than at the lower elevated station Nauders (1330
m a.s.l.), where the increase is 1.29 °C. If only the last two epochs, 2006-2012 and 2012-2017, are taken into account, it becomes apparent that the station Weißsee (2540 m a.s.l.) shows the greatest increase in the mean temperature values of these time periods of 0.57 °C. Possible reasons and explanations for the elvation-dependend warming are given in Rangwalla and Miller (2012). In general, the temperature increase goes well in line with the alpine wide temperature increase, which has a significantly higher amplitude than the global average (Beniston, 2006). The seasonal development of the temperature trends
shows a stronger increase in temperatures in spring and summer of 2.73 °C and 2.64 °C, respectively, in 65 years compared to the winter and autumn temperatures. For these, the positive trend is clearly weakened and amounts to only 1.6 °C for winter and 0.69 °C for autumn. If one compares the seasonal temperatures with the reference period 1961 - 1990, it is noticeable that in the case of summer and spring temperatures, only positive anomalies occur from the beginning of the 1990s onwards (Fig. 6). The summers of 2015, 2017 and in particular the summer of 2003 are conspicuous for their high temperatures and are
known to have brought profound change to permafrost-affected systems (Ravanel et al., 2017). In the case of spring temperatures, the spring of 2007 stands out in particular, and to a lesser extent 2011 and 2017. Conditions causing these heat waves are expected to increase throughout the 21st century (Gobiet et al., 2014). Although positive anomalies also accumulate from this period onwards in the case of autumn and winter temperatures, there are also periods or years showing comparatively cold temperatures. Particularly warm winters were recorded in 1988, 1989, 2006 and 2015, with mean temperatures more than
3 °C higher than the long-term average. In the case of autumn temperatures, the warm autumn of 2006 stands out in particular. For the period before the beginning of the 1990s, periods/years with positive and negative anomalies are visible, whereby the strong anomalies are mostly in the negative range. The summer temperature anomalies show the lowest variance. Particularly striking is the period between 1970-1982 with its continuously comparatively low summer and autumn temperatures and a period of relatively cold winters from 1962-1970.

### 4.2.2 Precipitation

All considered meteorological stations, except for the station Nauders (1330 m a.s.l.), recorded a slightly positive trend in terms of precipitation, translating to an increase between 53 mm and 241 mm during the investigation period. The positive trend for the station Plangeroß (1605 m a.s.l.) is expressed in an increase in precipitation of 152 mm per 65 years (Fig. A3). The mean annual precipitation accounts to 931 mm/yr during the period of investigation and to 957 mm/yr at the Weißsee
station (2540 m a.s.l.) in the period from 2007 to 2017. The greatest positive trend is recorded in autumn (52 mm per 65 years), the least in winter (18 mm per 65 years). On average, most precipitation falls in the summer months, with an average of 362 mm/yr, and the least in winter, with an average of 171 mm/yr. Throughout the Alps, there is only a very small or no clear trend with regard to the annual precipitation development in the 20st century (Beniston, 2006). But during the 21st century, a decrease



in summer precipitation, an increase in winter precipitation and an increase in the frequency and intensity of extreme
precipitation events are predicted for the European Alps (Gobiet et al., 2014).

Looking at the precipitation anomalies, there is a clear increase from the mid-1990s for positive summer and autumn
precipitation anomalies, which is particularly pronounced in the period 1995-2002. For the 1999-2002 period and the most
recent time step from 2013-2017, this can also be seen in the spring precipitation (Fig. 6). Another period with clustered
positive precipitation anomalies can be observed in the case of spring and summer precipitation between 1963 and 1967 and
for autumn precipitation between 1972 and 1981, with both showing single years with just slightly positive or negative
anomalies. Relatively dry summers were recorded from 1954-1984 and 1990-1994, dry autumns from 1984-1991. In the case
of winter precipitation, there are some positive and negative anomalies, but these do not occur in clusters. Nevertheless, the
winters with high precipitation in 2011 and 2012 should be mentioned here.

### 4.2.3 Snow cover

The onset and duration of snow cover is described below for the Obergurgl station (1942 m a.s.l.), as this is the closest and
highest station with a long data series. As with the other parameters, the values are difficult to transfer to the study area, but
can provide an indication of general trends and anomalies in snow cover onset and duration (Fig. A4). Looking at the entire
study period from 1953 to 2017, there is a slightly negative trend both for the start of the snow cover and for its duration. This
fits in with the results of Olefs et al. (2020) , who found an elevation-dependent reduction in snow depth and duration in Austria
between 1961 and 2020, but this only applies to elevations below 2000 m.

Although this is not always the case, in general, the data show that when the onset takes place earlier, the duration of snow
cover is longer and vice versa (Fig. 6). Particularly striking is the period from 2007-2016 where snow cover onset anomalies
show consistently negative values and are also associated with a long duration of snow cover, particularly between 2007 and
2011. Although no long time series data is available for Weißsee station, the early onset and long duration of the snow cover
for this period can also be observed here. This effect is also evident in a period between 1956 and 1962 and in two shorter
periods from 1971 to 1974 and 1978 to 1981. The opposite is visible for the epoch 1983-1997, as the snow cover tends to set
in late and often melts again more quickly.

### 4.3 Flow velocities

### 4.3.1 General trends

For the whole period of investigation, we derived maximum flow velocities ranging between 0.089 m/yr and 1.721 m/yr and
mean flow velocities ranging between 0.035 m/yr and 0.485 m/yr for the eight individual rock glaciers (Fig. 4). All of the
investigated rock glaciers, except for RG 08, which has the highest maximum flow velocities in the epoch 1953 to 1971, show
the highest mean and/or maximum values either in the epoch 1997-2006 or 2012-2017. RG 01, RG 02, RG 03, RG 06, RG 07
and RG 08 show a distinct acceleration of flow velocities beginning in the epoch 1997-2006. This fits well with other studies





investigating rock glacier morphodynamics in the European Alps over a multi-decade period, all observing a significant acceleration of flow velocity in the early to late 1990s (Roer et al., 2005; Delaloye et al., 2010; Kellerer-Pirklbauer and Kaufmann, 2012; Micheletti et al., 2015; Scapozza et al., 2014; Hartl et al., 2016; Kaufmann et al., 2018; Kellerer-Pirklbauer and Kaufmann, 2018; Kellerer-Pirklbauer et al., 2018; Kenner et al., 2020).

All of the investigated rock glaciers of the Kauner Valley show an increase in flow velocities in the most recent epoch (2012-
2017) compared to the previous 2006-2012 epoch. For this period of time, measurements with a higher temporal resolution show an annual increase in flow velocities from 2005 or 2007/08 and a particularly strong increase from 2010 onward with a maximum in 2015 and a slowdown since then (Hartl et al., 2016; Bodin et al., 2018; Kaufmann et al., 2018; Kellerer-Pirklbauer and Kaufmann, 2018; Kenner et al., 2020).

Exceptions to the general trend can particularly be seen in the case of RG 04. This rock glacier is characterised by very low
and relatively constant flow velocities, which even decrease slightly in the two periods following 2006. Many studies mentioned periods of slight decrease or constant flow velocities following the strong acceleration in the 1990s (Delaloye et al., 2010; Kellerer-Pirklbauer and Kaufmann, 2012; Hartl et al., 2016). But there is no known example in the literature which shows a decrease in flow velocities over the entire surface area of the rock glacier in recent years, relative to the period at the beginning of the 1950s with no increase in the 1990s.

Although there is a general trend towards higher flow velocities for all rock glaciers, apart from RG 04, regarding flow velocity patterns and trends, there are different characteristics observable on the individual rock glaciers.

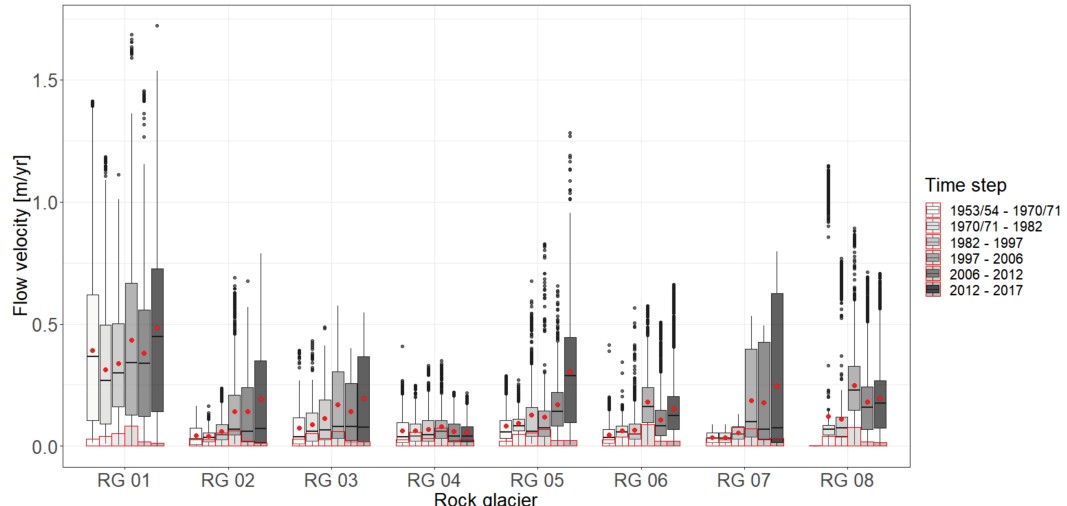

**Figure 4. Flow velocities for individual rock glaciers for six distinct epochs from 1953 to 2017. Rock glacier KT 09 was not included in the flow velocity analysis and rock glacier RG 08 had to be excluded from time step 1 due to extensive decorrelations in the frontal part because of large displacements.**



### 4.3.2 Magnitude of change

Although some rock glaciers show their minimum mean flow velocity in the epoch 1970/71-1982 and some their maximum
in the epoch 1997-2006, in the following chapter we compare the magnitude of the velocity increases between the first and the
last epoch.

In the first epoch, all rock glaciers, except for RG 01 (0.393 m/yr) and RG 08 (just manual mapping possible) which show
higher values, display similar mean values ranging from 0.035 m/yr to 0.082 m/yr. If the maximum flow velocities are taken
into account, this similarity is put into perspective somewhat, as the range here is significantly greater and lies between 0.089
m/yr and 0.409 m/yr. In the epoch 2012-2017, the range of the mean values increases significantly and is now between 0.056
m/yr and 0.305 m/yr and 0.485 m/yr for RG 01. The strongest proportional increase can be seen in rock glaciers RG 07 and
RG 02, where mean flow velocities increased by 602.00% and 356.03% and maximum flow velocities increased by 799.10%
and 383.20%, respectively. Beside the rock glacier destabilisation that has been observed for some rock glaciers in recent years
(Roer et al., 2008; Scotti et al., 2017), these are the highest relative movement changes compared to other studies (c.f.Roer,
2005; Kellerer-Pirklbauer and Kaufmann, 2012, 2018; Kenner et al., 2020). In contrast, in the case of RG 04, the average and
maximum flow velocity is reduced by -12.51% and -47.14%, respectively. Since rock glaciers RG 02, RG 07 and RG 04 show
similar exposition, size, and elevation ranges, we assume that the different behaviour is explained by a topographic or structural
control rather than different external forcing.

The relative changes regarding the remaining rock glaciers ranges between 23.45% and 271.87% for mean flow velocity and
21.77% and 348.20% for maximum flow velocity. If RG 04, which has a very low slope, is not taken into account, one could
say that higher elevated rock glaciers change their relative flow velocity to a greater extent. Apart from this observation, none
of the topographic factors could explain the different magnitude of the change.

### 4.3.3 Local temporal peaks – sensitivity of rock glaciers to external forcing

On rock glaciers RG 01 and RG 08, higher flow velocities have been measured between 1953/54 and 1970/71 compared to
the subsequent periods. Although there are not many studies covering this period, this phenomenon has also been observed in
case studies on other individual rock glaciers in the Austrian (Kellerer-Pirklbauer and Kaufmann, 2012; Kellerer-Pirklbauer et
al., 2018) and Swiss Alps (Kenner et al., 2020) and is explained by decennial variations in mean annual air temperature
(Delaloye et al., 2010). In the epoch 1997 to 2006, higher flow velocities were measured compared to the epochs before and
after. This is especially the case for rock glaciers RG 01, RG 03, RG 06 and RG 08, although caution is required in the
interpretation due to the higher error values. Those higher velocities might be explained by high flow velocities in the years
2000/1, 2003/4 and 2006/7 reported for many rock glaciers in the Alps, due to high summer temperatures (Delaloye et al.,
2010). In addition, Kaufman et al. (2018) also found considerably higher mean flow velocities in the period from 2002-2006
than from 1992-2002 and 2006-2009 for Tschadinhorn rock glacier, Hohe Tauern (Austria). These peaks might not be found
on the other rock glaciers due to superimposing effects over the long time steps and indicate a slightly different sensitivity,



response or response time of individual rock glaciers to intra-annual, inter-annual or multi-annual fluctuations in external forcing parameters. In the case of the flow velocity peak 1953/54-1970/71, this occurs on the three investigated rock glaciers displaying the lowest elevation and differing substantially in the other characteristics measured. The rock glaciers with higher flow velocities in the period 1997-2006 cannot be clearly related to the rock glacier characteristics and it may also be due to the higher error, resulting in a misinterpretation.

### 480   4.3.4 Patterns of flow velocity and elevation dependency

To better place the general trends in a spatial and temporal context, flow velocities in individual epochs and rock glaciers were analyzed for altitudinal zones of 20 meter along the rock glacier surface. Fig. 8 clearly shows the different size and height coverage of the various rock glaciers. Furthermore, it becomes apparent that many rock glaciers do not move uniformly, but have zones with higher and lower flow velocities. The zones of higher flow velocity are usually, but not always, located in the

rock glacier front. Exceptions to this are mainly RG 03, which shows a gradual change in its flow pattern over time (see Sect. 4.6.1), RG 06 which in particular shows its maximum flow velocities in an area above a thermokarst depression (see Sect. 4.6.3) in the latest epoch and RG 08 where maximum flow velocities in the front gradually decrease and are shifted to higher elevations (see Sect. 4.6.2).

If we look at the percentage increase in the mean values of the last two epochs, the two lowest elevated rock glaciers, RG 01

and RG 08, show the smallest increases, while the sharpest increase was measured on the highest elevated rock glacier RG 05, whereas the remaining rock glaciers showing relatively similar increase rates. This could point to the fact that from 2006 to 2017 higher elevated rock glaciers enter an unstable state as a reaction of changes in the external forcing. Since rock glaciers tend to react with a certain time lag to changes in e.g. temperature, our data cannot be used to make any statements about e.g. temperature limits or similar. As it is very likely that such a time lack differs between single rock glaciers, we cannot find a

general elevation-dependent temporal change regarding the elevation classes.







**Figure 5.** Flow velocities per elevation class for all individual rock glaciers and six epochs between 1953 and 2017. In the plots, flow velocity maps of the individual rock glaciers are displayed for all time steps





### 4.4 Possible implications of changes in external forcing for rock glacier flow velocities

Kenner et al. (2017) synthesise findings for external factors controlling rock glacier flow velocity. Accordingly, an increase in the permafrost temperature, which changes the viscosity, hardness, and shear- and crushing strength of the permafrost ice, can thus increase its internal plastic deformation. Another factor would be the increase in water availability and water pressure,

which reduces the friction resistance in the shear zone. The former is primarily determined by changes in air temperature leading to changes in ground temperature and the timing and duration of snow cover. The latter can be controlled by precipitation, snowmelt, the formation of new drainage systems and melting permafrost ice. As our analysis covers periods of 5 to 17 years, it is difficult to identify individual meteorological factors that cause changes in rock glacier morphodynamics, as possible influences and reactions superimpose. In addition, in situ measurements of permafrost temperature, water

availability and the formation of new drainage systems were not the goal of this study. Nevertheless, the following sections describe possible implications of changes in the meteorological forcing (see Sect. 4.2) based on the development of the flow velocity for the six epochs between 1953 and 2017 (Fig. 6).

### 4.4.1 Temperature

As described by numerous studies, this development of temperatures fits well with the development of flow velocities (e.g.

Roer, 2005; Kääb et al., 2007; Delaloye et al., 2010; Scapozza et al., 2014; Hartl et al., 2016; Kenner et al., 2017; Kenner et al., 2020). Even though our rock glacier analysis started with the aerial images from 1953, we had a look at temperature data before this date since rock glacier dynamics can have a significant temporal delay. Although not covered in Fig. 4, we observed exclusively positive temperature anomalies ranging between 0.5 °C and 1°C between 1946-1951. Relatively warm temperatures were measured throughout the Alps during this period (Beniston, 2006). This could be a possible explanation for

the local peak in flow velocities of RG 01 and RG 08 between 1953 and 1971 and is also suggested as an explanation by Delaloye et al. (2010). However, this would also imply either that these rock glaciers take longer to react to the increase in temperature or that they take longer to slow down after this increase compared to the other rock glaciers studied. It could also indicate that the remaining rock glaciers have not yet reached a certain system state and have therefore hardly or not at all reacted to the increased temperatures of this period.

When looking at the strong increase in flow velocities from 1997 onwards, it turns out that the spring and autumn temperatures may primarily be responsible for the increase, as the average winter time temperature actually decreases and the autumn mean temperature remains constant in the case of the sharp increase in the epoch 1997-2006. This is also supported by the fact that period of high positive winter anomalies between 1982 and 1997 did not lead to an increase in flow velocities. As the positive spring and summer anomalies already began in the first half of the epoch 1982-1997, while a sharp increase in flow velocities

is only evident from 1997 onwards, it shows that the increase in flow velocity, if simply controlled by temperature, is slightly delayed.



**Figure 6. (a)** Mean flow velocities of individual rock glaciers for six epochs between 1953 and 2017. **(b)** Annual mean air temperature anomalies and **(c)** seasonal mean air temperatur anomalies at the weather station Obergurgl-Vent (1938 m a.s.l.). **(d)** Annual total precipitation anomalies and **(e)** seasonal total precipitation anomalies at the weather station Plangeroß (1605 m a.s.l.). **(f)** Snow cover onset and **(g)** duration at the weather station Obergurgl (1942 m a.s.l.). For (c) and (e) spring is defined as March-May, summer as June-August, autumn as September-November and winter as December-Februry.





This could be due to a delayed warming of the permafrost ice or to the duration of the formation of new drainage systems (Kenner et al., 2017; Kenner et al., 2020), which also might explain the varying magnitude of the increases. The local peak of

some rock glaciers between 1997 and 2006 could be explained by the particularly strong increase in spring temperatures or by the heatwave in the summer of 2003, which has also led to very high flow velocity rates in annual studies (e.g.Kellerer-Pirklbauer and Kaufmann, 2012, 2018). The further increase in flow velocities in the epoch 2012-2017 could be due to the fact that, in addition to spring and summer temperatures, winter and autumn temperatures also show exclusively positive anomalies in this time step. The low, otherwise constant or falling flow velocities in the period 1970/71 to 1982, on the other hand, fit in

well with the relatively low summer and autumn temperatures.

### 4.4.2 Precipitation

While many studies, especially recent ones, emphasise the role of liquid water in rock glacier movement (Ikeda et al., 2008), especially in the shear horizon (e.g.Kenner et al., 2017; Cicoira et al., 2019a; Kenner et al., 2020), only some show a correlation between precipitation and movement (Micheletti et al., 2015; Hartl et al., 2016; Eriksen et al., 2018), while others find no or

only a weak connection (Kenner et al., 2017; Kenner et al., 2020). In the latter, rock glacier acceleration is explained, among other things, by an increase in runoff efficiency due to the formation of new drainage pathways in the permafrost body.

In our study, the development of flow velocities also corresponds well with the development of precipitation. In the epoch from 1953 to 1997, no clear accumulation of positive or negative anomalies can be observed in the individual time steps. In case of the epoch 1971 to 1997, the negative summer anomalies are balanced out by positive autumn anomalies and vice versa.

In the epoch from 1997 to 2006 and between 2012 and 2017, positive precipitation anomalies occur for spring as well as for summer and autumn. This does not apply to the epoch from 2006 to 2012, where only positive summer anomalies are increasingly observed. This fits in very well with the determined flow velocities and could be a further explanation for the strong increase from 1997 to 2006, a constant or slight drop between 2006 and 2012 and a renewed increase from 2012 to 2017.

### 4.4.3 Snow Cover

Snow cover onset and duration have been shown to be important factors in the development of rock glacier flow velocities, as it controls the time span of liquid water availability as well as the temperature in the subsurface due to the winter cooling intensity (Kenner et al., 2017; Kenner et al., 2020). As for the other two parameters, temperature and precipitation, links can be found between the temporal development of snow cover onset and duration and the evolution of flow velocities in the rock

glaciers studied. In the last three epochs from 1997 to 2017, the snow cover sets in relatively early. In combination with the amount of snow and the temperature, this can decrease the rock glacier deceleration in winter, by isolating the rock glacier from cold winter temperatures, which in turn favours warming in spring and summer. This could have led to an increase in the flow velocity, especially in the most recent time period, since in addition to the early onset of the snow cover, only positive temperature anomalies occur in autumn. The decrease or constant flow rate in the 2006- 2012 epoch could be explained by the



relatively long duration of the snow cover, which leads to a shorter availability of liquid water. In the period before 1997, it is

more difficult to establish a connection. This may be due to the fact that the time periods are larger and thus positive and

negative anomalies balance each other out, but possibly also to the fact that the factor snow cover must always be seen in

connection with the temperature, which only changed drastically from the beginning of the 1990s. This possibly has led to the

formation of new drainage systems, causing a tipping point of flow velocities to a higher level, which in turn might change the

value on the influence of the snow cover and precipitation on the flow velocities.

## 4.5 Surface and volume changes on rock glaciers

### 4.5.1 General temporal trends of all rock glaciers

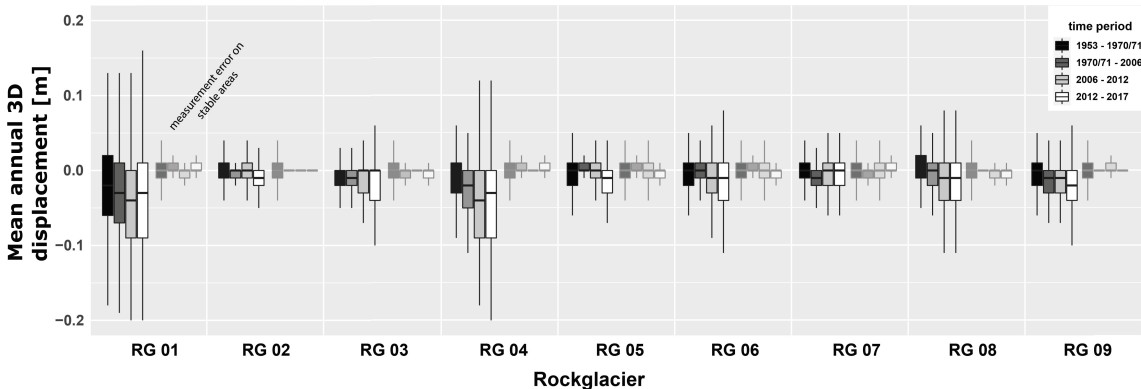

**Figure 7.** **Mean annual 3D displacements on the single rock glaciers and corresponding stable areas for the four epochs.**

The calculation of the mean annual 3D displacements could be carried out on a total of 9 rock glaciers in four epochs. Figure

7 shows the values of these 3D displacements on the rock glaciers and additionally the values calculated on stable areas, in

order to give an estimation of the accuracy of the measurements. The mean values range from 0.031 (RG 05, epoch 1970-

2006) to -0.047 (RG 04, epoch 2006-2012). In the entire study period, the values are predominantly in the negative range, with

only 4 rock glaciers showing positive values very close to zero. These values are in good agreement in terms of their order of

magnitude with Kellerer-Pirklbauer & Kaufmann (2018) and Kaufmann et al. (2018), who found values of -0.016 m/yr to -

0.058 m/yr and 0.0008 m/yr to 0.013 m/yr for two rock glaciers in the Austrian Alps over a similar study period. For most rock

glaciers, except RG01 and RG 04, the mean values of the 3D displacements in the first epoch are very close to zero and show

a sharp shift to the negative in the epoch 1970/71-2006 or 2006-2012. It is clearly visible that almost all rock glaciers show a

scattering of the 3D displacements in the negative and the positive range. This scattering can be explained by the flowing

movements and the resulting complex topography of migrating ridges and associated depressions on the one hand and the

partial "pushing forward" of the entire rock glaciers on the other. Over time, there is a clear tendency towards negative values,

which is also visible in the mass balances of the individual rock glaciers (Fig. 8). Those are decreasing constantly, suggesting

a successive subsidence of the landforms. Differences appear in the investigated rock glaciers with respect to the magnitude



of the 3D displacements. This is particularly evident in rock glaciers RG 01 and RG 04, where both the mean values and the
scatter are significantly more pronounced compared to the others, especially for the epochs since 2006. Both rock glaciers
differ with respect to size, slope, spanned altitudinal zones and relief, so that these parameters do not seem to be causal for this
behavior. In contrast, the other rock glaciers show more or less similar scatter and similar magnitudes of 3D displacements.
Also with regard to the temporal course, the rock glaciers cannot be divided into different subgroups. Here, one main group
exists, consisting of RG 01 and RG 03 to RG 09. Those show a constant increase in the dispersion of 3D displacements with
a clearly visible shift of the mean value into the negative range. The values tend to increase from 1953 to 2012 and in the epoch
2012 to 2017 remain roughly at the level of the epoch from 2006 to 2012 or decrease slightly. An exception is RG 02, which
shows 3D displacements between 1953 and 2012 that scatter very close around zero with a slightly negative trend in the second
epoch between 1970/71 and 2006. Overall, however, the changes are within the range of changes on the associated stable
surfaces, so that actual 3D displacements cannot clearly be assumed here. Only for the last epoch of the study period from
2012 to 2017, a clear shift of the 3D displacements into the negative range is visible. Even if these changes are on a low level,
it can be assumed that RG 02 changes significantly later in time than the other rock glaciers.

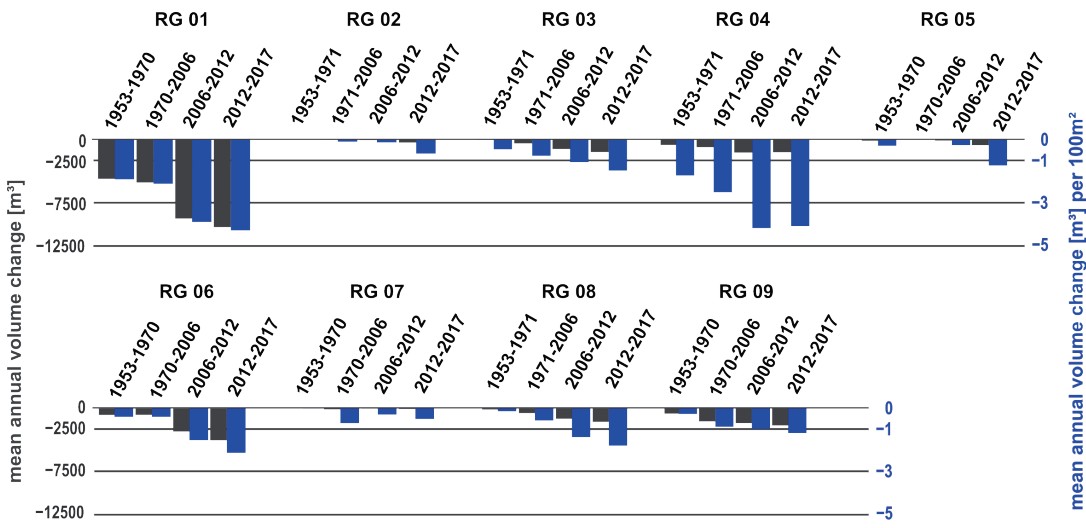

**Figure 8: Volume change of the individual rock glaciers for four epochs between 1953 and 2017.**

**4.5.2 Temporal and spatial trends of the single rock glaciers**

To better place the general trends in a spatial and temporal context, 3D displcements in individual epochs were analyzed for
individual elevation zones. Figure 9 shows the measured 3D displacements in the individual epochs for 20 m elevation zones
along the rock glacier surface and the corresponding maps. In addition, based on the maximum extent of each rock glacier, the
volume change within each epoch was calculated. The data show that the individual rock glaciers span different elevation
levels and that some of the studied objects have very different sizes. Within each rock glacier, there are zones of positive 3D





displacements as well as areas of negative change. These changes are mostly spatially clustered, but in some cases they also show a clear temporal clustering.

Overall, the picture already described for the general trends is confirmed. Thus, all rock glaciers show a very clear dispersion of 3D displacements into the positive and into the negative value range. The fact that this scattering can be explained by the
movement of the rock glacier becomes very clear when looking at the spatial patterns on the maps. Thus, areas with negative changes are followed by positive changes downslope. Therefore, the characteristic topography for rock glaciers is formed (Frehner et al., 2015) or the rock glacier advances. Furthermore, Fig. 7 and Fig. 9 show that changes in activity occur between the individual epochs. These changes in activity correspond to the general trends already described, but are again spatially more finely resolved in Figure 9. This change between epochs is particularly clear in RG 08. Thus, after a high activity phase
in the area of the front of the rock glacier in the first two epochs, these zones become almost inactive in the epochs after 2006 (only subsidence are still visible later). On the other hand, RG 02 and RG 07 show hardly any 3D displacements in the epochs before 1970 and 2006, respectively, and show subsidence thereupon (c.f. Sect. 4.6.2).

Nevertheless, the trend towards stronger negative 3D displacements is also confirmed for the other rock glaciers. Here, spatial patterns also exist, suggesting that active and inactive areas are shifting, but a clear elevation dependence is not apparent. The
shift in patterns is apparently more due to the dynamics of the individual rock glaciers and less a consequence of elevation. An exception is RG 05, whose rock glacier area extends over a large altitudinal range and whose highest areas are above 3000 m. Here, areas exist that apparently show very clear activity in terms of subsidence only in the last two epochs starting in 2006. This fits well with the statements about flow velocities, but a general altitude-dependent trend in the rock glaciers cannot be derived from the 3D displacement data.

Looking at the absolute volume change in the individual epochs, it becomes clear that all rock glaciers show negative volume balances in almost all epochs.  Exceptions here are RG 02 and RG 07, which show almost no volume loss in the first epoch, and RG 05, for which this was the case in the second epoch. Overall, however, the negative volume changes increase very significantly from epoch to epoch. This trend is almost linear for the rock glaciers RG 02, RG 03, RG 06, RG 08, RG 09, but with different characteristics. RG 01 as the largest rock glacier also shows the highest volume changes overall, although a
jump in the magnitude of the volume changes can be identified here after the second epoch. RG 04, on the other hand, shows a linear trend from the first to the third epoch, which then reverses from 2012 onwards.

A temporal outlook is not possible since none of the rock glaciers became completely inactive during the observation period and could be used as a reference. Rather, some of the rock glaciers have shown that an inactive phase can be followed by more active phases. An exception may be at RG 04, where a reversal of the trend in volume changes is visible after 2012, which
could indicate that here the ice body could thaw in the medium term and thus the rock glacier could become inactive.

A better picture regarding trends would be obtained if the analysis of 3D displacements were combined with the analysis of movement rates. This would allow conclusions to be made as to whether rock glaciers tend to successively sink, followed by a decrease in velocity or whether subsidence is in line with velocities. This synopsis will be done in the following chapter.





Figure 9. 3D displacements per elevation class for all individual rock glaciers and four epochs between 1953 and 2017. In the plots, maps of 3D displacements are displayed for all time steps.



### 4.6 Special cases

#### 4.6.1 Antropogenic influence on rock glacier morphodynamics

In addition to the influences of natural external forcing, the Kauner Valley also provides a good insight into the consequences
of anthropogenic interventions in high mountain landscapes. This is also visible with regard to the rock glacier dynamics. As
for example we attribute the change in flow velocity pattern on RG 03 to such anthropogenic influences. In the time between
1979 and 1982, the Kauner Valley glacier road was built which intersects the rock glacier in its upper part. This lead to a
separation of the upper and lower parts of the rock glacier and results in considerably higher flow velocities as well as increased
3D displacements being measured in the area directly below the constructed road in the following periods, especially from
1997 onwards. We suspect that this deep disturbance of the rock glacier has enabled a more efficient heat transfer, even into
the deeper layers of the rock glacier, and that new and more efficient drainage systems have developed below the road. This
disturbance of the rock glacier system due to a change in the thermal and hydrological regime in combination with a change
of external meteorological forcing could explain the change in the flow velocity and 3D displacements pattern. Although a
change can already be observed in the 1982-1997 epoch, a strong increase in flow velocity was measured since 1997, which
makes a delayed reaction of the rock glacier to the road construction 17 years before very likely.

#### 4.6.2 Inactivation vs reactivation of rock glaciers

It is known that rock glaciers can become inactive due to topographic effects or the loss of ice. Both factors may be linked to
the velocities and also to negative 3D displacements (subsidence), which are governed by external forcing as well as internal
forcing (slope, altitude and the presence of ice). Regarding topography and elevation, it is evident that there are rock glaciers
in our watershed that tend to become inactive due to low slope and/or volume loss since 1953. This can be clearly observed in
the example of RG 04, which shows strong subsidence accompanied by comparatively low and constant flow velocities
throughout the study period and has even slightly decelerated since 2006. Another example shows that also parts of a glacier
can become inactive. This is observable on RG 08, where the flow velocities exceed 1.5 m/yr in 1953-1971, progressively
decrease until 2006 and are close to zero in the two most recent epochs. In the 2006-2017 epoch, a slight subsidence is still
visible in this area, with a decreasing trend in the latest epoch. In contrast, as for most of the rock glaciers in our catchment,
an acceleration of flow velocities is visible in the upper part of this rock glacier since the epoch 1997-2006. Here, we assume
that the topography and altitude in the area of the front in combination with the separation of the upper part favour this
inactivation, whereas apparently the upper areas have subsequently left their stable system state and started to accelerate.
Whether changes in external forcing or the strong movements at the rock glacier front before 2006 are the cause of these
changes cannot be answered conclusively with our data.
In contrast, RG 07 and to a lesser extent RG 02 show flow velocities until 1997 that are barely above the error value and
furthermore show hardly any detectable 3D displacements at least in the first epoch. This is followed by the strongest relative
acceleration that is present in our catchment. Such a behaviour has already been described by Michelleti et al. (2015) for a



small rock glacier in the Hérens Valley (Switzerland). In our view, the large changes in the flow velocity of RG 02 and RG 07
can be seen as a reactivation, although flow patterns on the rock glacier surface suggest that flow velocity before 1997 was
probably just too small to be detected by our method. The sudden and sharp change might indicate a change in the internal
structure of the rock glacier system and mechanism of flow.

### 4.6.3 Thermokarst phenomena

Thermokarst is a widespread phenomenon in periglacial landscapes, and such thermokarst depressions can sometimes be filled
with water forming lakes (Soare, 2021). This might become a more common feature on rock glaciers due to warming and
degradation of permafrost which favours the formation of such depressions also in alpine terrain (Kääb and Haeberli, 2001).
Thermokarst depressions can be observed on RG 04 and particularly on RG 06. Here, in the course of the observation period
since 1953, a thermokarst lake developed, which subsequently changed in extent and shape and shifted its location by about
40 m, meaning that this thermokarst depression is part of the moving rock glacier system.
These thermokarst depressions are not only characterised by severe subsidence, but also affect the flow velocity of the
surrounding areas when they occur on active rock glaciers. This is evident on RG 06, where high flow velocities were measured
in the area above and below the thermokarst lake during the entire study period and especially from 1997 onwards. This
phenomenon can also be observed on RG 04 where a hot spot of higher flow velocities is found around a thermokarst
depression, which decrease since 1997. Here, displacements directions are measured contrary to the main flow direction
towards the depression, a phenomenon also described in Kellerer-Pirklbauer and Kaufmann (2018).
This indicates that the more efficient heat conduction also in deeper permafrost layers through the depression in combination
with the sinking motion and the liquid water, released by the strong melting of the permafrost leads to an increase in the flow
velocities in such areas. In the case of RG 06, the thermal influence of the lake water and the intermittent drainage play surely
a role in the development of flow velocities especially in the areas below the lake.

### 4.7 Analysis of velocity and 3D displacements

On the one hand, the movement of the rock glacier body can result in a change in the complex surface topography, as negative
3D displacements are often linked to positive 3D displacements and characteristic flow lobes are formed and/or the rock glacier
advances in the area of the front. In the case of flowing rock glaciers in the area of shear surfaces, despite the movement of
parts or the whole rock glacier body, there should be no or only a slight volume changes of the rock glacier.  On the other
hand, the movement of a rock glacier can also be the consequence of melting of the ice body or the pore ice. If such melting
zones are at zones with higher slope inclinations, the rock glacier tends to both move and subside, but with a tendency to a
negative volume balance. If slope inclinations are low, the rock glacier tends to subside with no or only slight movement,
which also leads to a negative volume balance. Therefore, it seems reasonable to analyze flow velocities and 3D displacements
together, as such an analysis can provide clues to the system state of a rock glacier and regarding a multi-temporal analysis
possibly the trends in its behavior.



Thus we plotted the mean values of the 3D displacements against the mean values of the flow velocities, where the scatter of the data is represented by the circles and ellipses (Fig. 10). For this analysis, data from the 1953-1970/71, 1970/71-2006, 2006-2012, and 2012-2017 epochs were used in order to make changes of this interaction over the entire study period visible.

From the analyses of flow velocities and 3D displacements in the previous chapters, it was expected that a majority of the rock glaciers would have overall flow velocities and 3D displacements of a similar magnitude. This was also expected for the relationship between flow velocity and 3D displacements. Both are visible in Fig. 10 for RG 02-RG 07. Exceptions are RG 01 and RG 04, which show complete different magnitude as well as a different relationship between velocity and 3D displacements compared to the other rock glaciers.

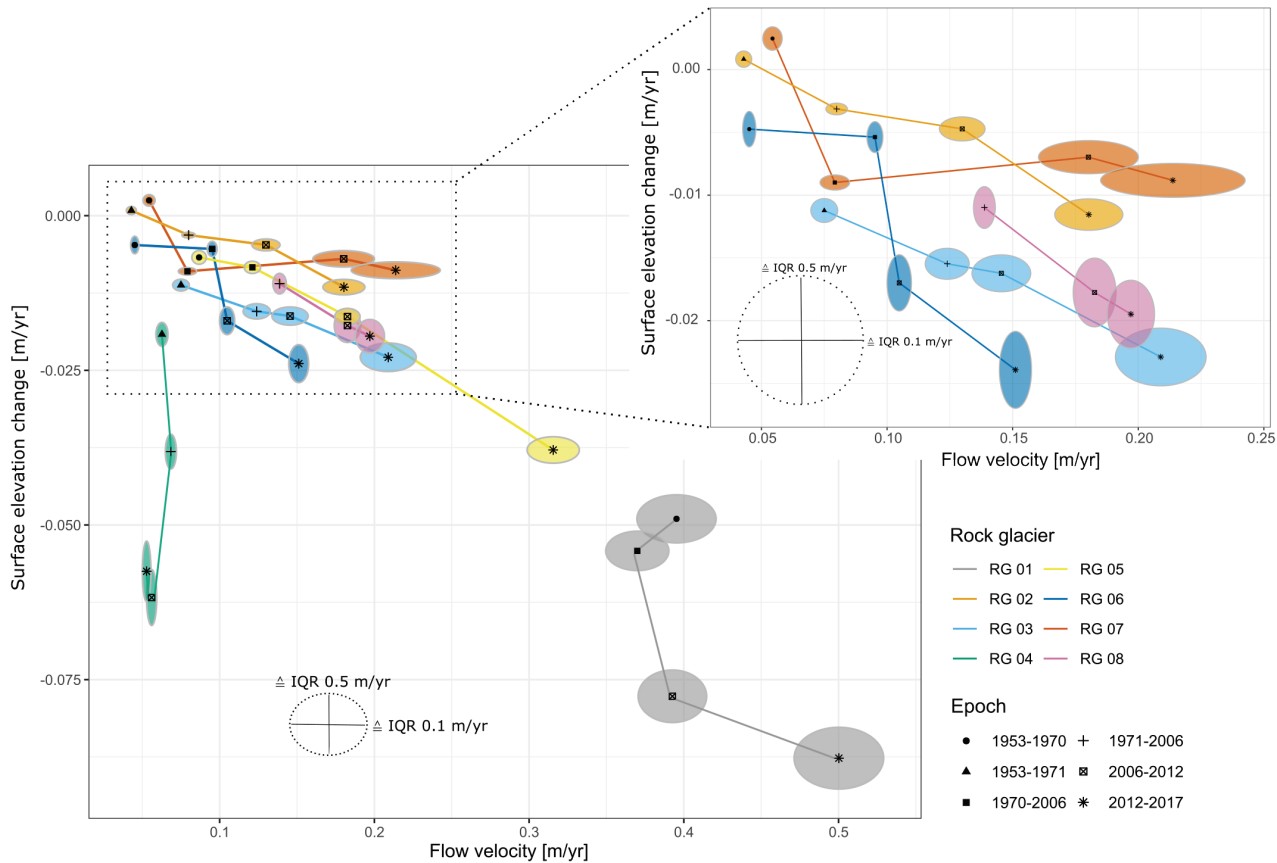

**Figure 10: Flow velocity rates plotted against 3D displacement rates. (a) shows all the investigated rock glaciers. (b) shows a subset of rock glaciers displaying a similar value range. The interquantile ranges are shown in scale in the illustrations, which is represented by the dashed ellipses.**

While RG 01 as the largest rock glacier of the catchment shows the highest velocities and the highest 3D displacements at all, RG 04 shows no or only a slight velocity, but very high 3D displacements. It is very obvious that in the case of RG 04 thawing permafrost is involved and that this thawing has successively increased from 1953 to 2012. Particularly striking then is the





reversing trend in the epoch 2012 to 2017, where the 3D displacements are lower than in the previous epoch, but are still at a higher level than in the epochs between 1953 and 2012. This fits also well with the volume losses in Fig. 10. The changes in flow velocities, on the other hand, are minimal and move along the zero value with very little scatter, which fits very well with the low slope gradients (20°) in the area of this rock glacier, which probably prevent movement of the rock glacier despite thawing permafrost. Although it remains to be seen whether the emerging trend will continue since 2017, a successive inactivation of this rock glacier seems to be very likely due to the available data and the visible trend.

A different behaviour is visible for RG 01, as this rock glacier shows only slight changes in the flow velocities in the first three epochs (slight decrease between 1970-2006), but a clear increase in negative 3D displacements between 2006 and 2012. This clear increase was already noted in Sect. 4.5.2 as a jump in magnitude after the second epoch, but this jump is not directly linked to a jump in flow velocities. This jump has obviously a delay, as it only becomes visible in the last epoch between 2012 and 2017. Both, the significant jump in magnitude as well as the delay could be the consequence of the changes in the above described external forcing in combination with the known ice body of the rock glacier (Hausmann et al., 2012). It seems that the external forcing leads to a significant melting of the permafrost body, which "stimulates" the rock glaciers to increase the flow velocity. In this context it would be very interesting, if, in which way and in which magnitude the ice body changed since the studies of Hausmann et al. (2012) which did their geophysical field surveys in 2002 and 2007 and thus right before the strong increase in 3D displacements since 2006 and the strong increase in flow velocity since 2012.

Regarding the other rock glaciers, we have included a zoomed section of the diagram for better visibility (Fig. 10b). The graph clearly shows that there are differences between the individual rock glaciers in this area. RG 02, RG 03, RG 05 and RG 08 show a tendency towards increasing flow velocities and increasing negative 3D displacements, which is also reflected in a clearly visible increase in the scattering both in terms of 3D displacements and in terms of flow velocities. Here, neither the mean ratio nor the ratio in the scatter (shape of the circles and ellipses) between flow velocity and 3D displacement seems to change visibly. But there are differences between the rock glaciers in terms of the first epoch as well as in terms of temporal changes. While the trend regarding the relation between 3D displacements and velocity seems to be constant for all three rock glaciers, the temporal course of changes is quite different. While the temporal change for RG 02 and RG 05 is relatively constant and shows only minor activity in the first epoch, RG 05 shows a much greater increase in terms of velocity and 3D displacements compared to the other rock glaciers. Phases with significantly increased activity and phases with only slightly increased activity alternate for RG 03 and RG 08 with an already existing activity in the first epoch of RG 08 and minor activity of RG 03. This alternation, in the case of RG 03, is the consequence of the delayed response to road construction (c.f. Sect. 4.6.1). In the case of RG 08, it is a shifting of activity zones from the area of the tongue to the upper part of the rock glacier, with some time lag between these activity phases, with only a slight increase in 3D displacements and velocity in the third epoch.

RG 06 and RG 07 do not show a stable relationship between 3D displacements and velocity over the study period. RG 07 indicates a clear increase in velocity in the second epoch, which is also accompanied by a negative 3D displacement. After the second epoch, the rock glacier appears to accelerate, but this acceleration is not associated with an increased 3D displacement;





rather, they decrease or remain at about the same level. Here, it appears to have been an activation (or reactivation) of the rock glacier in the second epoch, causing the system to move and the rock glacier to advance overall, resulting in an overall decrease in volume loss after a period of higher volume loss in epoch 2. RG 06 shows an increase in velocity in epoch 2, which is not accompanied by higher 3D displacements. In the third epoch, the 3D displacementsincreased with only a slight increase in

velocity. The increase in 3D displacements hold on in the epoch between 2012 and 2017, but is accompanied by an increase in velocity. It is very likely that this temporal behaviour is the consequence of a thermokarst phenomenon including the appearance of a lake (c.f. Sect. 4.6.3). This lake indicates the melting of permafrost in the third epoch and the following activation of this part of the rock glacier regarding flow velocity, probably due to the heat transfer by the water and the building of subsurface flow routes.

In summary, the temporal decoupling of vertical and horizontal movements observed by Ulrich et al. (2021) over a seasonal period can also be observed over a longer period for some rock glaciers. This indicates that both processes have a different sensibility to external forcing parameters, are governed by different external forcing parameters or have a varying time lag.

## 5 Conclusions

The aim of this study was to investigate the multi-decadal and catchment wide morphodynamic changes of rock glaciers, based
on a spatial analysis and by using high resolution topographic information from aerial images and LiDAR acquisitions. These data were analysed in the context of the climate variability.

It could be demonstrated that the approach of the combination of different remote sensing techniques for the detection of vertical and horizontal 3D displacements is well suited to extend the study period back into the mid of the 20$^{th}$ century and thus identify trends in rock glacier dynamics and relate them to climate changes evident over such a long period, as the derived
changes were above the determined error values.

As a general result, we were able to demonstrate a significant increase in flow velocities in the epoch 1997 to 2006 and an increase in subsidence to varying degrees over the entire study period. Both observations can be explained by changes in external forcing. The sharp increase especially in spring and summer temperatures since the 1990s leads to a change in the flow properties of the permafrost body due to a warming of the permafrost ice. Although the thawing of permafrost ice cannot
be distinguished from compaction due to a loss of pore space, trends to negative mass balances suggest a progressive thaw of the permafrost body throughout the study period. Furthermore, the melting of the ice body might create new drainage systems. This results in more water being available to the system, which is crucial for horizontal movement in shear zones. Flow velocity in this catchment area can also be linked to changes in precipitation pattern, which again governs water availability and the onset and duration of snow cover, which controls the time span of liquid water availability as well as the temperature in the
subsurface due to the winter cooling intensity.

 Although we were able to identify a general trend in rock glacier morphodynamics, the catchment wide view also shows a slightly different response of individual rock glaciers to similar external forcing regarding timing, magnitude and local

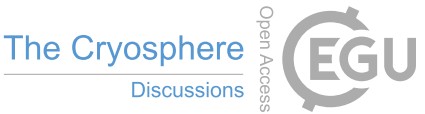

temporal peaks and the relationship between 3D displacements and flow velocities. No characteristic could be identified that explains the different responses to external forcing over the entire study period. Elevation is suspected to play a role evidenced
by some observations, although the investigation of altitudinal zones of the individual rock glaciers did not yield an altitude dependent temporal trend. The different behaviour could be explained by different sensitivity, response or response time of individual rock glaciers to intra-annual, inter-annual or multi-annual fluctuations and changes in external forcing parameters. For some rock glaciers internal structure and topography might explain different reactions, as two rock glaciers of similar size, exposition, elevation and elevation range showed contrasting reactions of inactivation and reactivation.

Beside the detection of two rock glaciers showing signs of full or partial inactivation, we were also able to show the influence of thermokarst on rock glaciers. Both phenomena will become more frequent during the 21st century as climate change progresses and permafrost degrades as a result.

We can also conclude that future investigations are necessary to better understand the climate forcing on rock glacier morphodynamics. Therefore, the analysis should be transferred to other catchments in order to identify differences and
similarities within the Alps. 3D displacements and flow velocity should be combined with downscaled reanalysis data to better understand catchment wide differences in external forcing on a longer timescale. If possible, future studies should combine borehole measurements or geophysical investigation to shed light on the internal structure of rock glaciers and clarify some of the assumptions and possible explanations of their behavior given in this study.

As a last important perspective, historical terrestrial images (if available) should be used with monoplotting tools. Mapping on
such images would help to shorten the time span of the individual epochs which is crucial to better differentiate the influence of individual forcing parameters, as it is very likely that there are changes within our analysed epochs. Beside this, historical terrestrial images would offer the opportunity to expand the analysis back to the 19th century and thus closer to the LIA in order to study an important period in terms of massive system changes in the glacial and periglacial regions of the Alps.





# Appendix

**Table A 1: Characteristics of the rock glaciers studied. Permafrost occurrence gives the pseudo-probability of permafrost (Otto et al. 2020). Area covered by 1850 glacier extent is ascertained according to LIA glacier extends (Fischer et al. ,2015) if not specified otherwise.\* As described in (Berger et al. 2004); \*\* as described in (Dusik et al. 2015).**

| Rock glacier | RG 01 | RG 02 | RG 03 | RG 04 | RG 05 | RG 06 | RG 07 | RG 08 | RG 09 |
|---|---|---|---|---|---|---|---|---|---|
| **Exposition** | W | NE | N | NE | W | E | E | N | NE |
| **Area** [km²] | 0.237 | 0.058 | 0.036 | 0.036 | 0.059 | 0.182 | 0.02 | 0.088 | 0.171 |
| **Slope [°]** | 25 | 26 | 26 | 20 | 31 | 24 | 30 | 30 | 22 |
| **Elevation** [m] min | 2446 | 2 615 | 2596 | 2 727 | 2702 | 2 695 | 2709 | 2 510 | 2627 |
| max | 2780 | 2 755 | 2787 | 2 812 | 3093 | 2 948 | 2861 | 2 761 | 2925 |
| **Permafrost occurrence** [%] min | 0 | 0 | 16.97 | 0 | 0 | 0 | 0 | 0 | 0 |
| max | 65.41 | 57.95 | 45.68 | 70.53 | 76.04 | 81.42 | 50.52 | 60.52 | 78.57 |
| mean | 14.49 | 33.22 | 48.99 | 47.08 | 48.24 | 44.04 | 20.54 | 38.33 | 37.57 |
| **Connection to the upslope unit** | TC; GFC | TC | GC; GFC | GC; GFC | GC; GFC | TC | TC | TC | TC; GFC |
| **Area covered by 1850 glacier extent** | Yes\* | No | Yes | Yes | No | No | No | No | Yes\*\* |

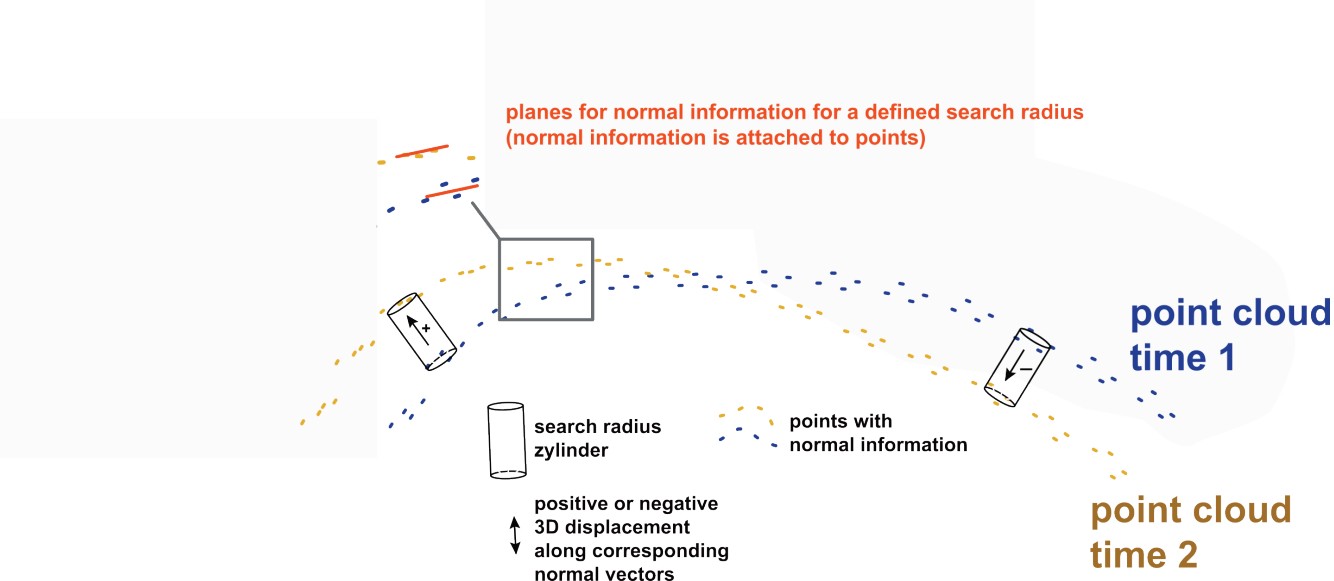

planes for normal information for a defined search radius
(normal information is attached to points)

point cloud time 1

search radius zylinder

points with normal information

positive or negative 3D displacement along corresponding normal vectors

point cloud time 2

**Figure A 1: Illustration of the calculation of 3D displacements based on Fey & Wichmann (2017) and Lague et al. (2013)**





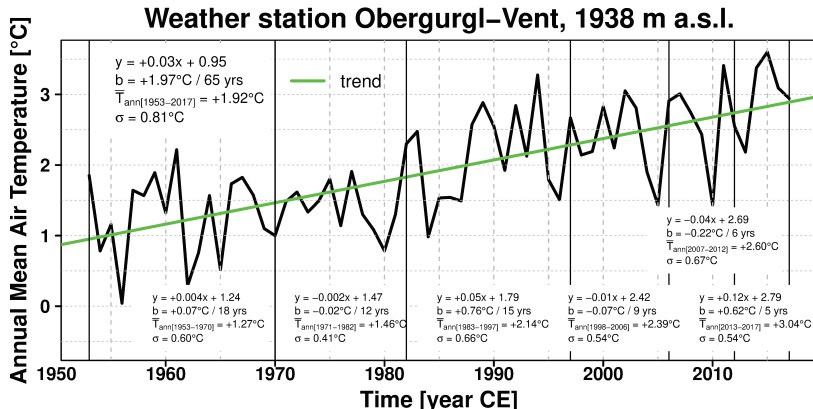

Figure A 2: Mean annual air temperature and its trend from the station Obergurgl-Vent (1938 m a.s.l.) for the period 1953 - 2017. Black lines indicate the epochs of flow velocity measurements.

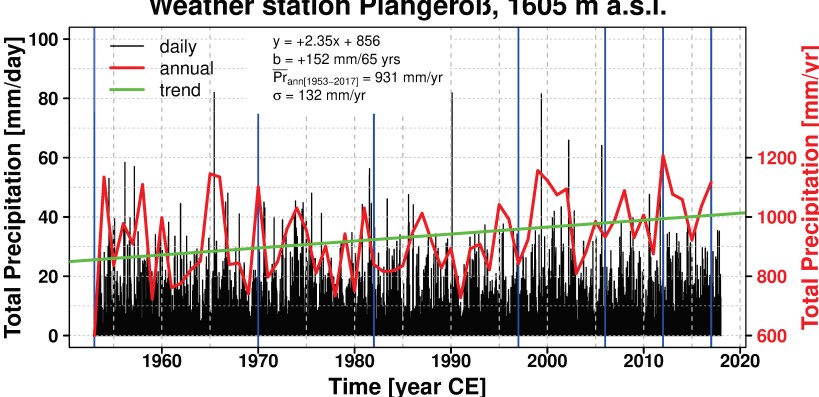

Figure A 3: Daily and annual precipitation sums with its trend of the station Plangeroß (1605 m a.s.l.) for the period 1953 - 2017. Blue lines indicate the epochs of flow velocity measurements

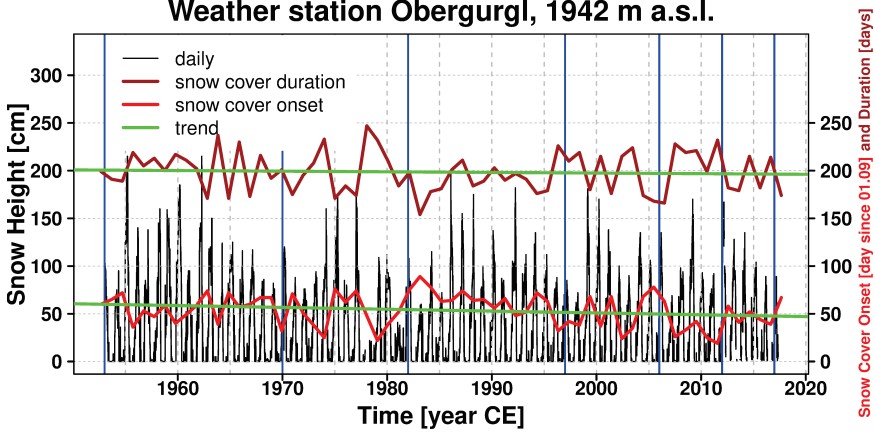

Figure A 4: Snow height, snow cover onset and duration with its trend derived from snow height data of the station Obergurgl (1942 m a.s.l.) for the period 1953 - 2017. Blue lines indicate the epochs of flow velocity measurements
**Code availibility**

The image correlation algorithm (IMCORR) is used for the calculation of the rock glacier flow velocities is implemented in the open source geoinformation system SAGA GIS. Furthermore, some modules of the commercial SAGA GIS extension SAGA LIS PRO 3D were used to calculate the 3D displacements of the rock glaciers. The software which was used to create digiatal elevation models and orthophotos from historical aerial images was the commercial software Agisoft Metashape.

**Data availibility**

The analysed metrological data is availible from the "Federal Misistry of Agriculture, Regions and Tourism" (BMLRT), the "Central Institute for Meteorology and Geodynamics" (ZAMG), the "Historical Instrumental Climatological Surface Time Series of the Greater Alpine Region" (HISTALP), the "Autonomous Province of Bozen/Bolzano" and "Tyrolean Hydropower AG " (TIWAG). The aerial images used to create digital elevation models and orthopotos are availible from the "Office of the Tyrolean Government-Department of Geoinformation" (https://www.tirol.gv.at/en/) and the "Austrian Federal Office of Surveying and Metrology" (BEV) (https: //www.bev.gv.at). The self-collected ALS data will presumably be made available after completion of the SEHAG ("Sensitivity of High Alpine Geosystems to Climate ChangeSince 1850") research project.

**Author Contributions:**

Conceptualization, FF, FH and MB; data curation, FF, LP, MA, JR, MP, MS, FH and MW; formal analysis, FF, FH, MP; funding acquisition, MB, FH, TH and NP; investigation, FF, FH; methodology, FF, FH, LP, MP, TH, MA, JR, MS, MW, NP and MB; project administration, MB, FH, TB and NP; supervision, FH, TH and MB; writing—original draft, FF, FH; writing—review and editing, FF, FH, LP, MP, MA, JR, MW, NP. All authors have read and agreed to the published version of the manuscript.

**Conflicts of Interest:**

The authors declare that they have no conflict of interest. The funders had no role in the design of thestudy; in the collection, analyses, or interpretation of data; in the writing of the manuscript, or in the decision topublish the results.

**Acknowledgments:**

The study was part of the SEHAG project ("Sensitivity of High Alpine Geosystems to Climate ChangeSince 1850") and financially supported by the German Research Foundation (DFG) and the Austrian ScienceFund (FWF). We gratefully acknowledge the DFG and the FWF for the financial support. We would also like to acknowledge the Tyrolean Hydropower





AG (TIWAG, Austria, Innsbruck), the Federal Misistry of Agriculture, Regions and Tourisme (BMLRT, Austria Vienna), Central Institute for Meteorology and Geodynamics (ZAMG, Austria, Vienna) and the Autonomous Province of Bozen/Bolzano for providing the meteorological data. We would like to acknowledge the Office of the Tyrolean Government (Department of Geoinformation, Austria, Innsbruck) for providing the historical images 1970 and 1971 as well as the
corresponding camera-calibration certificates. We would also like to thank the Austrian Federal Office of Surveying and Metrology (BEV, Austria, Vienna) for providing the historical aerial images of 1953/54, 1982 and 1997. We acknowledge the Kaunertaler Gletscherbahnen GmbH for the free use of the toll road in the UpperKauner Valley. The open access publication of this article was supported by the Open Access Fund of the Catholic University of Eichstätt-Ingolstadt.

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
