# Peer review of "Multi-decadal (1953 – 2017) rock glacier kinematics analysed by highresolution topographic data in the Upper Kauner Valley, Austria"

_The Cryosphere, 2021_

## Author Comment (AC1)

**General comments**

*We thank referee 1 for the detailed and constructive suggestions for improving our manuscript. We agree with the comments and suggestions and will address the individual suggestions in more detail below. Please find our answers in blue in the text.*

This is a very interesting study investigating the long-term evolution of a relatively large sample of individual rock glaciers. Despite the increasing number of studies on rock glacier dynamics and evolution, there is still a lack of knowledge on the past velocities of rock glacier. This study aims at filling this gap and gives very interesting results. The analyses are thorough, very detailed and original. The errors are systematically considered and their analyses are carried out in depth.

I have however a major concern regarding the length and the structure of the manuscript. First, the text is very long and it should be reduced by about 20%. Second, and most important, the manuscript is not well structured. The results and discussion are merged into a single very long chapter, which does not allow the reader to have a clear view on the most important results of the study. The results must definitely be separated from the discussion, which is the classical way for a research paper. The references to the literature must be systematically moved to the Discussion chapter, allowing the keep the Result section more strictly factual (typical examples P16L424-428). There is also a countless number of subtitles. As a consequence of all of this, we get progressively lost. In the end we lose the main information, which is a pity because the quality of the analyses is very good and the results very interesting. Therefore, a strong effort must be made to improve the structure of the manuscript and to make it clearer.

Examples of modifications to the structure to be made :

- Move chapter 4.1 after 4.2
- Chap 4.4 (and 4.3.3., which should be merged with 4.4) should be moved in the Discussion and condensed.

The difficulty for such a study relies on its intrinsic interest : whereas similar studies generally consider one or two landforms, here a large amount of data is available for 9 rock glaciers. Thus, the authors must find a way between presenting sufficient data without losing the reader in two many details. A way to do it would be to focus more on the general trends and to reduce a bit the analyses of the exceptions and of the special cases.

*We agree with the comments on the length and structure of the manuscript. As we have generated a lot of data on the different rock glaciers, it was very difficult to highlight the really important results and discussion points. However, your comments have led to more clarity about what the important statements and results are. We will more clearly provide the important points and elaborate them better. In addition to the proposed changes to the introduction, we will separate, restructure and condense the results and discussion sections for better structure and clarity. In the results section, we will present the strictly factual results of the rock glacier inventory, the changes in the meterological forcing, the flow velocity analysis and the surface elevation change analysis, particularly the last two points in a more general way. In the discussion section, we will discuss the flow velocities and volumetric changes in the regional context and work out the similarities and differences in the reaction. As a further point, we will relate the flow velocities and volumetric changes to the*

*changes in the forcing parameters (temperature, snow cover and precipitation) and discuss them more systematically on the basis of the existing literature.*

The introduction is a bit lengthy and not well structured. Up to line 55 it's a long summary on the general characteristics of a rock glacier. Not everything is useful, thus I suggest to shorten this part and to keep only what is necessary. Another issue is that we must wait the end of the introduction to know the goal of the study. Ok, the precise objective must be presented after the state of the art, but the general objective, or at least the topic of the paper must be stated much earlier. Thus, I recommend to reorganize the introduction and to better structure it (see specific comments).

*We will significantly revise and shorten the passage on general rock glacier characteristics. In addition, we will state the aim of the study in a sentence at the beginning of the introduction.*

The state of the art is generally good, but additional references on the current state/velocities of rock glaciers, including destabilizing ones, could be added. For example Kummert et al. 2018, Vivero & Lambiel 2019, Marcer et al. 2021,…

*Although we have attempted to present information on the current status, destabilisation and velocities of rock glaciers on different temporal scales and their influencing factors in the introduction (P2&3, L55-72), we understand the suggestion to to support these statements with more recent literature. We will incorporate the recommended literature at the appropriate passages.*

In such a study it would really help to have a Google Earth link to visualize the rock glaciers, or/and pictures of each rock glacier.

*We agree. In complement to figure 1 we will include a .kmz file with the locations of the rock glaciers in the suppementary part of the paper.*

The results of the rock glacier inventory are presented in the Study area section, whereas the method for achieving it is presented after, in the Material and Methods section. This is not coherent. Since this rock glacier inventory is part of this study, the results must be moved in the corresponding section and removed from the Study area section.

*We agree. We will only mention the previous studies on rock glaciers in the Kaunertal in the study area section and present the results of the inventory in the results section.*

The calculated 3D displacements are changes normal to the surface. As explained by the authors, they are an alternative to the traditional DoD, and even a better quantification of the thinning/thickening processes on an ice-sursaturated permafrost body (see Vivero & Lambiel 2019 for a similar study). But this is not 3D displacement. The latter is rather a displacement that considers the 3 components x, y and z. As such this defines the displacement parallel to the slope angle, and thus the real displacement, contrary to the horizontal 2D displacement. The titles and text related to this must then be reformulated.

We agree that 3D displacement is a misleading term. As Referee 2 has pointed out, our methodology for calculating volumes by gridded '3D displacements' is invalid as this leads to small but systematic errors in volumes. Therefore, for the calculation of the volumes we have to apply the classical 2.5 D method via DoDs and will determine the uncertainties according

to Anderson (2019). Since a test has shown that there are hardly any differences in the representation of surface changes between the previous approach and the DoD method, we will also present these as DoDs in order to avoid confusion. Although there are now some studies that directly compare point clouds, as mentioned in your comment, we have come to the conclusion that using DoDs throughout the paper increases comparability to other studies and enhances comprehensibility.

If I understand well the chart on Snow cover onset, snow arrived roughly early September around the years 2010. This means that what you consider as the snow cover onset in fact corresponds to the first snow, meaning that snow can then melt completely until new snow falls. Hence, this parameter cannot have any influence on the rock glacier kinematics. Much more important is the date when a substantial snow cover is established (~50 cm), allowing ground insolation. In addition, I suggest to add as a parameter the date of complete snow melt in spring. This has a strong influence on the MAGST and thus on rock glacier kinematics. See PERMOS 2019. Permafrost in Switzerland 2014/2015 to 2017/2018. Noetzli, J., Pellet, C. and Staub, B. (eds.), Glaciological Report Permafrost No. 16–19 of the Cryospheric Commission of the Swiss Academy of Sciences, 104 pp.

*For the analysis of snow cover onset and duration, we followed the criteria described in Peng et al., 2013. They describe the snow cover onset as the first day of the first five consecutive days with snow in fall (considered September to January) and the end of the snow cover as the last day of the last 5 consecutive days with snow in the melt season (February to July). The duration is calculated by counting the number of days between the snow onset and snow end.*

*We agree that this is probably not the decisive factor for the morphodynamic development of rock glaciers with regard to snow. We have made a further analysis, for the onset of a substantial snow cover (50cm) and the date of complete snow melt and will integrated this into the study.*

The interpretations of the velocities and surface changes regarding the external parameters are sometimes rather hypothetical and should more systematically rely on existing literature. This would be much easily achieved by moving these interpretations in the Discussion chapter.

*We agree with this. As mentioned before, we will separate results and discussion, and in the discussion of morphometric changes in relation to external paremeters, we will refer more systematically to existing literature.*

**Specific comments**

P1L12. Two times "change" in the same sentence.

*We will replace change with shift.*

P1L20. In the rest of the manuscript you don't talk about vertical 3D, but only 3D. Be consistent. But take also in consideration my comment above about 3D.

For a detailed answer, see comment above. Since we will change the methodology completely to a DoD analysis, we will call it surface elevation change.

P2L31. **are** responsible

*We will correct this.*

P2L32. **generally** coarse debris layer (the coarseness depends on the lithology).

*We will add generally.*

P2L33. landforms

*We will correct this.*

P2L38. Remove "also". If the origin is periglacial, then the ice forms by freezing of water.

*We will remove also.*

P3L77-80. Here you present the results of a specific study on velocity variations for selected rock glaciers. But it must be moved around L60, where you talk about rock glacier velocities. In addition, it appears weird to give details for a specific region only for one study. Thus, either you stay more general, or you keep these details but, in the meantime, you must give similar details for the other referenced studies.

*We have decided to describe the details of the study by Groh & Blöthe (2019) in more detail, as they cover the same research area but with a different temporal and thematic focus.*

*We will move the more detailed description of the study to chapter 2, study area, as it fits better here.*

P3L85. **of** rock glaciers

*We will correct this.*

Figure 1: add the location of the study area in Austria; add the location of the highest summit.

*We will add the national borders to the overview map for better orientation and included the location of the highest peak.*

P4L103. Why "pseudo" ? It sounds weird.

*The authors of the permafrost map (https://doi.org/10.1594/PANGAEA.917719) refer to it as a pseudo probability of permafrost being present but do not specify the term. In the corresponding publication (Schrott et al. 2012), pseudo probability does not appear either. However, as this is not relevant for our study and the permafrost map is only shown for illustration purposes, we will remove peudo.*

P4L106-108. Obviously the road was built for the ski activities. You could make it clearer and say a bit more on the anthropogenic influence.

*We will further elaborate very briefely on the anthropogenic influences, as except for the glacier road there is no influence on rock glaciers and their kinematics.*

P6L127. To avoid repetition replace the second "Berger et al. (2004) by "The latter".

*We will replace this.*

P6L150-154. This refers to the state of knowledge on factors controlling rock glacier kinematics. Therefore, it should be moved into the introduction.

*As this information is already included in the introduction, we will delete it from the methods and material section.*

Table 1: Mi**n**istry

*We will correct this.*

P7L170. This is an open reproach towards the company that can be critical. I suggest to moderate your sentence.

*This was not intended as a reproach to the company, but we understand that it can be understood as such and will chang the sentence.*

Table 2: Uniformize the font

*We will correct this.*

P8L200-201. How many GCPs did you use ?

*We used 101 GCPs, which we picked very carefully directly from the point cloud in stable areas and as evenly as possible over the entire study area.*

*We will specify the numer of GCPs.*

P10L240. ha**d**. In general, check the tenses. Sometimes the present is used, sometimes the past (L245: better were than are).

*We will correct this and and check tenses used throughout the text.*

P11L274. Figure number ?

*It should read figure 4. We will add the figure number.*

P12, chap. 3.6. See my general comment on the 3D displacements.

*See your answer in the general comments.*

P12L300. **a LoD**

*We will correct this.*

P13L320-322. Syntax problem with this sentence.

*We see the problem and will rephrase the sentence.*

P15L366-367. The end of the sentence is strange.

*We wanted to express the reduced positive temperature trend of winter and autumn temperatures compared to summer and spring temperatures. We will rephras the sentence to express this more clearly.*

P15L367. El**e**vation.

*We will correct this.*

P15L375. You could complete with additional references.

*In an earlier version of the manuscript, we had included additional references at this point, such as Gruber et al. (2004). Unfortunately, The Cryosphere limits the number of references for research papers to 80, so we had to remove some references.*

P15L389-390. I don't understand this sentence. You mean that P increased from 931 mm/yr to 957 mm/yr at Weißsee ? Please reformulate. And in the following lines it is not clear of which station you are talking about. And why not showing the data for Weißsee station ?

*We wanted to express that in the period under investigation (1953-2017) the mean annual precipitation of the station Plangeroß was 931 mm/yr and in the case of the station Weißsee in the period since the recording (2007-2017) the mean annual precipitation was 957 mm/yr. In the following passage we describe the data of the station Plangeroß. We will completely rewrite this passage.*

*We have not plotted the station data, as we have only used them to check whether the stations with longer time series, which are not located directly in the area, measure valid data. We can include a graph in the appendix with the measured data (temperature, precipitation and snow depth) of the Weißsee station.*

P16L408. Honestly the tiny decrease in the snow duration cannot be considered as a trend. it only takes one year with a positive anomaly for the trend to reverse. And how do you calculate the snow cover onset ? From which snow depth do you consider that the snow cover is permanent ?

*We agree that this cannot be seen as a trend. For the analysis of snow cover onset and duration, we followed the criteria described in Peng et al., 2013. They describe the snow cover onset as the first day of the first five consecutive days with snow in fall (considered September to January) and the end of the snow cover as the last day of the last 5 consecutive days with snow in the melt season (February to July). The duration is calculated by counting the number of days between the snow onset and snow end.*

*We agree that this is probably not the decisive factor for the morphodynamic development of rock glaciers with regard to snow. We will make a further analysis, as suggested in the general comments, for the onset of a substantial snow cover (>50cm) and the date of complete snow melt and integrat this into the study.*

P16L422. How much were the velocities for this period ? According to Fig. 8 they should not have been much higher than 0,5 m/yr. Such displacements should not have provoked decorrelation.

*In the period 1953 - 1971, the maximum flow velocity for RG 8 occurred in the area of the front and could only be determined by manual mapping of a few individual blocks. The maximum flow rate was 1.66 m/yr, which corresponds to a total movement of 29.85 m during this period. Therefore, the rate of movement could no longer be determined by image correlation. One should not confuse the maximum flow velocity with the average flow velocity. The maximum flow velocity for RG 08 is still over 1 m/yr in the following epoch (this is somewhat difficult to see in Figure 4, as the maximum flow velocities in the boxplot are outliers and are therefore only shown as dots). Based on the topography, movement pattern, and elevation of the rock glacier front, we suspect the "end" of a rock glacier destabilisation, as is often observed today, as a reaction to the positive temperature anomalies in the 1940s. However, this is only a speculation, as no data are available before 1953.*

*We will rephrase this passage to make this clearer.*

P17L433. You could also reference to the PERMOS reports.

*As we have shown before, the references at The Cryosphere are limited to 80. Since we have already used the six references with which we substantiate the statement, we have opted for these. Nevertheless, we find the PERMOS reports exciting and will include them if possible.*

Figure 4: What do the red dots and bars indicate ?

*The red dots indicate the mean value, the red bars indicate the insignificant measurements.*

*We will add this description to the legend of figure 4.*

P18L454. Space before "Roer"

*We will correct this.*

P18L458. Could it be differently ? At the scale of the study area the changes in external forcing are the same for all the rock glaciers.

*We agree and will reformulated the sentence.*

P19L482. Fig **5**. To compare the size of the different rock glaciers the scale should be the same, and obviously it is not (in any case it is too small to verify it).

*No, the scale of the rock glacier maps is not the same, but it is indicated in the maps, admittedly a little small. We will separate the graphs and maps to create 2 figures for better readability and we will adjust the scale.*

P19L483-484. … which is so normal ! I don't know any rock glacier showing uniform velocities on its entire surface.

*We are aware of that. We just wanted to describe the patterns we see in the flow velocity maps. We will modify the sentence.*

P19L491-492. This is highly speculative. With such a low sample it is not possible to conclude anything about the link between rock glacier acceleration and altitude. And there is no objective explanation why higher rock glaciers would react more than lower ones.

*The reasoning behind this was hat the higher-elevated rock glacier RG05 only reacted so strongly in the last epoch, as a delayed reaction to the temperature increase due to the higher elevation. However, we understand the concern that the sample is far too small for such a statement and will delete this sentence.*

P20-21, Figure 5. Figure a bit complicated. Everything is too small and thus difficult to read. I suggest to make 2 figures with 1) the charts and 2) the maps.

*We will do that. If this separation takes up too much space, we will move the charts to the appendix. In addition, we will mark the areas with measurements below the error value and place a desity plot or violin plot of the velocities next to the maps in order to have a better representation of the development of the different kinematic areas as requested by referee 3.*

P22L523. What do you mean by "system state" ?

*By system state we mean the change of the rock glacier or permafrost body to increased temperatures, for example the formation of drainage systems.*

P22L225. "summer" instead of "autumn".

*We will correct this.*

P22L530. But the velocities are not only controlled by air temperature but also, and in a large portion, by the historic development of the snow cover, including the date of complete snow melt.

*We are aware of this, and will address it in the reformulated chapter of the discussion that deals with the influence of external factors on rock glacier velocities. For this we will do the additional analysis of the complete snowmelt.*

Figure 6: Indicate the period of comparison regarding the anomalies in T and P. I guess 1961-1990 ?

*You are correct, the reference period for anomalies is 1961-1990. We will indicate this in the caption of figure 6 and in the method section.*

P25L570. But generally a long duration of the snow cover is related to a thick snow cover, and thus leads to increasing liquid water, considering also that the latter is available all along the snow melt period.

*We agree. Since we will change the analysis of the snow cover, as mentioned before, all related paragraphs will change as well and we will take this comment into account when we revise the corresponding paragraph.*

P25L582. Looking at Fig 7 the value for RG 05 seems to be lower than 0.031

*Since figure 7 is a box plot, the mean value is not shown, but the mean value is actually 0.031 m/yr. The boxplots can be misleading here as outliers are not plotted, we will clarify this in the caption.*

P25L591. What is this other rock glacier pushed forward ?

*This is to describe the changes observed in the "3d displacement" maps and associated boxplots. These are made up of positive and negative changes. We wanted to use the term "pushing forward" to describe the frequently observed advance of rock glacier front due to flow. We will reformulate this paragraph to make it clearer.*

P26L598-606. Please refer to the corresponding Figure. This is an example of too long paragraph regarding the data that have to be presented. The same could be said in 3 lines. Not necessary to give all these details for RG 02.

**We will take this comment into account when we revise the results and discussion sections of the paper. We have already given the structure of the revision of the results and discussion section in the general comments section.**

P26L615. …different sizes. We already know this.

*We agree and will deleted the sentence.*

P27L618-621. Despite the fact that the maps are tiny (please increase the size, for instance by making 2 figures), I rather see patterns of positive or negative changes instead of scattering. Or you mean scattering at a larger scale ? But anyway the figures are too small to be analyzed by the reader.

*We agree. We have tried to integrate all the information about the flow velocity and "3d displacements" into one figure each, so that everything can be seen and compared at a glance, but we see that this has been done at the expense of recognisability and will separate the figures. You are right it should be patterns of positive and negative changes instead of scattering. We will change this accordingly in the text.*

P27L621-622. I don't understand the sentence. And avoid references in the middle of a sentence.

*We wanted to express that the evolution of the characteristic flow bulges on the rock glaciers can be seen here. We will formulate the passage in the text in a more comprehensible way.*

P29, chap. 4.6.1. I don't see any particular evolution for this rock glacier, since most of the landforms studied show an increase in velocities from 1997. This section is highly speculative and I suggest to delete it.

*Since on this rock glacier, unlike most others, not only the flow velocities increase, but also the pattern of lower and higher flow velocities changes, we have linked this to the construction of the glacier road. Since the construction of the glacier road, the high flow velocities occur below this road. However, you are right a direct connection is rather speculative. Since we are deleting the special cases chapter, we will only briefly refer to the anrtopogenic influence of this rock glacier due the the construction of the glacier road.*

P29L671. RG04 is obviously a push moraine (i.e. frozen sediments – probably a rock glacier – deformed by the LIA glacier advance). This is highlighted by the back-creeping movement towards the former glacier position and the strong subsidence, indicating high ice content. This must be considered in the analysis.

*We will take this into account in the analysis, although we have decided to delete the chapter on special cases and include it very briefly in the discussion.*

P29L681-687. Ok for the possible reactivation, but it would be interesting to propose some hypothesis to explain such a reactivation process.

*Perhaps we should reconsider the term reactivation. Here we have followed Michelleti et al. (2015), who described similar behaviour for a similarly small rock glacier in the Hérens Valley (Switzerland). As described in the text, the flow bulges indicate that the rock glacier must have already moved before 1997, but the flow velocities in the epochs 1953-1997 are barely above the error value that can be achieved with our method. Therefore, the rock glacier is transitioning from a transitional state with presumably very low flow rates to an active state with measurable, higher flow rates.*

*In the text we give a very general explanation: "The sudden and sharp change might indicate a change in the internal structure of the rock glacier system and mechanism of flow" (P30L686-68). Hereby we wanted to express the assumption in very general terms that changes in external factors could lead to the formation of new drainage systems in the rock glacier and/or a formation of new or reactivation of older shear zones, which could strongly influence the kinematic behaviour of the rock glacier.*

P30L702. Permafrost is a thermal phenomenon. It can thus not melt.

*We agree. We have decided to delete the chapter on special cases, but will take the comment into account in the revised discussion.*

P30L708. "… in the area of shear surfaces…" : what do you mean exactly ?

*Here we wanted to express that the horizontal movement of the rock glacier (either in the area of the shear zone or by internal plastic deformation) can result in positive and negative values in the surface elevation changes, but a net volume change can only result from material input, output, compaction or melting of the ice body. We see that the formulation is not well comprehensible and will express this differently in the revised version.*

P30L709. "change" without s

*We will correct this.*

P31L720. "similar magnitude". Do you mean similar values ? Because it is evident that horizontal velocities are expected to be much higher than "3D" changes.

*We see that this is an incorrect formulation. In this sentence we wanted to express that in the previous analyses a good spatial and temporal view of the individual rock glaciers was possible, but the comparison with each other and the relationship between flow velocity and surface elevation change is facilitated by figure 10.*

---

## Author Comment (AC2)

General comments

*We thank you for seeing the potential of our study and for helping us to improve our mauscript with your objective and constructive comments. We see the need to revise the analyses as well as to shorten and restructure the manuscript and will do so according to your suggestion. We have already addressed some of the points in a similar way in the responses to reviewer 1's comments, but will address your comments again in detail below.* Please find our answers in blue in the text.

I highly agree with the opinion of the anonymous referee #1. Because of the high potential of the present paper I would like to add some personal comments helping the authors to improve their paper.

Permafrost studies are currently a hot topic in view of climate change. The authors focus on mountain permafrost, i.e., they want to understand the spatio-temporal change of rock glacier kinematics not only locally (single rock glacier) but on a more regional scale (several rock glacier, e.g., located in a valley or catchment area). The authors want to find out how nearby rock glaciers react (geometrically) to changing environmental conditions, i.e., MAAT, precipitation, snow cover, etc.

Change detection analysis is based on archival aerial photographs and ALS data. The proper processing of these data is not easy and requires a lot of knowledge and experience. I am confident that the data has been processed accordingly.

My mayor concern is on data analysis which has already been addressed by the anonymous referee #1. I'm referring to page 12 where the concept of 3D displacements on rock glaciers is outlined. The authors should clarify the term 3D displacement. To my understanding 3D displacement is a 3D vector describing the dislocation/movement of a point or distinct feature of an object/surface in space (and time). However, the authors of the paper interpret 3D displacement as a distance into a normal direction following the idea of Lague et. al. (2013). Commonly, this algorithm is called M3C2. This algorithm has same advantages, especially in interpreting surface change and its significance. The authors' quantitative analysis of the rock glacier kinematics is based on 2D/horizontal displacements and on volume change. The latter, however, has not been carried out in a fully correct way. Since volumetric change, as implemented in the paper, is based on gridded '3D displacements' (cp. P12L303-304) the obtained volumetric changes are inherently wrong. The authors would have derived a correct result if they had taken (0,0,1) = vertical axis as a reference direction. Due to the specific kinematics (e.g., extending creep internal mass transport) and the geometry (e.g., steep frontal slope) of rock glaciers the obtained volumetric changes are preferably/systematically negative (see Figue 8). I advise the authors to re-evaluate volumetric change. The authors may use M3C2 (properly modified) or a simple difference of digital elevation models (DoD).

*First of all, we agree with you that the term 3D displacement is misleading. After reading your comment we understand that the calculation of the volumes by gridding the point cloud based suface elvation changes is subject to a small but systematic error. This has a slight effect on the absolute volumes, but hardly on the relative change of the volumes over time. Therefore, the conclusions derived from the calculations will not change. Nevertheless, we see the need to improve the calculation of the volumes! We will therefore determine the volume via a classic DoD analysis and determine the uncertainty according to Anderson (2019) -*

https://doi.org/10.1002/esp.4551 *. In order to avoid confusion and to facilitate comparability with other studies that determine the surface changes of rock glaciers by means of DoD analysis, we will also change the other analysis of surface elevation changes on the basis of DoDs.*

In any case, the authors should include profound error analysis, i.e., significance analysis, for their velocity data (2D, horizontal) and their volumetric change results (1D, vertical, integral value obtained for an area).

*We have systematically analysed the errors of flow velocity analysis and surface elevation change analysis on stable areas. The approach to determaine the significance of flow velocities of rock glaciers is published in a peer-reviewed journal (Fey&Krainer 2020). We include this value in all figures and will show the areas below the LoD values in the respective maps in the revised version. In the case of uncertainty analysis of volumes, we will follow Anderson (2019).*

The paper will benefit from a more formal structure, such as

*We agree that restructuring will bring more clarity and better readability.*

Introduction (please clearly specify the research questions),

*We will rewrite the introduction as suggested by Reviewer 1 and clarify the research questions. The rewrite will condense the section on general rock glacier characteristics and add a more detailed description of the current status/velocities of rock glaciers.*

Study area,

Material (First: aerial photographs and ALS data; Question: What is the reason for not using recent aerial photographs? There is lots of data available at BEV! A comparative analysis would have boosted all results obtained. Second: Supplementary material, such as meteorological data, etc.

*We agree that a separation of materials and methods would provide a clearer structure and overview, especially since many different data were used. We will revise the materials and methods section of our manuscript according to your suggestions.*

*We know the availible BEV data as we have obtained many of the historical aerial photographs here. We decided against using the available more recent BEV aerial photographs because we have collected and/or postprocessed ALS data ourselves that cover more recent study period (2006-2017) and provide the desired information we need to answer our research question.*

Methods (photogrammetric mapping, georeferencing, SfM; processing/georeferencing of ALS data; 2D-displacement measurements (orthophoto-orthophoto, orthophoto-hillshade, hillshade-hillshade; software used; precision/accuracy assessment). Question: What is the reason to use hillshades instead of original elevation data?; computation of volumetric change (method, precision/accuracy assessment); Supplementary material (explain data aggregation, etc.)

*Since the elevation model of the 1997 dataset has a relatively poor resolution, a calculation of the flow velocity by image correlation with the 2006 ALS dataset in the elevation data was not successful. We state this in the manuscript (P10L241-242). In order to avoid using an additional data type for the determination of the flow velocities, we have also used hillshades for the epochs 2006-2012 and 2012-2017. The calculation of the flow velocity with the help of hillshades is a common procedure that is used in many studies (e.g. Dusik et al. 2015; Bollmann et al. 2015)*

*Nevertheless, we will restructure the methods section according to your suggestions.*

Results (present the results obtained. Maybe, you can find a good way to also aggregate the results)

We will present the strictly factual results in the revised version.

Discussion (discuss the kinematics (movement, volumetric change) of the rock glaciers in a regional context. Is there a correlation in space and time? Interrelate the kinematic information with the supplementary data.)

We will adopt the suggestion for the structure of the disscusion.

Summary (optionally)

Specific comments

The title should reflect the content of the paper. Do you really want to address morphodynamics? Did you mean kinematics? Morphodynamics would imply process understanding.

*We will change the title accordingly.*

The paper is too long and could benefit from shortening. Maybe, it is not important to discuss each rock glacier in detail. Is there a common response? If not, why?

*The restructuring and revision of the introduction, the methods and materials section and the results and disscusion sections will result in the publication being significantly shortened, also because we will keep the results and discussion section more general where appropriate and possible and, for example, delete the special cases section.*

Some of the figures are too small and too overloaded and thus they are not readable.

*We assume that you are referring mainly to figures 5 and 9. in these figures we will separate the diagrams and maps and make them into two figures. If this takes up too much space, we will move the diagrams of the altitudinal zones to the appendix.*

Figure 8: Mean annual volume change (m2) per 100m2 = mean surface height change (cm) !!!

*We agree. We will specify the mean surface height change in cm in the revised version.*

---

## Author Comment (AC3)

First of all, my apologies for this late review.

*Thank you for your objective, helpful and constructive comments, which will certainly help to improve our study. Please find our answers in blue in the text.*

The paper is one of the first publication comparing the evolution of rock glacier kinematics for a set of landforms located in a single catchment area over a period of more than 60 years. The analysis is mostly based on historical aerial photographs and more recent airborne laser scanning data made available by the authors team. It permits to capture the evolution of rock glacier kinematics at roughly at a decadal time step.

This is a very interesting paper suffering however from several weaknesses, which I strongly recommend to improve in order to consider it for publication. The paper is relatively long and needs to be significantly shorten, either via the text content or the concision of some sentences or paragraphs. Any repetition must be avoided. I agree that this is a difficult exercise. The content of the illustration is mostly excellent, but usually much too small, what is deserving the paper. Some very important results are lost in large figures (e.g. evolution of the velocity flow field) and must be highlighted. Maybe some additional figures are needed.

*This was noted in the same way by all other referees. We will restructure and shorten the study. Furthermore, we will omit the chapter on special cases and only briefly address it in the discussion and try to generalise the results.*

The structure of the paper must be revised. The description of all rock glaciers, including their spatial flow pattern and connection to upslope unit must come in entrance. It helps the eventual splitting of some rock glaciers in distinct sub-areas to be envisaged. Then the results are presented. Finally a distinct discussion section must come. At present results and discussion are mixed. The discussion must avoid to be too hypothetical.

*As noted by referee 1, we will present the results of the rock glacier inventory and the more detailed characterisation of the rock glaciers studied at the beginning of the results section. In the following, we will present the results of the flow velocity, surface elevation change (former 3D Distance) and the volume change analyses. Only then will we discuss the results.*

*Since this will substantially change the structure and length of the individual passages, it is also possible that some passages noted in the comments will be removed. Wherever we retain the passages, we will take the comments into account, as detailed below.*

The methodology to calculate the rock glacier flow rate (single value) is unclear. It looks to be a mean of all parts of the rock glacier where any data is available, whatever the kinematic behavior. What is the sense of doing so ? Marginal areas, not moving homogeneously with the main rock glacier body, should not be taken in consideration. In addition, for some rock glaciers, it looks that calculating a mean velocity for the entire landform has no sense regarding the heterogeneity of the kinematic behavior over both space and time. Separating some rock glaciers in two or several kinematic sub-aeras could provide results (and conclusions) differing from the current ones.

*After reading your comment and thinking about it, we understand your concern about calculating the mean value of the entire rock glacier, as is done in many studies, as this is not*

*representative due to the multimodal distribution of the velocities on some rock glaciers. Therefore, we would propose to keep Figure 4 (boxplot with mean and LoD) and Figure 6 (mean of flow velocities and influencing factors) for the comparison of the rock glaciers among each other. However, in a new figure, we will present the maps of flow velocity and the corresponding violin plots or density plots of flow velocity to illustrate the heterogeneity and the change of the different kinematic zones on the individual rock glaciers over time. In addition, we will make a figure showing the area of the rock glaciers whose flow velocity is above the LoD and above the maximum LoD of all time slices (proportion of "active" area). In this way, we believe that the heterogenity and change in the "active" area can be well represented. In this context, active means detectable as active with our method, which we will also describe this in the text.*

The "3D displacement" is not one, meaning it is not a displacement in xyz coordinates, but an inadequate terminology to define somehow a vertical movement only, but not exactly. What is the interest of applying such an approach (movement normal to the surface)?

*We agree that 3D displacement is a misleading term, which shoud discribe a point cloud based surface elevation change. Although this approach can be advantageous in complex trains (c.f. Lague et al. 2013), we will replace the analysis with a DoD analysis in the revised version, as Referee 2 noted that the calculation of the volume by gridding of the point cloud based surface elevation changes is not correct. Although this will just very slighly change the results, this approach may be easier to understand and provide better comparability with other studies that largely follow the classic 2.5 D (DoD) approach.*

Both abstract and conclusions must be revised accordingly. They have not been reviewed, because they may change after having adapted the analysis procedure.

Since about L500, I have not performed an in-depth review.

The additional references indicated in my review are suggestions only.

The location of all rock glaciers must be provided.

*In addition to figure 1, which shows the locations of the rock glaciers in the study area, we will add a kmz file with the locations of the rock glaciers studied in the supplementary merterial part of the study.*

Detailed comments :

Title : I guess it is more the multi-decadal kinematics of the rock glacier which is analyzed and not the morphodynamics

*We Agree. We will change the title accordingly.*

L14 : nine or eight ? Weird statement.

*In the case of the point cloud based surface elevation change there are nine and for the flow velocity analysis there are eight. The ninth could not be considered for the flow velocity*

*analysis due to shadowing and lack of surface structure which made image correlation impossible. This is specified in more detail at the end of the introduction and in the methods section.*

L33: "or pure ice". To be avoided. This would be a debris-covered glacier.

*We will remove "or pure ice" from the sentence.*

L38: "in part" can be omitted. Ice build-up within the ground might be the dominant process and the embedding of external (e.g. glacier, snowpatch) ice might be inexistent.

*We will delete this passage due to the requested restructuring of the introduction of Referee 1.*

L42: Active layer is consisting of unconsolidated debris (not only "boulders")

*We will correct this.*

L43: I don't see the causal relationship between the thermal regime driven by freeze-thaw cycles and the air-filled porosity of the active layer. There is also air and water advection. Are there some references to propose ?

*We will remove this sentence from the manuscript as reviewer 1 requested that we shorten the very general introduction to rock glaciers up to line 55 and go more into their (recent) kinematics.*

L43: "These", but which ones ?

*By these we mean the previously mentioned freeze-thaw cycles and the air convection. This sentence will also be deleted in the revised version, due to the requested shortening of the general introduction by Referee 1.*

L46: No, the debris size is not smaller, but the proportion of coarser debris per volume is less.

*We will correct this.*

L51: What does mean "long-term"?

*Since the previous sentence talks about seasonal changes, by long-term we mean the change in thermal forcing over several years. We will specify this.*

L51-54: See also Cicoira et al. 2021 - A general theory of rock glacier creep…

*We will add this reference.*

L.54 : The shear zone is maximally a few meters thick.

*We will specify this.*

L57ff: This is only valid in the European Alps.

*We will limit the validity of the statement to the European Alps.*

L59. Velocity decrease since the 1990s. Which of the mentioned studies are reporting this ? I agree that some rock glaciers are decelerating, but the general trend is a significant continuation of the acceleration (e.g. PERMOS 2019 in the Swiss Alps… must not be very different in the Austrian Alps, a couple of tens kilometers eastward)

*We did not want to talk about a general deceleration of rock glaciers following the acceleration in the 1990s, but to show that there were phases (years or multiple years), in which constant or decreasing velocities were measured. This is also partly reflected in our study, in RG 01, 03, 06, 08. In these examples, lower flow velocities were measured in the period 2006 - 20012 than between 1997 - 2006 and 2012 - 2017. At least in two of the metioned studies this is also the case (e.g. Kellerer-Pirklbauer and Kaufmann, 2012 - for three rock glacier in the Hohen Tauern Range, Austria or Kenner et al. 2020 - for Schafberg rock glacier in the Swiss Alps). We will formulate this more clearly and also note your comment about annual measurements and that this is mainly due to the deceleration drop in 2005/06.*

L70. No one of both mentioned references is showing this, but Delaloye et al. 2013 – Rapidly moving rock glaciers… - and Eriksen et al. 2018 - Recent Acceleration of a Rock Glacier Complex, Ádjet ...- are doing so. About destabilization, see also Marcer et al. 2019 - Evaluating the destabilization susceptibility of ...

*We agree that in the studies mentioned a maximum of 4m/yr is given. In an earlier version we had Vivero & Labiel 2019, which give 60-75 m in about a year, in the citation, this had to be deleted due to the limit of 80 references in the cryosphere.*

*We will add "...up to sveral tens of meters…" and include Vivero and Lambiel (2019) and Marcer et al. (2019) into the citation.*

L83 I would suggest "e.g." because there are other studies, sometimes difficult of being accessible. Maybe also Kummert et al., (under final review in ESPL) - Pluri-decadal evolution of rock glaciers surface velocity and its impact on sediment export rates towards high alpine torrents. See also Kääb et al. 2020 - Inventory, motion and acceleration of rock glaciers... for an example outside of the Alps

*We agree. We will insert e.g. in the citation.*

L85. There is something wrong in this sentence

*We will rephrase this sentence.*

L89. Are the rock glaciers the same, so 8 of 9 ?

*Yes, we will specify this. The reason was already given in a previous answer.*

L94. Never begin a chapter with a figure. But besides, it would good to precise what are the used coordinates, what is the unit (m ?) and to add (or replace them by) lat/long coordinates.

*The coordinate system used is ETRS89 / UTM zone 32N EPSG:25832 and the unit is meter. We will mention this in the caption.*

L97. m. a.s.l.

*We will correct this.*

L108. Anthropic influence on the rock glacier as well ? Which ones precisely ?

*Yes, there is also anthropogenic influence on a rock glacier due to the construction of the glacier road between 1979 and 1982, which intersects rock glacier RG03. We will mention this here.*

L110. Inactive rock glaciers. How was this classification done ? On which parameters ? Does it fit with the IPA Action Group Rock glacier inventories and kinematics definition ?

*Here we follow the classical classification into active - motion (from image correlation) inactive - ice contained; no measurable motion (melting, visible on DoDs) and fossil (e.g. Krainer & Ribis 2012). The methodology is described in chapter 3.1 Rock glacier inventory (P6L133-141). We will describe the classification in more detail in the methods chapter on the rock glacier inventory.*

L113. Replace by something like : Finally, eight active rock glaciers representing different characteristics and conditions were investigated in detail regarding flow velocities and one more regarding vertical displacements

*We agree that the formulation is clearer this way and will adopt the sentence in a similar way.*

L121. The rock glacier moving downwards, it does not make sense to write that it reaches the "highest elevation". Would it not be that it is located at the highest elevation range among the nine selected rock glaciers ?

*We will put the description in the results section, as Referee 1 asked us to put the results of the inventory in the results section. We will take this comment into account when we rewrite the manuscript.*

L129. Is the layer below the ice rich permafrost body really ice free ? Because it is very difficult to conceive an active rock glacier which is only frozen in its upper part. Where is the shearing zone developing on the long term ?

*Here we describe the results of the geophysical investigations of Hausmann et al. 2012 at the Ölgruben rock glacier (RG 01). Since we are not experts in the interpretation of geophysical surveys, we can only report the results of the study as published.*

151-153. There was a paper in 2008 (Delaloye et al. - Recent interannual variations of rock glacier creep in the European Alps) showing that there was an almost good similarity of interannual variations of rock glacier velocity over the entire European Alps, confirmed a decade later by Kellerer-Pirklbauer et al. (2018) at EUCOP - Interannual variability of rock glacier flow velocities in the European Alps. There was also a short communication at ICOP 2016 by Staub et al. - Rock glacier creep as a thermally‹ -driven phenomenon: A decade of inter-annual observations from the Swiss Alps - showing that the interannual variations are

basically driven by shifts in mean ground surface temperature for a period of about 2.5 years. For sure this is also influencing the liquid water content within the permafrost.

*This is why we write liquid water availability and not precipitation, the increased liquid water availability can have several causes, which according to Kenner et al. (2020) is mainly controlled by the ground temperature and the onset and duration of the snow cover. This is described in the introduction (P3L66-63). We will remove the explanation in the methods section, as Referee 1 noted that this should be explained in the introduction.*

153-155. The effects of liquid water availability and snow cover on rock glacier morphodynamics must be precised. Does it mean on-set and melt-out of the snow cover influencing the ground surface temperature or water equivalent of the snow pack which will melt out in spring/suemmer and directly influencing the rock glacier hydrology. Or both ?

*The influence of various drivers of rock glacier flow velocity is nicely displayed in Kenner et al. (2020), Figure 5. Although they just derive this from the study of one rock glacier, we think this is a very good overview and we are following it to some extent in our discussion of climate factors.*

*Both. As the influence of GST indiectly influences water availibility as well. The onset of winter snow cover determines the winter cooling intensity, which in turn influences the timespan with liquid water in the active layer. The strongest influence, however, is shown by the timing of snow melt, but less by the snow water equivalent (Kenner et al. 2020).*

Finally, the paper is focusing on decadal velocity changes. What relation to short-term (less than annual) changes ?

*We do not know exactly to which passage this comment refers, as we are now in the method section in cronological order. We refer to findings about short-term change and their relation to decadal velocity changes in the discussion.*

L276ff. This is not a 3D displacement, but something else (the surface change normal to the surface). But what is the interest of doing so and not calculating simply the vertical displacement? What are the advantages on a rock glacier ?

*See previous answer.*

L323 "showed good agreement". Please, provide values, figure or table.

*A "hard" quantitativ comparison of values seems difficult to perform, as methedology and time spans differ. Nevertheless, we will include a table in the appendix.*

L.326 Is the stable area so stable? In principle, bedrock is more suited to be stable than a debris slope.

*We agree that bedrock can generally be considered more stable than a debris solpe. However, on bedrock, which usually has lower roughness, the errors are underestimated, especially in the image correlation analysis. Since we know the change from our point cloud based surface elevation change analysis in these areas, we know that they are stable.*

L361-363. The introductive part of the sentence could be avoided.

*We will correct this.*

L.367. A significant trend cannot be calculated over 11 years only. What is this data meaning ? Is it a difference between the mean of the two periods or a trend in 11 years as expressed in the two previous sentences.

*The two previous sentences describe the trend over 65 years at the Obergurgl-Vent and Nauders stations. The one meant compares the increase in the mean value in the last two epochs of the study, including the Weißsee station. We will make this clearer when rewriting the results and discussion section.*

L370. Precise what are these seasons, e.g. spring is MAM, summer JJA ?

*We will precise this in the methods section where we describe the meteorological analysis and in the caption of the corresponding illustration.*

L371-372 Conditions causing heat waves in future is not the purpose of this paper looking back into the past. The sentences could be removed.

*Agreed. We will remove this sentence.*

L388. +152 mm in 65y. Is it a lot or not ? What is the annual value ?

*We give the mean annual precipitation value for the study period in the next sentence (931 mm/yr). In the previous sentence we give the range of trends for all the stations studied (53 mm - 241 mm). A positive trend of 152 mm in 65 years is already quite a lot in our view.*

L393-395. This sentence about precipitation predictions could be removed (not of interest for this paper)

*Agreed. We will remove this sentence.*

L408. Snow melt trend: How much ? What are the starting dates and durations ?

*Referee 1 has noted that the small decreases in the snow parameters are not a trend. We will therefore delete the sentence. In addition, we will reanalyse the snow parameters and add the onset of a significant snow cover (>50) and the complete snow melt as parameters.*

L420. Provided max flow values are valid for a single period or as a mean for 1953 to 2017 ?

*These values are for the whole period of inverstigation (1953 – 2017). We state this in the beginning of the sentence: "For the whole period of investigation…"*

L421. How is the mean value spatially calculated ? How is delimitated the area taken into consideration for the calculation ? It looks from the figures that they comprise marginal areas (with velocity close to 0) to sectors moving much faster. Why not to split into sub-areas and perform a comparative temporal analysis ? See also the definition of moving areas within the IPA Action Group Rock glacier inventories and kinematics -

https://www.unifr.ch/geo/geomorphology/en/research/ipa-action-group-rock-glacier/ - documents (kinematics as an optional attribute in rock glacier inventories)

*As described in the general comments, we will show the change in the "active" areas (areas above LoD and LoDmax). In addition, we will show the flow velocities of the individual time periods as a violin plot or density plot as these representations highlight the heterogeneity and its changes.*

L.421. A mean velocity of 3.5 cm/year is rising some questions about the accuracy and reliability of the results (in particular changes over time). Is such a low value significant ?

*As described in the general comments, the mean values play a subordinate role in the revised version, as the development of flow velocities over the entire rock glacier is better shown in the boxplot and future violin or density plots. The value mentioned represents the mean value for RG 03 in the time slice 1970-1982. In this time period, maximum movements of 1.51 m (0.089 m/yr) were measured and the significance value of the measurements is 0.027 m/yr with about half of the measurements exceeding this value. It can therefore be said that significant movement on the rock glacier can be measured in this time slice. In the revised version, as described, we will refer less to the mean values of the entire rock glaciers and take a better look at their uncertainty.*

L424. One should note that the period 1997-2006 is marked by the peak of 2000-01 (described for instance by Ikeda et al. 2003 - Rapidly moving small rockglacier at the lower limit of the mountain permafrost belt… - and 2008 - Fast deformation of perennially frozen debris in a warm rock glacier…) and the famous 2003-04 peak (e.g. Delaloye et al. 2008), The period 2012-2017 is embedding the extreme peak of 2015 (e.g. Kellerer-Pirklbauer et al. 2018, PERMOS 2019), whereas the period 2006-2012 contains no peak of activity.
L431. 2006 is often a low, the period with the lowerst velocity recorded since 2000 (e.g. PERMOS). More generally the sentence is difficult to understand unambiguously.
L431-433. You could refer to Delaloye et al. 2008 for the low in 2006 in the European Alps and to PERMOS 2019 for the description of the entire period.

*We have tried to include studies with higher temporal resolution in explaining the change in our epochs (e.g. P17L429-433 and P18L470-73). We will improve content and sentence structure with the help of the literature listed here.*

L434. RG 04 : A detailed spatial analysis (of the morphodynamics) is necessary, with the help of (time-lapsed) maps. It should be the same for the other rock glaciers.

*Referees 1 and 2 have requested that the detailed analyses of the individual rock glaciers be greatly shortened and that we rather focus on general similarities and differences in the revised version. But we will describe the general trends, simialrities and differernces of the spatial development. The development of the individual rock glaciers can then be examined by the reader in the flow velocity maps and newly added violin or density plots.*

L435-7. "Many studies mentioned periods of slight decrease or constant flow velocities following the strong acceleration in the 1990s". Not really. This is mostly related to the deceleration drop in 2005-06. Read the related papers (already mentioned earlier), and in particular the PERMOS reports.

*We will revise the passage with the help of the literature mentioned.*

L437-9. There are various examples of recent deceleration (or absence of acceleration), particularly in the Swiss Alps (e.g. Aget – see PERMOS 2019 – or Dirru – see Delaloye et al. 2013 – Rapidly moving rock glaciers…- Cicoira et al. 2019 - Water controls the seasonal rhythm of rock glacier flow – and Kummert et al. (under final review in ESPL) Pluri-decadal evolution of rock glaciers surface velocity and its impact on sediment export rates towards high alpine torrents). Probably Val Sassa and Val dal'Acqua rock glaciers in the Swiss national park have done so, but on a longer term since the end of the LIA (e.g. when comparing Chaix 1923 - https://www.persee.fr/docAsPDF/globe_0398-3412_1923_num_62_1_5609.pdf - p.11 and more recent measurements ba the National parc - https://www.parcs.ch/snp/pdf_public/2016/33398_20160921_121930_Sassa_Aqua_Bericht_2012.pdf, the movement rate appears to have been divided by 20 along the last century) . There are also some examples in Roer's PhD (2005). See Roer et al. 2005 - Rockglacier acceleration in the Turtmann valley (Swiss Alps): Probable controls

*Of course we agree with your statement. However, the second half-sentence is decisive for the statement, which says:"… relative to the speeds at the beginning of the 1950s". We are not aware of any studies that have measured lower or constant flow velocities in recent years than in the 1953 - 1970 epoch. However, we have to put the statement into perspective, since the uncertainty of the mean value only allows the conclusion that the flow velocity remain constant.*

What is the mean velocity of RG04 ? This must be given. What is the uncertainty of the values.

*The mean values range from 0.08 m/yr (1997-2006) to 0.06 m/yr (2012-2017). So we agree that we should not really speak of a decrease in mean velocity, but rather of a constant mean velocity, considering uncertainty. What is decreasing, however, are the maximum velocities. As stated above in the revised version we will give uncertainty values for the means.*

L452 (and others around). Why to be so precise in the values, taking into account their uncertainty?

*We agree. We will reduce the value by a maximum of two decimal places and also indicate the uncertainty when giving the values.*

L.451-453. Increase of the 2012-2017 velocity in comparison to which period ? Note that a pluri-decadal acceleration by a factor 2 to 10 has been observed in the Swiss Alps as well (PERMOS 2019 or other related documentation), e.g. Gemmi/Furggentälti, Grosses Gufer, Tsarmine.

*Compared to the epoch 1953/54-1970/71. We write this in the introductory sentence to this chapter (P18L444-446). We will include the PERMOS report and relativise the statement to: "…one of the highest relative changes in flow velocity compared to other studies."*

L.453-454. About rock glacier destabilization, see also Delaloye et al. 2013, Eriksen et al. 2018, Marcer et al. 2019 (already mentioned earlier in this review) and Marcer et al. 2020 - Investigating the slope failures at the Lou rock glacier front…

*We are aware of these publications, but the maximum number of references on the cryosphere is limited to 80. However, we will check whether the publications mentioned are more appropriate than those already cited.*

L.456-458. Agreed, but this must come in the discussion part and must be explain in details (provide maps/topographical profiles, etc.)

*We will take this into account in the complete revision of the results and discussion part of the paper.*

L.459. "The relative changes regarding the remaining rock glaciers ranges between 23.45% and 271.87%" is a huge difference ! 23% means about constant velocity and 271% an acceleration by a factor close to 4 ! This is not the same behavior.

*We do not state that the behaviour is the same, but only present the range of values that the other rock glaciers show. However, we will emphasise the big difference and the resulting diversity of behaviour.*

L.459-460. I don't understand the sense of the sentence… If you remove RG04, because it has a very low slope, then one could say that higher elevated rock glaciers (in the set) are steeper, but not that they change their relative flow velocity to a greater extent. If they do so, this is then because of their elevation (and eventually thermal state/structure) or steepness ?

*We will remove this sentence as referee 1 has noted that this statement is not valid due to the small sample size.*

L.463. What are these "topographic factors" ?

*Here we mean the topographical factors listed in Table A1 (exposition, slope, elevation). We will specify this in the revised version.*

L.464. "On rock glaciers RG 01 and RG 08, higher flow velocities have been measured between 1953/54 and 1970/71 compared to the subsequent periods". Be more precise. What are the subsequent periods ?

Agree... but there was then an increase since the 1990s.

*By the following epochs we mean the epochs 1970/71 - 1982 and 1982 – 1997 for RG 01. Regarding RG 08, the highest maximum flow velocities (by manual mapping) of the entire study period were measured between 1953 and 1970.*

*We will precise this.*

L474-476. "These peaks might not be found on the other rock glaciers due to superimposing effects over the long time steps and indicate a slightly different sensitivity, response or response time of individual rock glaciers to intra-annual, inter-annual or multi-annual fluctuations in external forcing parameters." Obscure sentence, which must be either precised, or removed.

*We will precise this sentence.*

L476. "the three investigated rock glaciers". Which ones ? There were only two mentioned at the beginning of the paragraph

*It must say two. We will correct this.*

L474. "differing substantially in the other characteristics measured". I do not understand.

*Here we wanted to express that the rock glaciers are the two with the lowest elevation, but differ in size, slope, exposure.... We will discuss this in more detail.*

L479. « higher error ». To be precised.

*The mean error for the 1997-2006 epoch is 0.08 m/yr. In comparison, it is 0.02 m/yr for the 2012-2017 epoch. We present this in chapter 4.1 but we will also specify it here.*

L481-2. "analyzed for altitudinal zones of 20 meter ». Why to do so ? And not comparing central to marginal zones, or else ?

*We looked at the altitude zones because we wanted to see if there was an altitude-dependent trend. As described above, we will show the active area and its change. Additionaly we will display flow velocity in violin or desity plots to better visualise the different reactions.*

L482-4. "rock glaciers do not move uniformly, but have zones with higher and lower flow velocities". It must come at the beginning... and frame the velocity analysis of the previous sections.

*We will state this at the beginning of the section of flow velocity results.*

L484-5. in the terminal section of the rock glacier ? In the front would mean in the frontal (talus) slope.

*Yes, we mean the terminal part of the rock glacier. We will correct this.*

L489. Relative instead of « percentage »

*We will correct this.*

L491-2. "This could point to the fact that from 2006 to 2017 higher elevated rock glaciers enter an unstable state as a reaction of changes in the external forcing". This is a very tricky interpretation, which in any case must be moved to the discussion section.

*As mentioned earlier, we will restructure the results and discussion section and will take this into account when doing so.*

L493. There is no lag to permafrost temperature. What temperature is talked about ?
L.494. "temperature limits or similar". I do not understand.
L.494. Time lack or time lag ?
*Here we wanted to express rock glacier RG 05 only shows a significant increase in flow velocity from 2006 onwards, especially in the higher altitude parts of the rock glacier. The assumption was that the delayed acceleration in comparison to the other rock glaciers*

*examined occurs due to the higher elevation. However, as Referee 1 also pointed out, our sample is too small to be able to prove this. We will therefore remove this hypothesis.*

L500. Figure 5. Great figure… if made larger. This is obvious here that most rock glaciers are not moving uniformly both in space and time. The multi-decadal kinematic analysis must imperatively be conducted on rock glacier sub-areas separately.

*We will respond to your proposal as described above. Although we have already tried to highlight the different kinematic zones and their development in the text, we will try to make this even clearer.*

Unit of the color scale ?

*The unit of the color scale ist given in the figure for every rock glacier individually in the map of the 2012 – 2017 epoch. Admittedly, this is a bit small and will be enlarged in the revised version.*

L.501. Section 4.4 is mostly an hypothetical discussion, not results.

*In the revised version, this chapter will be included in the discussion section, more systematically supported by literature.*

L579. Figure 7. Never start a chapter with a figure. Moreover, I don't understand what is this 3D displacement. Is it the vertical shift at fixed locations ?

*Admittedly, 3D displacment is not the correct term. The method is outlined in Figure A1. It is an established method to calculate the changes between two point clouds. It is commonly known as M3C2 (c.f. Lague et al. 2013). Since we did not use the M3C2 algorithm implemented in CloudCompare, but rather followed Fey and Wichmann (2017) and calculated this in SAGA LIS, we do not call the procedure M3C2, although it is very similar.*

*As described above, we will work with DoDs in the revised version. Tests have shown that this will hardly change the results, but we think it will make the results more accessible and easier to compare with other studies.*

L582. "0.031". Unit ? How is such a value calculated ? What is then its meaning for the rock glacier geometry change ?

*The unit is m/yr. This value is derived by calculating the mean of all surface elevation changes and than deviding it by the number of years beween measurements. Since there was hardly any subsidence due to extension or ice melting on RG 05 until 2006, the positive value could be due to debris input from the steep south-western adjacent slopes or to erroneous measurements. We will redo the analysis of the surface elevation changes and the volume changes and take better account of the errors and uncertainties.*

L592. This is not a mass balance, as the value looks not to be computed over the entire landform, which is also changing in geometry.
L608. Figure 8. How is this calculated ? How to take into account the geometry change ? The quality of the figure is poor (labels are much too large for instance).

*The values were calculated over the entire rock glacier, by gridding the point cloud based surfaces elevation changes, only snow patches were excluded. What is true, however, is that the calculation of the volume from the point cloud is not quite trivial. We will replace the volume calculation with a classic DoD volume calculation and improve the associated figure.*

L614-5. This is a setting, not a result. This should come before any kinematic (or morphodynamic) analysis. In addition, the aspect is for sure very different as well. What about the connection to the upslope unit ?

*We will include a small chapter in the results section describing the results of the rock glacier inventory and, in more detail, the characteristics of the rock glaciers studied.*

L616. "These changes are mostly spatially clustered, but in some cases they also show a clear temporal clustering". I guess, this is what is explained in the next paragraphs? If not, please do so.

*Yes, this is described in the next paragraphs.*

L.618. "Overall, the picture already described for the general trends is confirmed." That is.. ? What is this picture ? What are the general trends ?

*The general trends are described in the previous chapter. These are: the values are predominantly in the negative range; For most rock glaciers the mean values of the point cloud based surface elevation change in the first epoch are very close to zero and show a sharp shift to the negative in the epoch 1970/71-2006 or 2006-2012; almost all rock glaciers show values of the point cloud based surface elevation change in the negative and the positive range. We will redo the surface elevation change and volume calculations and will take this into account in the complete revision of the results and discussion section.*

L620 (and elsewhere). Avoid all "clear" and "when looking"... You have to show/express for the reader what is so "clear" "when looking", but not let him figure out.

*Agree. We will take this into account in the complete revision of the results and discussion section.*

L621. "Therefore, the characteristic topography for rock glaciers is formed". I do not understand.

*Here we wanted to express that in the negative and positive patterns of the point cloud based elevation changes one can recognise the development of the flow bulges through compression and extention. We will formulate this differently.*

L622. "or the rock glacier advances". It does. It has been shown before. But how is this scattering looking like ?

*Here we wanted to express that the rock glacier advance can also be seen in the patterns of the point cloud based surface changes. We will reformulate this.*

L622. « changes in activity ». Activity in what ? A vertical displacement is in particular the sum of the downslope movement of the rock glacier, the strain pattern

(compression/extension), aggradation/melt of excess ice. If the flow rate of the rock glacier is increasing, the related component of the vertical movement is increasing proportionally.

*Activity is the wrong term in this context. Here we wanted to express that the patterns and magnitude of negative and positive changes are changing spatially and temporally. We will rephrase this.*

*As we show in chapter 4.7 and figure 10, flow velocity and vertical movement do not necessarily change proportionally. This was also shown for seasonal changes by Ulrich et al. 2021.*

L625. "inactive". The activity must be related to the flow rate, not the vertical movement.

*Here we refer to the flow rate. We agree that we need to precise this because this chapter is mainly about point cloud based elevation changes.*

L627. "show hardly any 3D displacements » Is it not a question of scale ? More generally, what about the uncertainty ?

*Yes, of course it is a question of scale, but the rock glaciers mentioned hardly show any values above the error (LoD) value. We will reformulate the sentence: "...show hardly any significant point cloud based elevation chenages." in addition to the error value (from stable surfaces), we will perform an uncertainty analysis following Anderson (2019).*

L628. Where to look at on the figure ?

*This can be seen in Figures 7, 8 and 9. In Figure 7, this is evident from the shift of the median towards more negative values in most cases. In Figure 8, the volume balances become more negative in most cases, and Figure 9 shows this spatially on the point cloud-based surface elevation change maps.*

L628. "Here". Where ?

*By "here" we mean in the case of the other rock glaciers mentioned in the previous sentence. We will rephrase this.*

L629. What are these active and inactive areas ? I guess the terminology is inappropriate.

*Yes, the therminology is not correct here. We mean areas with higher and lower changes in the point cloud based surace elevation change. We will rephrase this.*

L629. "eleveation dependency". Absolute or relative to the rock glacier extent?

*By this we mean that the investigation of the altitude zones on the respective rock glaciers shows no altitude-dependent change. We will make this clearer.*

"Looking" at the figures, it becomes obvious that aggradation has frequently occurred at the front, whereas subsidence is systematic in the rooting zone, no ?

*If one considers all time periods and rock glaciers, we believe that this requires a more differentiated interpretation.*

L632. "show very clear activity in subsidence". How much, please ?

*The mean annual surface elevation change for RG 05 decreases from -0.019 m/yr in 1970 - 2006 to -0.023 m/yr in 2012 - 2017. This is also reflected in the volume changes (Figure 8) and spatially better resolved in the elevation classes in Figure 9. We will carry out this analysis again with DoDs and will give values in the revised text.*

L.635ff. Figure ? Where to see that ? Could it be else (than what has been observed) ? How is a null or a positive balance possible ? There should be a feeding of the rock glacier, which is equaling or exceeding the melt of excess ice at the front. For most rock glaciers, it cannot be reached because the motion rate is too fast (should be only a couple of cm/year maximally for most landforms) or there is no connection with any active feeding mechanism (for all glacier forefield-connected rock glaciers or when a small glacier occupied the rock glacier rooting zone during the Little Ice Age, what should be the case for most rock glaciers of concern by this study).

The methodology to calculate the volume balance must be explained, as well as its limitations and uncertainties.

*The slighly positive values are due to the errors and uncertaunties in the measurements. We will make the analysis of the surface and volume changes based on a DoD analysis and determine the errors and uncertainties based on Anderson (2019) - Uncertainty in quantitative analyses of topographic change: error propagation and the role of thresholding.*

A volume balance cannot be calculated for a rock glacier unit, which is not entirely covered by the data.

*The data cover the entire area of the rock glacier, although snow lies on the surface of the rock glacier in some small areas, as this would lead to bias in the results, the areas in which snow lies in one epoch were excluded for all epochs. Therefore, in our opinion, this approach is valid for the comparative analysis of volume changes.*

L637-8. Provide figure.

*We refer here to figure 8. We will include a reference to this figure.*

L650. Figure 9. The figure is very interesting, but too complicated, too small, and almost impossible to read

*We will split the figure and make one figure with the diagrams and one with the maps.*

L657. Provide illustration, map, figure.

*As all referees are in favour of shortening the whole paper and referees 1 & 2 suggest deleting the chapter on special cases, we will only mention this briefly in the discussion. We will nevertheless add the road to the maps of the rock glacier in Figures 5 and 9.*

L660. What about local loading (by displaced debris) ?

*This might be an explination as well. We will discuss here this.*

L664. Provide values !

*We will insert them in the text and refer to the corresponding figures. In addition, we will indicate the uncertainties in the revised version.*

L665. But most rock glaciers in the European Alps accelerated since the 1990s ! Why to state here specifically that "a strong increase in flow velocity was measured since 1997, which makes a delayed reaction of the rock glacier to the road construction 17 years before very likely". This is tricky and even false (i.e. no specific reaction).

*We do not mean the acceleration of the rock glacier in general since 1997, but, as described before, the rather unusual change in the flow pattern. The highest velocities occur at the in the area of the front in the time beween 1953 and 1982, after construction of the road the highest flow velocities are found below the road. Maybe the explenation of this is a bit speculative, we will only mention the possible antropogenic influence of the rock glacier in the revised version, in which the entire chapter will be shortened significantly.*

L.667. "It is known". Add reference(s)

*We will add references e.g. Ikeda and Matsuoka (2002).*

L667. "both factors". Which factors ?

*The topographical effects and the loss of ice mentioned in the sentence before. We will precise this.*

L669. Slope, altitude and ice occurrence are not an internal forcing. They are almost not changing over time. The ice/water content ratio does it.

*We will correct this accordingly.*

L669. "It is evident". ???

*We describe this in the following sentences for RG 04 and RG 08.*

L690. Thermokarst lakes "become a more common feature on rock glaciers due to warming and degradation of permafrost ». But there are not so frequent ! And only where massive (glacier) ice is embedded in the rock glacier.

*We agree. If we retain the passage, we will add "...where massive glacier ice is embedded in the rock glacier".*

L.693. "shifted its location". Or evolved in size and location consecutively to rapid ice melt at its margins.

*Of course the lake evolved in size and location consecutively to rapid ice melt at its margins, but it also shifted its location due to rock glacier flow.*

L705ff. This section must be heavily synthetized. See also comments on vertical movement in 4.x.x. Has not been reviewed, because the 3D displacement is somehow an obscure concept to me in this paper.

*We will redo the analysis of surfave elevation changes and volume changes as described before. Having done this, we find this aspect of linking flow velocity and surface elevation change interesting. There is only one study (Ulrich et al. 2021) that deals with this in a seasonal context and there are no studies on this yet for a long-term consideration.*

*However, we agree that this chapter can be shortened. In addition, we will change the figure by removing the interquantile range and instead show the uncertainty ranges of the surface elevation and the flow velocities.*

Figure 10 looks very interesting, whereas it should be adapted to rock glacier sub-areas. The insert in the upper right is not fully necessary.

*We will remove the insert.*

Table A1:

Elevation: I've tried to identify the rock glaciers on Google Earth. I don't know how these elevations have been determined, what do they represent. In particular, the max elevation appears to be often exaggerated.

*We will include a kmz file with the rock glaciers in the supplementary material to make them easier to find on Google Earth. The elevations were calculated from our ALS dataset from 2017. This is available in the coordinate system ETRS89 / UTM zone 32N (EPSG:25832). In this coordinate system the ellipsoid GRS 1980 is used, so the elevations are m above GRS 1980. Google Earth uses the WGS 84 / Pseudo-Mercator (EPSG:3857) coordinate system. This uses the ellipsoid WGS 84, which may be the reason for inconsistencies.*

Connection to the upslope unit : Reference and abbreviations ?

*The classification was done according to the IPA Action Group: Rock glacier inventories and kinematics - Baseline Concepts  Inventorying Rock Glaciers. We will  insert the reference and explain the abbreviations.*

RG 01 : I would say GFC for the main unit.

*We agree. However, one lobe is also TC, so we will keep this but change the order.*

RG 03: GFC. There is no glacier in connection with the rock glacier at present.

*We agree. We will change this.*

RG 04: GFC. There is no glacier in connection with the rock glacier at present.

*We agree. We will change this.*

RG 05: Not sure about the site. The rock glacier I guess is RG-05 is TC, but maybe it is another one.

*No you are right. We will change this.*

RG 06: But for sure with a glacier in the rooting zone during LIA, as attested by the thermokarst lake development in probably glacier ice embedded into the rock glacier.

*We know that there is clear evidence that a glacier must have been in the root zone during the LIA. However, the glacier inventory (Fischer et al. 2015) does not show a glacier here. We will add a note that there must have been a small glacier there.*

RG 09: GFC. Why TC ?

*Since in our understanding it is a poly-connected rock glacier.*

Connection to the upslope unit and area covered by 1850 glacier extent : These two characteristics show that the rock glaciers cannot be treated all in the same way. This is extremely important.

*We will take these two characteristics more into account when revising the discussion.*

---

## Author Response (AR1)

We would like to thank all reviewers for seeing the potential and relevance of our study and for supporting us with their objective, helpful and constructive comments. These have certainly helped to improve our manuscript.

We have made the following major changes to the analysis and the manuscript:

- Adoption of a classical digital elevation model of difference (DoD) analysis instead of the previously used "3D displacement" analysis to calculate the volume and surface changes

- Application of the LoD to calculate mean flow velocities

- profound uncertainty analysis

- Comprehensive restructuring, reformulation and shortening of the results and discussion section

- Separation of the methods section into materials and methods

- General improvements, rewording and shortening of the manuscript

- Comprehensive revision of the illustrations

- Preparation of supplementary materials

In the following, we respond point-by-point to the comments of the individual reviewers. Please find our answers in blue in the text.

**Referee 1**

**General comments**

This is a very interesting study investigating the long-term evolution of a relatively large sample of individual rock glaciers. Despite the increasing number of studies on rock glacier dynamics and evolution, there is still a lack of knowledge on the past velocities of rock glacier. This study aims at filling this gap and gives very interesting results. The analyses are thorough, very detailed and original. The errors are systematically considered and their analyses are carried out in depth.

I have however a major concern regarding the length and the structure of the manuscript. First, the text is very long and it should be reduced by about 20%. Second, and most important, the manuscript is not well structured. The results and discussion are merged into a single very long chapter, which does not allow the reader to have a clear view on the most important results of the study. The results must definitely be separated from the discussion, which is the classical way for a research paper. The references to the literature must be systematically moved to the Discussion chapter, allowing the keep the Result section more strictly factual (typical examples P16L424-428). There is also a countless number of subtitles. As a consequence of all of this, we get progressively lost. In the end we lose the main information, which is a pity because the quality of the analyses is very good and the results very interesting. Therefore, a strong effort must be made to improve the structure of the manuscript and to make it clearer.

Examples of modifications to the structure to be made :

- Move chapter 4.1 after 4.2
- Chap 4.4 (and 4.3.3., which should be merged with 4.4) should be moved in the Discussion and condensed.

The difficulty for such a study relies on its intrinsic interest : whereas similar studies generally consider one or two landforms, here a large amount of data is available for 9 rock glaciers. Thus, the authors must find a way between presenting sufficient data without losing the reader in two many details. A way to do it would be to focus more on the general trends and to reduce a bit the analyses of the exceptions and of the special cases.

*We agree with the comments on the length and structure of the manuscript. As we have generated a lot of data on the different rock glaciers, it was very difficult to highlight the really important results and discussion points. However, your comments have led to more clarity about what the important statements and results are. We will more clearly provide the important points and elaborate them better. In addition to the proposed changes to the introduction, we separated, restructured and condensed the results and discussion sections for better structure and clarity. In the results section, we present the strictly factual results of the rock glacier inventory, the changes in the meterological forcing, the flow velocity analysis and the surface elevation change analysis, particularly the last two points in a more general way. In the discussion section, we discuss the flow velocities and volumetric changes in the regional context and work out the similarities and differences in the reaction. As a further point, we relate the flow velocities and volumetric changes to the changes in the forcing parameters (temperature, snow cover and precipitation) and discuss them more systematically on the basis of the existing literature.*

The introduction is a bit lengthy and not well structured. Up to line 55 it's a long summary on the general characteristics of a rock glacier. Not everything is useful, thus I suggest to shorten this part and to keep only what is necessary. Another issue is that we must wait the end of the introduction to know the goal of the study. Ok, the precise objective must be presented after the state of the art, but the general objective, or at least the topic of the paper must be stated much earlier. Thus, I recommend to reorganize the introduction and to better structure it (see specific comments).

*We revised and shortened the passage on general rock glacier characteristics. In addition, we state the aim of the study in a sentence at the beginning of the introduction.*

The state of the art is generally good, but additional references on the current state/velocities of rock glaciers, including destabilizing ones, could be added. For example Kummert et al. 2018, Vivero & Lambiel 2019, Marcer et al. 2021,…

*We have added a reference to Vivero & Lambiel 2019 and Marcer et al. 2021 to the section on rock glacier destabilisation.*

*P2L58-58*

In such a study it would really help to have a Google Earth link to visualize the rock glaciers, or/and pictures of each rock glacier.

*We agree. In complement to figure 1, which showes two pictures of the studied rock glaciers, we included a .kmz file with the locations of the rock glaciers in the suppementary part of the paper.*

The results of the rock glacier inventory are presented in the Study area section, whereas the method for achieving it is presented after, in the Material and Methods section. This is not coherent. Since this rock glacier inventory is part of this study, the results must be moved in the corresponding section and removed from the Study area section.

*We agree. We mention the previous studies on rock glaciers in the Kaunertal in the study area section and present the results of the inventory in the results section.*

The calculated 3D displacements are changes normal to the surface. As explained by the authors, they are an alternative to the traditional DoD, and even a better quantification of the thinning/thickening processes on an ice-sursaturated permafrost body (see Vivero & Lambiel 2019 for a similar study). But this is not 3D displacement. The latter is rather a displacement that considers the 3 components x, y and z. As such this defines the displacement parallel to the slope angle, and thus the real displacement, contrary to the horizontal 2D displacement. The titles and text related to this must then be reformulated.

*We agree that 3D displacement is a misleading term. As Referee 2 has pointed out, our methodology for calculating volumes by gridded '3D displacements' is invalid as this leads to small but systematic errors in*

*volumes. Therefore, for the calculation of the volumes we appied the classical 2.5 D method via DoDs and determined the uncertainties according to Anderson (2019). Since a test has shown that there are hardly any differences in the representation of surface changes between the previous approach and the DoD method, we present these as DoDs as well in order to avoid confusion.*

*Although there are some studies that directly compare point clouds, as mentioned in your comment, we have come to the conclusion that using DoDs throughout the paper increases comparability to other studies and enhances comprehensibility.*

If I understand well the chart on Snow cover onset, snow arrived roughly early September around the years 2010. This means that what you consider as the snow cover onset in fact corresponds to the first snow, meaning that snow can then melt completely until new snow falls. Hence, this parameter cannot have any influence on the rock glacier kinematics. Much more important is the date when a substantial snow cover is established (~50 cm), allowing ground insolation. In addition, I suggest to add as a parameter the date of complete snow melt in spring. This has a strong influence on the MAGST and thus on rock glacier kinematics. See PERMOS 2019. Permafrost in Switzerland 2014/2015 to 2017/2018. Noetzli, J., Pellet, C. and Staub, B. (eds.), Glaciological Report Permafrost No. 16–19 of the Cryospheric Commission of the Swiss Academy of Sciences, 104 pp.

*For the analysis of snow cover onset and duration, we followed the criteria described in Peng et al., 2013. They describe the snow cover onset as the first day of the first five consecutive days with snow in fall (considered September to January) and the end of the snow cover as the last day of the last 5 consecutive days with snow in the melt season (February to July). The duration is calculated by counting the number of days between the snow onset and snow end.*

*We agree that this is probably not the decisive factor for the morphodynamic development of rock glaciers with regard to snow. We made a further analysis, for the onset of a substantial snow cover (50cm) and the date of complete snow melt (P11L277-280). We present (Fig. 5 and Fig.12.), describe (P15L382-383) and discuss (P29L635-649) the results.*

The interpretations of the velocities and surface changes regarding the external parameters are sometimes rather hypothetical and should more systematically rely on existing literature. This would be much easily achieved by moving these interpretations in the Discussion chapter.

*We agree with this.*

*We have moved the interpretation of the velocity and surface height changes to the discussion. They have also been rewritten, shortened and more literature has been added.*

*P26-29L573-654*

**Specific comments**

P1L12. Two times "change" in the same sentence.

*We will replaced change with shift.*

P1L20. In the rest of the manuscript you don't talk about vertical 3D, but only 3D. Be consistent. But take also in consideration my comment above about 3D.

For a detailed answer, see comment above. We changed the methodology completely to a DoD analysis. The terms vertical 3D and 3D displacement have been replaced with the expression surface elevation change.

P2L31. **are** responsible

*We corrected this.*

P2L32. **generally** coarse debris layer (the coarseness depends on the lithology).

*We added generally.*

P2L33. landforms

*We corrected this.*

P2L38. Remove "also". If the origin is periglacial, then the ice forms by freezing of water.

*We removed also.*

P3L77-80. Here you present the results of a specific study on velocity variations for selected rock glaciers. But it must be moved around L60, where you talk about rock glacier velocities. In addition, it appears weird to give details for a specific region only for one study. Thus, either you stay more general, or you keep these details but, in the meantime, you must give similar details for the other referenced studies.

*We have decided to describe the details of the study by Groh & Blöthe (2019) in more detail, as they cover the same research area but with a different temporal and thematic focus.*

*We moved the more detailed description of the study to chapter 2, study area, as it fits better here.*

*P4L101-104*

P3L85. **of** rock glaciers

*We corrected this.*

Figure 1: add the location of the study area in Austria; add the location of the highest summit.

*We added the national borders to the overview map for better orientation and included the location of the highest peak.*

P4L103. Why "pseudo" ? It sounds weird.

*The authors of the permafrost map (*https://doi.org/10.1594/PANGAEA.917719*) refer to it as a pseudo probability of permafrost being present but do not specify the term. In the corresponding publication (Schrott et al. 2012), pseudo probability does not appear either. However, as this is not relevant for our study and the permafrost map is only shown for illustration purposes, therefore, we removed peudo.*

P4L106-108. Obviously the road was built for the ski activities. You could make it clearer and say a bit more on the anthropogenic influence.

*We have not described anthropogenic influences in detail, as these have little influence on the development of rock glaciers. An exception is RG03, which is intersected by the road. We have clarified this in the chapter study area.*

*P4L93-96*

P6L127. To avoid repetition replace the second "Berger et al. (2004) by "The latter".

*We replaced this.*

P6L150-154. This refers to the state of knowledge on factors controlling rock glacier kinematics. Therefore, it should be moved into the introduction.

*As this information is already included in the introduction, we deleted it from the methods and material section.*

Table 1: Mi**n**istry

*We corrcted this.*

P7L170. This is an open reproach towards the company that can be critical. I suggest to moderate your sentence.

*This was not intended as a reproach to the company, but we understand that it can be understood as such and changed the sentence.*

Table 2: Uniformize the font

*We corrected this.*

P8L200-201. How many GCPs did you use ?

*We used 101 GCPs, which we picked very carefully directly from the point cloud in stable areas and as evenly as possible over the entire study area.*

*We specified the numer of GCPs (P8L190)*

P10L240. ha**d**. In general, check the tenses. Sometimes the present is used, sometimes the past (L245: better were than are).

*We will checked and corrected tenses used throughout the text.*

P11L274. Figure number ?

*It should read figure 4. We will add the figure number.*

P12, chap. 3.6. See my general comment on the 3D displacements.

*See your answer in the general comments.*

P12L300. **a LoD**

*We corrected this.*

P13L320-322. Syntax problem with this sentence.

*We rephrased the sentence.*

*P13L320-322*

P15L366-367. The end of the sentence is strange.

*We wanted to express the reduced positive temperature trend of winter and autumn temperatures compared to summer and spring temperatures. This sentence was removed during the reformulation.*

P15L367. El**e**vation.

*We corrcted this.*

P15L375. You could complete with additional references.

*In an earlier version of the manuscript, we had included additional references at this point, such as Gruber et al. (2004). Unfortunately, The Cryosphere limits the number of references for research papers to 80, so we had to remove some references.*

P15L389-390. I don't understand this sentence. You mean that P increased from 931 mm/yr to 957 mm/yr at Weißsee ? Please reformulate. And in the following lines it is not clear of which station you are talking about. And why not showing the data for Weißsee station ?

*We wanted to express that in the period under investigation (1953-2017) the mean annual precipitation of the station Plangeroß was 931 mm/yr and in the case of the station Weißsee in the period since the recording (2007-2017) the mean annual precipitation was 957 mm/yr. In the following passage we describe the data of the station Plangeroß.*

*We completely rewrote this passage.*

*We plotted the data of the Weißsee station in Fig. 5.*

P16L408. Honestly the tiny decrease in the snow duration cannot be considered as a trend. it only takes one year with a positive anomaly for the trend to reverse. And how do you calculate the snow cover onset ? From which snow depth do you consider that the snow cover is permanent ?

*We agree that this cannot be seen as a trend. For the analysis of snow cover onset and duration, we followed the criteria described in Peng et al., 2013. They describe the snow cover onset as the first day of the first five consecutive days with snow in fall (considered September to January) and the end of the snow cover as the last day of the last 5 consecutive days with snow in the melt season (February to July). The duration is calculated by counting the number of days between the snow onset and snow end. We agree that this is probably not the decisive factor for the morphodynamic development of rock glaciers with regard to snow.*

*We made a further analysis, as suggested in the general comments, for the onset of a substantial snow cover (>50cm) and the date of complete snow melt and integrat this into the study.*

*Furthermore, we have rephrased it so that it is not dicribed as a significant trend. (P15L382-383)*

P16L422. How much were the velocities for this period ? According to Fig. 8 they should not have been much higher than 0,5 m/yr. Such displacements should not have provoked decorrelation.

*In the period 1953 - 1971, the maximum flow velocity for RG 8 occurred in the area of the front and could only be determined by manual mapping of a few individual blocks. The maximum flow rate was 1.66 m/yr, which corresponds to a total movement of 29.85 m during this period. Therefore, the rate of movement could no longer be determined by image correlation. One should not confuse the maximum flow velocity with the average flow velocity. The maximum flow velocity for RG 08 is still over 1 m/yr in the following epoch (this is somewhat difficult to see in Figure 4, as the maximum flow velocities in the boxplot are outliers and are therefore only shown as dots). Based on the topography, movement pattern, and elevation of the rock glacier front, we suspect the "end" of a rock glacier destabilisation, as is often observed today, as a reaction to the positive temperature anomalies in the 1940s. However, this is only a speculation, as no data are available before 1953.*

*We rephrased this passage. (P16L402-404)*

P17L433. You could also reference to the PERMOS reports.

*We have now cited the PERMOS report in several places in the manuscript.*

Figure 4: What do the red dots and bars indicate ?

*The red dots indicate the mean value, the red bars indicate the insignificant measurements.*

*We added the description to the legend of figure 4.*

P18L454. Space before "Roer"

*We corrcted this.*

P18L458. Could it be differently ? At the scale of the study area the changes in external forcing are the same for all the rock glaciers.

*This sentence was deleted in the process of the rewrite.*

P19L482. Fig **5**. To compare the size of the different rock glaciers the scale should be the same, and obviously it is not (in any case it is too small to verify it).

*We have completely revised the illustration and made the scale the same.*

P19L483-484. … which is so normal ! I don't know any rock glacier showing uniform velocities on its entire surface.

*We are aware of that. We just wanted to describe the patterns we see in the flow velocity maps. Reviewer 3 places great emphasis on describing the herterogeneity of rock glacier movement, so we include this in the results section.*

P19L491-492. This is highly speculative. With such a low sample it is not possible to conclude anything about the link between rock glacier acceleration and altitude. And there is no objective explanation why higher rock glaciers would react more than lower ones.

*The reasoning behind this was hat the higher-elevated rock glacier RG05 only reacted so strongly in the last epoch, as a delayed reaction to the temperature increase due to the higher elevation.*

*We have deleted this sentence and given another possible explanation for the deviating behaviour (P26L566-572).*

P20-21, Figure 5. Figure a bit complicated. Everything is too small and thus difficult to read. I suggest to make 2 figures with 1) the charts and 2) the maps.

*We have completely revised the illustration. The diagrams have been removed, as the information they contain can also be seen on the maps. Furthermore, we have enlarged the maps and added violin plots to allow a better representation of the development of the different kinematic zones, which was requested by reviewer 3.*

*Fig. 7*

P22L523. What do you mean by "system state" ?

*By system state we mean the change of the rock glacier or permafrost body to increased temperatures, for example the formation of drainage systems.*

P22L225. "summer" instead of "autumn".

*We corrected this.*

P22L530. But the velocities are not only controlled by air temperature but also, and in a large portion, by the historic development of the snow cover, including the date of complete snow melt.

*We are aware of this, and will addressed it in the reformulated chapter of the discussion.*

*P29L635-649*

Figure 6: Indicate the period of comparison regarding the anomalies in T and P. I guess 1961-1990 ?

*You are correct, the reference period for anomalies is 1961-1990. We indicated this in the caption of the figure and in the method section.*

P25L570. But generally a long duration of the snow cover is related to a thick snow cover, and thus leads to increasing liquid water, considering also that the latter is available all along the snow melt period.

*As described earlier, we have extended the analysis of the snow cover and reformulated it in the text.*

P25L582. Looking at Fig 7 the value for RG 05 seems to be lower than 0.031

*Since figure 7 is a box plot, the mean value is not shown.*

*The mean values of the modified analysis (DoD-analysis) are shown in the revised version in Figure 9.*

P25L591. What is this other rock glacier pushed forward ?

*This is to describe the changes observed in the surface elevation change maps and associated boxplots. These are made up of positive and negative changes. We wanted to use the term "pushing forward" to describe the frequently observed advance of rock glacier front due to flow.*

*We have rephrased this to rock glacier advance in the cases where we have used pushing forward.*

P26L598-606. Please refer to the corresponding Figure. This is an example of too long paragraph regarding the data that have to be presented. The same could be said in 3 lines. Not necessary to give all these details for RG 02.

*We have taken this comment into account in revising, restructuring and reformulating the results and discussion section.*

P26L615. …different sizes. We already know this.

*We deleted this sentence.*

P27L618-621. Despite the fact that the maps are tiny (please increase the size, for instance by making 2 figures), I rather see patterns of positive or negative changes instead of scattering. Or you mean scattering at a larger scale ? But anyway the figures are too small to be analyzed by the reader.

*We have revised the illustration and enlarged the maps significantly. Wherever we have referred to scattering in the text, we have replaced it with patterns of positive and negative change.*

P27L621-622. I don't understand the sentence. And avoid references in the middle of a sentence.

*We reformulated the sentense.*

P29, chap. 4.6.1. I don't see any particular evolution for this rock glacier, since most of the landforms studied show an increase in velocities from 1997. This section is highly speculative and I suggest to delete it.

*The particular development is not indicated by a different acceleration, but by the change in the pattern of high and low velocities.*

*We have deleted the passage and describe the development of the velocity pattern in short P19L432-4433.*

P29L671. RG04 is obviously a push moraine (i.e. frozen sediments – probably a rock glacier – deformed by the LIA glacier advance). This is highlighted by the back-creeping movement towards the former glacier position and the strong subsidence, indicating high ice content. This must be considered in the analysis.

*We have deleted the chapter on special cases and taken this into account in the discussion of the results. P25L552-555*

P29L681-687. Ok for the possible reactivation, but it would be interesting to propose some hypothesis to explain such a reactivation process.

*Although we have deleted the chapter on special cases, the special case is discussed in the subsection atypical development of flow velocities. Here we also present possible explanations resulting from the different behaviour of RG04 and RG07.*

*P25-26L555-565*

P30L702. Permafrost is a thermal phenomenon. It can thus not melt.

*We deleted this sentence.*

P30L708. "… in the area of shear surfaces…" : what do you mean exactly ?

*Here we wanted to express that the horizontal movement of the rock glacier (either in the area of the shear zone or by internal plastic deformation) can result in positive and negative values in the surface elevation changes, but a net volume change can only result from material input, output, compaction or melting of the ice body.*

*We clarify this in the discussion of surface elevation changes. P30L670-676*

P30L709. "change" without s

*We corrected this.*

P31L720. "similar magnitude". Do you mean similar values ? Because it is evident that horizontal velocities are expected to be much higher than "3D" changes.

*As suggested by reviewer 3, we have carried out the analysis of flow velocities and surface elevation changes on sub-surfaces. Therefore, the description of the results of the analysis has been completely rewritten. We took your comment into account by doing so.*

**Referee 2**

**General comments**

I highly agree with the opinion of the anonymous referee #1. Because of the high potential of the present paper I would like to add some personal comments helping the authors to improve their paper.

Permafrost studies are currently a hot topic in view of climate change. The authors focus on mountain permafrost, i.e., they want to understand the spatio-temporal change of rock glacier kinematics not only locally (single rock glacier) but on a more regional scale (several rock glacier, e.g., located in a valley or catchment area). The authors want to find out how nearby rock glaciers react (geometrically) to changing environmental conditions, i.e., MAAT, precipitation, snow cover, etc.

Change detection analysis is based on archival aerial photographs and ALS data. The proper processing of these data is not easy and requires a lot of knowledge and experience. I am confident that the data has been processed accordingly.

My mayor concern is on data analysis which has already been addressed by the anonymous referee #1. I'm referring to page 12 where the concept of 3D displacements on rock glaciers is outlined. The authors should clarify the term 3D displacement. To my understanding 3D displacement is a 3D vector describing the dislocation/movement of a point or distinct feature of an object/surface in space (and time). However, the authors of the paper interpret 3D displacement as a distance into a normal direction following the idea of Lague et. al. (2013). Commonly, this algorithm is called M3C2. This algorithm has same advantages, especially in interpreting surface change and its significance. The authors' quantitative analysis of the rock glacier kinematics is based on 2D/horizontal displacements and on volume change. The latter, however, has not been carried out in a fully correct way. Since volumetric change, as implemented in the paper, is based on gridded '3D displacements' (cp. P12L303-304) the obtained volumetric changes are inherently wrong. The authors would have derived a correct result if they had taken (0,0,1) = vertical axis as a reference direction. Due to the specific kinematics (e.g., extending creep internal mass transport) and the geometry (e.g., steep frontal slope) of rock glaciers the obtained volumetric changes are preferably/systematically negative (see Figue 8). I advise the authors to re-evaluate volumetric change. The authors may use M3C2 (properly modified) or a simple difference of digital elevation models (DoD).

*First of all, we agree with you that the term 3D displacement is misleading. After reading your comment we understand that the calculation of the volumes by gridding the point cloud based suface elvation changes is subject to a small but systematic error. This has a slight effect on the absolute volumes, but hardly on the relative change of the volumes over time. Therefore, the conclusions derived from the calculations will not change. Nevertheless, we see the need to improve the calculation of the volumes!*

*Therefor, we determined the volume via a classic DoD analysis and calculated the uncertainty according to Anderson (2019) - https://doi.org/10.1002/esp.4551 . In order to avoid confusion and to facilitate comparability with other studies that determine the surface changes of rock glaciers by means of DoD analysis, we changed the other analysis of surface elevation changes on the basis of DoDs.*

In any case, the authors should include profound error analysis, i.e., significance analysis, for their velocity data (2D, horizontal) and their volumetric change results (1D, vertical, integral value obtained for an area).

*We have systematically analysed the errors of flow velocity analysis and surface elevation change analysis on stable areas. The approach to determaine the significance of flow velocities of rock glaciers is published in a peer-reviewed journal (Fey&Krainer 2020).*

*We included this value in Figure 6 and show the areas below the LoD values in the maps of Figure 7.*

*In the case of uncertainty analysis of volumes, we followed Anderson (2019) and display the uncertainty in Figure 9.*

The paper will benefit from a more formal structure, such as

*We have completely restructured the paper according to your suggestions.*

Introduction (please clearly specify the research questions),

*We revised the introduction as suggested by Reviewer 1 and clarified the research questions in the beginning of the introduction.*

Study area,

Material (First: aerial photographs and ALS data; Question: What is the reason for not using recent aerial photographs? There is lots of data available at BEV! A comparative analysis would have boosted all results obtained. Second: Supplementary material, such as meteorological data, etc.

*We agree that a separation of materials and methods would provide a clearer structure and overview, especially since many different data were used.*

*We will have revised the materials and methods section of our manuscript according to your suggestions.*

*We know the availible BEV data as we have obtained many of the historical aerial photographs here. We decided against using the available more recent BEV aerial photographs because we have collected and/or postprocessed ALS data ourselves that cover more recent study period (2006-2017) and provide the desired information we need to answer our research question.*

Methods (photogrammetric mapping, georeferencing, SfM; processing/georeferencing of ALS data; 2D-displacement measurements (orthophoto-orthophoto, orthophoto-hillshade, hillshade-hillshade; software used; precision/accuracy assessment). Question: What is the reason to use hillshades instead of original elevation data?; computation of volumetric change (method, precision/accuracy assessment); Supplementary material (explain data aggregation, etc.)

*Since the elevation model of the 1997 dataset has a relatively poor resolution, a calculation of the flow velocity by image correlation with the 2006 ALS dataset in the elevation data was not successful. We state this in the manuscript (P9L221-223).*

*In order to avoid using an additional data type for the determination of the flow velocities, we have also used hillshades for the epochs 2006-2012 and 2012-2017. The calculation of the flow velocity with the help of hillshades is a common procedure that is used in many studies (e.g. Dusik et al. 2015; Bollmann et al. 2015)*

*We restructured the methods section according to your suggestions.*

Results (present the results obtained. Maybe, you can find a good way to also aggregate the results)

*We have separated the results and discussion sections and now describe the general trends and the deviations from these trends in the results section.*

Discussion (discuss the kinematics (movement, volumetric change) of the rock glaciers in a regional context. Is there a correlation in space and time? Interrelate the kinematic information with the supplementary data.)

*In the revised discussion section, we now discuss both stream velocities and surface elevation changes in a regional context and interrelate these to the meteorological data.*

Summary (optionally)

*Since the publication is very long anyway, we decided against a summary.*

**Specific comments**

The title should reflect the content of the paper. Do you really want to address morphodynamics? Did you mean kinematics? Morphodynamics would imply process understanding.

*We changed the title accordingly.*

The paper is too long and could benefit from shortening. Maybe, it is not important to discuss each rock glacier in detail. Is there a common response? If not, why?

*The restructuring and revision of the introduction, the methods and materials section and the results and disscusion sections has led to the publication being significantly shortened, despite additional illustrations.*

Some of the figures are too small and too overloaded and thus they are not readable.

*We assume that you are referring mainly to figures 5 and 9.*

*We have significantly revised these illustrations so that the maps are now much larger and easier to read. In the process, we have deleted the boxplots contained therein, as the information contained therein can also be read from the maps.*

*See Fig. 7 and Fig. 10*

Figure 8: Mean annual volume change (m2) per 100m2 = mean surface height change (cm) !!!

*We agree and have changed the unit to m or m/yr.*

**Referee 3**

First of all, my apologies for this late review.

The paper is one of the first publication comparing the evolution of rock glacier kinematics for a set of landforms located in a single catchment area over a period of more than 60 years. The analysis is mostly based on historical aerial photographs and more recent airborne laser scanning data made available by the authors team. It permits to capture the evolution of rock glacier kinematics at roughly at a decadal time step.

This is a very interesting paper suffering however from several weaknesses, which I strongly recommend to improve in order to consider it for publication. The paper is relatively long and needs to be significantly shorten, either via the text content or the concision of some sentences or paragraphs. Any repetition must be avoided. I agree that this is a difficult exercise. The content of the illustration is mostly excellent, but usually much too small, what is deserving the paper. Some very important results are lost in large figures (e.g. evolution of the velocity flow field) and must be highlighted. Maybe some additional figures are needed.

*This was noted in the same way by all other referees. We restructure, rewritten and shorten the study. In the process, we have separated the methods section into materials and methods, as suggested by Referee 2, and separated the results and discussion sections, avoiding repetitions and deleting unnecessary parts. All illustrations have been completely revised to improve readability and information transfer.*

The structure of the paper must be revised. The description of all rock glaciers, including their spatial flow pattern and connection to upslope unit must come in entrance. It helps the eventual splitting of some rock glaciers in distinct sub-areas to be envisaged. Then the results are presented. Finally a distinct discussion section must come. At present results and discussion are mixed. The discussion must avoid to be too hypothetical.

*As noted by referee 1, we now present the results of the rock glacier inventory and the more detailed characterisation of the rock glaciers studied at the beginning of the results section. In the following, we present the results of the flow velocity, surface elevation change (former 3D Distance) and the volume change analyses. Only then will we discuss the results.*

*We have not subdivided the rock glaciers into sub-areas, but we emphasise the heterogeneity more prominently in the text and have added violin plots in Fig. 7 to enable the reader to better assess the development of different velocity zones from the maps.*

The methodology to calculate the rock glacier flow rate (single value) is unclear. It looks to be a mean of all parts of the rock glacier where any data is available, whatever the kinematic behavior. What is the sense of doing so ? Marginal areas, not moving homogeneously with the main rock glacier body, should not be taken in consideration. In addition, for some rock glaciers, it looks that calculating a mean velocity for the entire landform has no sense regarding the heterogeneity of the kinematic behavior over both space and time. Separating some rock glaciers in two or several kinematic sub-aeras could provide results (and conclusions) differing from the current ones.

*We have now applied the maximum LoD of all time slices of the respective rock glacier when calculating the mean values. In this way, areas that do not move homogeneously with the rock glacier/have values close to zero are excluded and at the same time the comparability of the mean values is preserved. In addition, we added violin plots of the flow velocities in Fig. 7. These illustrate the heterogeneity and the change of the different*

*kinematic zones on the individual rock glaciers over time. Also, areas below the LoD are now shown in this figure. In the supplementary PDF there is a figure showing the area proportions of the rock glaciers above LoD and maximum LoD.*

The "3D displacement" is not one, meaning it is not a displacement in xyz coordinates, but an inadequate terminology to define somehow a vertical movement only, but not exactly. What is the interest of applying such an approach (movement normal to the surface)?

*We agree that 3D displacement is a misleading term, this was replaced by surface elevation change. Although this approach can be advantageous in complex trains (c.f. Lague et al. 2013) , we replaced the analysis with a DoD analysis in the revised version, as Referee 2 noted that the calculation of the volume by gridding of the point cloud based surface elevation changes is not correct. Although this changed the results just slightly, this approach may be easier to understand and provide better comparability with other studies that largely follow the classic 2.5 D (DoD) approach.*

Both abstract and conclusions must be revised accordingly. They have not been reviewed, because they may change after having adapted the analysis procedure.

Since about L500, I have not performed an in-depth review.

The additional references indicated in my review are suggestions only.

The location of all rock glaciers must be provided.

*In addition to figure 1, which shows the locations of the rock glaciers in the study area, we will add a kmz file with the locations of the rock glaciers studied in the supplementary merterial part of the study.*

**Detailed comments :**

Title : I guess it is more the multi-decadal kinematics of the rock glacier which is analyzed and not the morphodynamics

*We Agree. We changed the title accordingly.*

L14 : nine or eight ? Weird statement.

*In the case of the point cloud based surface elevation change there are nine and for the flow velocity analysis there are eight. The ninth could not be considered for the flow velocity analysis due to shadowing and lack of surface structure which made image correlation impossible.*

*We specified this in more detail at the end of the introduction and in the methods section.*

L33: "or pure ice". To be avoided. This would be a debris-covered glacier.

*We removed "or pure ice"*

L38: "in part" can be omitted. Ice build-up within the ground might be the dominant process and the embedding of external (e.g. glacier, snowpatch) ice might be inexistent.

*We deleted this passage due to the requested restructuring of the introduction of Referee 1.*

L42: Active layer is consisting of unconsolidated debris (not only "boulders")

*We corrected this.*

*L34*

L43: I don't see the causal relationship between the thermal regime driven by freeze-thaw cycles and the air-filled porosity of the active layer. There is also air and water advection. Are there some references to propose ?

*We removed this sentence from the manuscript as reviewer 1 requested that we shorten the very general introduction to rock glaciers up to line 55.*

L43: "These", but which ones ?

*By these we mean the previously mentioned freeze-thaw cycles and the air convection. This sentence will also be deleted in the revised version, due to the requested shortening of the general introduction by Referee 1.*

L46: No, the debris size is not smaller, but the proportion of coarser debris per volume is less.

*We corrected this.*

*L35*

L51: What does mean "long-term"?

*Since the previous sentence talks about seasonal changes, by long-term we mean the change in thermal forcing over several years.*

L51-54: See also Cicoira et al. 2021 - A general theory of rock glacier creep…

*We added Cicoira et al. 2021.*

*L52*

L.54 : The shear zone is maximally a few meters thick.

*We specified this.*

*L42*

L57ff: This is only valid in the European Alps.

*We will have limited the validity of the statement to the European Alps.*

*L44-45*

L59. Velocity decrease since the 1990s. Which of the mentioned studies are reporting this ? I agree that some rock glaciers are decelerating, but the general trend is a significant continuation of the acceleration (e.g. PERMOS 2019 in the Swiss Alps… must not be very different in the Austrian Alps, a couple of tens kilometers eastward)

*We did not want to talk about a general deceleration of rock glaciers following the acceleration in the 1990s, but to show that there were phases (years or multiple years), in which constant or decreasing velocities were measured. This is also partly reflected in our study, in RG 01, 03, 06, 08. In these examples, lower flow velocities were measured in the period 2006 - 20012 than between 1997 - 2006 and 2012 - 2017. At least in two of the metioned studies this is also the case (e.g. Kellerer-Pirklbauer and Kaufmann, 2012 - for three rock glacier in the Hohen Tauern Range, Austria or Kenner et al. 2020 - for Schafberg rock glacier in the Swiss Alps).*

*We refer to the PERMOS report in many places in the revised discussion section.*

L70. No one of both mentioned references is showing this, but Delaloye et al. 2013 – Rapidly moving rock glaciers… - and Eriksen et al. 2018 - Recent Acceleration of a Rock Glacier Complex, Ádjet ...- are doing so. About destabilization, see also Marcer et al. 2019 - Evaluating the destabilization susceptibility of ...

*We agree that in the studies mentioned a maximum of 4m/yr is given. In an earlier version we had Vivero & Labiel 2019, which give 60-75 m in about a year, in the citation, this had to be deleted due to the limit of 80 references in the cryosphere.*

*We added "...up to sveral tens of meters..." and included Vivero and Lambiel (2019) and Marcer et al. (2019) into the citation.*

*L58-59*

L83 I would suggest "e.g." because there are other studies, sometimes difficult of being accessible. Maybe also Kummert et al., (under final review in ESPL) - Pluri-decadal evolution of rock glaciers surface velocity and its impact on sediment export rates towards high alpine torrents. See also Kääb et al. 2020 - Inventory, motion and acceleration of rock glaciers... for an example outside of the Alps

*We have added e.g. to the citation. Otherwise, we have left the citation as it is, as it supports the statement sufficiently well.*

L85. There is something wrong in this sentence

*We rephrased this sentence.*

L89. Are the rock glaciers the same, so 8 of 9 ?

*Yes, we will have specified this.*

*e.g. L71-72*

L94. Never begin a chapter with a figure. But besides, it would good to precise what are the used coordinates, what is the unit (m ?) and to add (or replace them by) lat/long coordinates.

*The coordinate system used is ETRS89 / UTM zone 32N EPSG:25832 and the unit is meter.*

*We have described this in the caption.*

L97. m. a.s.l.

*We corrected this.*

L108.  Anthropic influence on the rock glacier as well ? Which ones precisely ?

*Yes, there is also anthropogenic influence on a rock glacier due to the construction of the glacier road between 1979 and 1982, which intersects rock glacier RG03.*

*We have mentioned this here.*

*L95*

L110. Inactive rock glaciers. How was this classification done ? On which parameters ? Does it fit with the IPA Action Group Rock glacier inventories and kinematics definition ?

*Here we follow the classical classification into active - motion (from image correlation) inactive - ice contained; no measurable motion (melting, visible on DoDs) and fossil (e.g. Krainer & Ribis 2012).*

*The methodology is described in chapter 4.1 Rock glacier inventory (L165-180). We have now described the creation of the inventory in more detail.*

L113. Replace by something like : Finally, eight active rock glaciers representing different characteristics and conditions were investigated in detail regarding flow velocities and one more regarding vertical displacements

*We have adopted the formulation.*

*L289-290*

L121. The rock glacier moving downwards, it does not make sense to write that it reaches the "highest elevation". Would it not be that it is located at the highest elevation range among the nine selected rock glaciers ?

*We will put the description in the results section, as Referee 1 asked us to put the results of the inventory in the results section.*

*We have reformulated this sentence.*

*L292-293*

L129. Is the layer below the ice rich permafrost body really ice free ? Because it is very difficult to conceive an active rock glacier which is only frozen in its upper part. Where is the shearing zone developing on the long term ?

*Here we describe the results of the geophysical investigations of Hausmann et al. 2012 at the Ölgruben rock glacier (RG 01). Since we are not experts in the interpretation of geophysical surveys, we can only report the results of the study as published.*

151-153. There was a paper in 2008 (Delaloye et al. - Recent interannual variations of rock glacier creep in the European Alps) showing that there was an almost good similarity of interannual variations of rock glacier velocity over the entire European Alps, confirmed a decade later by Kellerer-Pirklbauer et al. (2018) at EUCOP - Interannual variability of rock glacier flow velocities in the European Alps. There was also a short communication at ICOP 2016 by Staub et al. - Rock glacier creep as a thermally‐‹-driven phenomenon: A decade of inter-annual observations from the Swiss Alps - showing that the interannual variations are basically driven by shifts in mean ground surface temperature for a period of about 2.5 years. For sure this is also influencing the liquid water content within the permafrost.

*This is why we write liquid water availability and not precipitation, the increased liquid water availability can have several causes, which according to Kenner et al. (2020) is mainly controlled by the ground temperature and the onset and duration of the snow cover. This is described in the introduction (L52-54).*

*We removed the explanation in the methods section, as Referee 1 noted that this should be explained in the introduction.*

153-155. The effects of liquid water availability and snow cover on rock glacier morphodynamics must be precised. Does it mean on-set and melt-out of the snow cover influencing the ground surface temperature or water equivalent of the snow pack which will melt out in spring/suemmer and directly influencing the rock glacier hydrology. Or both ?

*The influence of various drivers of rock glacier flow velocity is nicely displayed in Kenner et al. (2020), Figure 5. Although they just derive this from the study of one rock glacier, we think this is a very good overview and we are following it to some extent in our discussion of climate factors.*

*Both. As the influence of GST indiectly influences water availibility as well. The onset of winter snow cover determines the winter cooling intensity, which in turn influences the timespan with liquid water in the active layer. The strongest influence, however, is shown by the timing of snow melt, but less by the snow water equivalent (Kenner et al. 2020).*

*L630-654*

Finally, the paper is focusing on decadal velocity changes. What relation to short-term (less than annual) changes ?

*We do not know exactly to which passage this comment refers, as we are now in the method section in cronological order.*

*We refer to findings about short-term change and their relation to decadal velocity changes in the discussion.*

*e.g. L635-640*

L276ff. This is not a 3D displacement, but something else (the surface change normal to the surface). But what is the interest of doing so and not calculating simply the vertical displacement? What are the advantages on a rock glacier ?

*We have now replaced the analysis with a classical DoD analysis. See previous answer.*

L323 "showed good agreement". Please, provide values, figure or table.

*Although a quantitative comparison is difficult because the time periods of the epochs differ. The results are very consistent if we consider that the epoch of the paper by Groh and Blöthe (2019) (2003-2015) includes three epochs of our study (1997 - 2006, 2006 - 2012 and 2012 - 2017).*

*The comparison is now included in the supplementary PDF in Table S 1.*

L.326 Is the stable area so stable? In principle, bedrock is more suited to be stable than a debris slope.

*We agree that bedrock can generally be considered more stable than a debris solpe. However, on bedrock, which usually has lower roughness, the errors are underestimated, especially in the image correlation analysis. Since we know the change from our surface elevation change analysis in these areas, we know that they are stable.*

L361-363. The introductive part of the sentence could be avoided.

*We deleted the introductive part.*

L.367. A significant trend cannot be calculated over 11 years only. What is this data meaning ? Is it a difference between the mean of the two periods or a trend in 11 years as expressed in the two previous sentences.

*The two previous sentences describe the trend over 65 years at the Obergurgl-Vent and Nauders stations. The one meant compares the increase in the mean value in the last two epochs of the study, including the Weißsee station.*

*This sentence has been deleted due to the shortening of the description of climate factors.*

L370. Precise what are these seasons, e.g. spring is MAM, summer JJA ?

*We have specified this in the description of the methodology (L276-277) and in the caption (Fig.12).*

L371-372 Conditions causing heat waves in future is not the purpose of this paper looking back into the past. The sentences could be removed.

*We removed this sentence.*

L388. +152 mm in 65y. Is it a lot or not ? What is the annual value ?

*We give the mean annual precipitation value for the study period in the next sentence (931 mm/yr). In the previous sentence we give the range of trends for all the stations studied (53 mm - 241 mm). A positive trend of 152 mm in 65 years is already quite a lot in our view.*

L393-395. This sentence about precipitation predictions could be removed (not of interest for this paper)

*We removed this sentence.*

L408. Snow melt trend: How much ? What are the starting dates and durations ?

*Referee 1 has noted that the small decreases in the snow parameters are not a trend. We will therefore delete the sentence. In addition, we will reanalyse the snow parameters and add the onset of a significant snow cover (>50) and the complete snow melt as parameters.*

*L380-389*

L420. Provided max flow values are valid for a single period or as a mean for 1953 to 2017 ?

*These values are for the whole period of inverstigation (1953 – 2017).*

*We stated this in the beginning of the sentence: "For the whole period of investigation…"*

*L391*

L421. How is the mean value spatially calculated ? How is delimitated the area taken into consideration for the calculation ? It looks from the figures that they comprise marginal areas (with velocity close to 0) to sectors moving much faster.  Why not to split into sub-areas and perform a comparative temporal analysis ? See also the definition of moving areas within the IPA Action Group Rock glacier inventories and kinematics - https://www.unifr.ch/geo/geomorphology/en/research/ipa-action-group-rock-glacier/ - documents (kinematics as an optional attribute in rock glacier inventories)

*We have now applied the maximum LoD of all time slices of the respective rock glacier when calculating the mean values. In this way, areas that do not move homogeneously with the rock glacier/have values close to zero are excluded and at the same time the comparability of the mean values is preserved. In addition, we added violin plots of the flow velocities in Fig. 7.  These illustrate the heterogeneity and the change of the different kinematic zones on the individual rock glaciers over time. Also, areas below the LoD are now shown in this figure. In the supplementary PDF there is a figure showing the area proportions of the rock glaciers above LoD and maximum LoD. (see general comments)*

L.421. A mean velocity of 3.5 cm/year is rising some questions about the accuracy and reliability of the results (in particular changes over time). Is such a low value significant ?

*The value mentioned represents the mean value for RG 03 in the time slice 1970-1982. In this time period, maximum movements of 1.51 m (0.089 m/yr) were measured. The revised mean value, taking into account the LoD and uncertainty, is now 0.14 m/yr ± 0.05 m/yr for this epoch and rock glacier.*

*This has been described and changed in the text, and a table with all mean values can be found in the supplementary PDF of the study. (Table S 1)*

L424. One should note that the period 1997-2006 is marked by the peak of 2000-01 (described for instance by Ikeda et al. 2003 - Rapidly moving small rockglacier at the lower limit of the mountain permafrost belt… - and 2008 - Fast deformation of perennially frozen debris in a warm rock glacier…) and the famous 2003-04 peak (e.g. Delaloye et al. 2008), The period 2012-2017 is embedding the extreme peak of 2015 (e.g. Kellerer-Pirklbauer et al. 2018, PERMOS 2019), whereas the period 2006-2012 contains no peak of activity.

L431. 2006 is often a low, the period with the lowerst velocity recorded since 2000 (e.g. PERMOS). More generally the sentence is difficult to understand unambiguously.
L431-433. You could refer to Delaloye et al. 2008 for the low in 2006 in the European Alps and to PERMOS 2019 for the description of the entire period.

*We describe the annual acceleration and deceleration peaks measured in the section "Possible implications of changes in external forcing for rock glacier flow velocities."*

L434. RG 04 : A detailed spatial analysis (of the morphodynamics) is necessary, with the help of (time-lapsed) maps. It should be the same for the other rock glaciers.

*Referees 1 and 2 have requested that the detailed analyses of the individual rock glaciers be greatly shortened and that we rather focus on general similarities and differences in the revised version. But we will describe the general trends, simialrities and differernces of the spatial development. The development of the individual rock glaciers can then be examined by the reader in the flow velocity maps and newly added violin plots.*

L435-7. "Many studies mentioned periods of slight decrease or constant flow velocities following the strong acceleration in the 1990s". Not really. This is mostly related to the deceleration drop in 2005-06. Read the related papers (already mentioned earlier), and in particular the PERMOS reports.

*We have deleted this sentance and describe the annual acceleration and deceleration peaks as previously described.*

L437-9. There are various examples of recent deceleration (or absence of acceleration), particularly in the Swiss Alps (e.g. Aget – see PERMOS 2019 – or Dirru – see Delaloye et al. 2013 – Rapidly moving rock glaciers…- Cicoira et al. 2019 - Water controls the seasonal rhythm of rock glacier flow – and Kummert et al. (under final review in ESPL) Pluri-decadal evolution of rock glaciers surface velocity and its impact on sediment export rates towards high alpine torrents). Probably Val Sassa and Val dal'Acqua rock glaciers in the Swiss national park have done so, but on a longer term since the end of the LIA (e.g. when comparing Chaix 1923 - https://www.persee.fr/docAsPDF/globe_0398-3412_1923_num_62_1_5609.pdf - p.11 and more recent measurements ba the National parc - https://www.parcs.ch/snp/pdf_public/2016/33398_20160921_121930_Sassa_Aqua_Bericht_2012.pdf, the movement rate appears to have been divided by 20 along the last century) . There are also some examples in Roer's PhD (2005). See Roer et al. 2005 - Rockglacier acceleration in the Turtmann valley (Swiss Alps): Probable controls

*Since RG04 shows rather constant velocities, considering the uncertainties, we have deleted this statement. However, the development of the flow velocities of the above-mentioned studies are mentioned in the revised discussion.*

What is the mean velocity of RG04 ? This must be given. What is the uncertainty of the values.

*The revised mean values range from 0.12 m/yr ± 0.03 (2012-2017) to 0.14 m/yr ± 0.07m/yr (1997 -2006).*

*All mean values are included in Table S 1 of the supplementary PDF.*

L452 (and others around). Why to be so precise in the values, taking into account their uncertainty?

*We reduced the value by a maximum of two decimal places and also indicate the uncertainty when giving the values.*

L.451-453. Increase of the 2012-2017 velocity in comparison to which period ? Note that a pluri-decadal acceleration by a factor 2 to 10 has been observed  in the Swiss Alps as well (PERMOS 2019 or other related documentation), e.g. Gemmi/Furggentälti, Grosses Gufer, Tsarmine.

*Compared to the epoch 1953/54-1970/71. We write this in the introductory sentence to this chapter.*

*We included the PERMOS report and relativise the statement to: "…one of the highest relative changes in flow velocity compared to other studies."*

L.453-454. About rock glacier destabilization, see also Delaloye et al. 2013, Eriksen et al. 2018, Marcer et al. 2019 (already mentioned earlier in this review) and Marcer et al. 2020 - Investigating the slope failures at the Lou rock glacier front…

*We added Marcer et al. 2021 when referring to rock glacier destabilisation.*

*e.g. L540-541*

L.456-458. Agreed, but this must come in the discussion part and must be explain in details (provide maps/topographical profiles, etc.)

*As this is not a case study of a single rock glacier, we will only give hypothetical explanations in the discussion, as a precise analysis of the causal factors of all rock glaciers would go far beyond the scope of this paper.*

L.459. "The relative changes regarding the remaining rock glaciers ranges between 23.45% and 271.87%" is a huge difference ! 23% means about constant velocity and 271% an acceleration by a factor close to 4 ! This is not the same behavior.

*We do not state that the behaviour is the same, but only present the range of values that the other rock glaciers show.*

*We emphasised the big difference and the resulting diversity of behaviour.*

*L424-425*

L.459-460. I don't understand the sense of the sentence… If you remove RG04, because it has a very low slope, then one could say that higher elevated rock glaciers (in the set) are steeper, but not that they change their relative flow velocity to a greater extent. If they do so, this is then because of their elevation (and eventually thermal state/structure) or steepness ?

*We removed this sentence as referee 1 has noted that this statement is not valid due to the small sample size.*

L.463. What are these "topographic factors" ?

*Here we mean the topographical factors listed in Table 4 (exposition, slope, elevation).*

*We have deleted this sentence because of the shortening and restructuring of the manuscript.*

L.464. "On rock glaciers RG 01 and RG 08, higher flow velocities have been measured between 1953/54 and 1970/71 compared to the subsequent periods". Be more precise. What are the subsequent periods ?

Agree... but there was then an increase since the 1990s.

*By the following epochs we mean the epochs 1970/71 - 1982 and 1982 – 1997 for RG 01. Regarding RG 08, the highest maximum flow velocities (by manual mapping) of the entire study period were measured between 1953 and 1970.*

*We precised this.*

*L426-430*

L474-476. "These peaks might not be found on the other rock glaciers due to superimposing effects over the long time steps and indicate a slightly different sensitivity, response or response time of individual rock glaciers to intra-annual, inter-annual or multi-annual fluctuations in external forcing parameters." Obscure sentence, which must be either precised, or removed.

*We have deleted the sentence in this form.*

L476. "the three investigated rock glaciers". Which ones ? There were only two mentioned at the beginning of the paragraph

*It must say two.*

*Due to the rewriting of the manuscript, this sentence was deleted.*

L474. "differing substantially in the other characteristics measured". I do not understand.

*Here we wanted to express that the rock glaciers are the two with the lowest elevation, but differ in size, slope, exposure....*

*Due to the rewriting of the manuscript, this sentence was deleted.*

L479. « higher error ». To be precised.

*The mean error for the 1997-2006 epoch is 0.08 m/yr. In comparison, it is 0.02 m/yr for the 2012-2017 epoch.*

*We present this in chapter 5.2 and specified it here.*

*L430*

L481-2. "analyzed for altitudinal zones of 20 meter ». Why to do so ? And not comparing central to marginal zones, or else ?

*We have omitted the analysis of altitudinal zones in the revised version.*

L482-4. "rock glaciers do not move uniformly, but have zones with higher and lower flow velocities". It must come at the beginning... and frame the velocity analysis of the previous sections.

*We have drawn attention to the heterogeneity in the text (L431-441). This is also evident from the maps of flow velocities and the violin plots (Fig.7).*

L484-5. in the terminal section of the rock glacier ? In the front would mean in the frontal (talus) slope.

*We have replaced front with terminal section throughout the text when referring to the terminal section.*

L489. Relative instead of « percentage »

*We corrected this.*

L491-2. "This could point to the fact that from 2006 to 2017 higher elevated rock glaciers enter an unstable state as a reaction of changes in the external forcing". This is a very tricky interpretation, which in any case must be moved to the discussion section.

*We have omitted this interpretation in the revised discussion.*

L493. There is no lag to permafrost temperature. What temperature is talked about ?
L.494. "temperature limits or similar". I do not understand.
L.494. Time lack or time lag ?

*We have omitted this interpretation in the revised discussion.*

L500. Figure 5. Great figure… if made larger. This is obvious here that most rock glaciers are not moving uniformly both in space and time. The multi-decadal kinematic analysis must imperatively be conducted on rock glacier sub-areas separately.

*In our study, we do not claim that they move uniformly. In the newly calculated mean values, we exclude areas with flow velocities close to zero. We argue that the heterogeneity is clearly shown in the velocity maps and the violin plots (Fig 7).*

Unit of the color scale ?

*The unit of the color scale is m/yr and is given in the figure caption.*

*Fig. 7*

L.501. Section 4.4 is mostly an hypothetical discussion, not results.

*In the revised version, we have clearly separated results and discussion and supported them more systematically with literature.*

*L573-654*

L579. Figure 7. Never start a chapter with a figure. Moreover, I don't understand what is this 3D displacement. Is it the vertical shift at fixed locations ?

*Admittedly, 3D displacment is not the correct term. The method is outlined in Figure A1. It is an established method to calculate the changes between two point clouds. It is commonly known as M3C2 (c.f. Lague et al. 2013). Since we did not use the M3C2 algorithm implemented in CloudCompare, but rather followed Fey and Wichmann (2017) and calculated this in SAGA LIS, we do not call the procedure M3C2, although it is very similar.*

*As described above, we worked with DoDs in the revised version. Tests have shown that this will hardly change the results, but we think it will make the results more accessible and easier to compare with other studies.*

L582. "0.031". Unit ? How is such a value calculated ? What is then its meaning for the rock glacier geometry change ?

*This is the mean surface elevation change of the rock glacier. The unit is m/yr. Since the analysis was recalculated with DoDs, the value has also changed slightly. The overview for all rock glaciers is now shown in an additional figure (Fig. 9).*

L592. This is not a mass balance, as the value looks not to be computed over the entire landform, which is also changing in geometry.
L608. Figure 8. How is this calculated ? How to take into account the geometry change ? The quality of the figure is poor (labels are much too large for instance).

*The values were calculated over the entire rock glacier, by gridding the point cloud based surfaces elevation changes, only snow patches were excluded. What is true, however, is that the calculation of the volume from the point cloud is not quite trivial.*

*We replaced the volume calculation with a classic DoD volume calculation and improved the associated figure (Fig. 8, 9 and 10). In addition, we have noted that due to the necessary masking of the snow, in many cases it is not possible to speak of a mass balance in the stricter sense.*

*L670-672*

L614-5. This is a setting, not a result. This should come before any kinematic (or morphodynamic) analysis. In addition, the aspect is for sure very different as well. What about the connection to the upslope unit ?

*We have now described the rock glacier characteristics at the beginning of the results section (L289-306). Furthermore, in the revised version, we have included the connection with the upslope unit both in the description of the results (Sect.5.5) and in the discussion (Sect.6.5).*

L616. "These changes are mostly spatially clustered, but in some cases they also show a clear temporal clustering". I guess, this is what is explained in the next paragraphs? If not, please do so.

*We have reformulated and restructured the entire paragraph on surface elevation changes.*

L.618. "Overall, the picture already described for the general trends is confirmed." That is.. ? What is this picture ? What are the general trends ?

*We have moved and reformulated this part.*

*L499-504*

L620 (and elsewhere). Avoid all "clear" and "when looking"... You have to show/express for the reader what is so "clear" "when looking", but not let him figure out.

*We have taken this comment into account in the revision and have now largely avoided these formulations.*

L621. "Therefore, the characteristic topography for rock glaciers is formed". I do not understand.

*Dieser teil ist jetzt in der diskussion der surface elevation changes zu finden und wurde deultich umformoliert und prezesiert.*

*L676-679*

L622. "or the rock glacier advances". It does. It has been shown before. But how is this scattering looking like ?

*We reformulated this.*

*L676-679*

L622. « changes in activity ». Activity in what ? A vertical displacement is in particular the sum of the downslope movement of the rock glacier, the strain pattern (compression/extension), aggradation/melt of excess ice. If the flow rate of the rock glacier is increasing, the related component of the vertical movement is increasing proportionally.

*Activity is the wrong term in this context. Here we wanted to express that the patterns and magnitude of negative and positive changes are changing spatially and temporally. We will rephrase this.*

*We now describe this in the discussion of surface elevation changes.*

*L673-682*

L625. "inactive". The activity must be related to the flow rate, not the vertical movement.

*In the revised version, when we use active and inactive, we only refer to the flow rates.*

L627. "show hardly any 3D displacements » Is it not a question of scale ? More generally, what about the uncertainty ?

*We address this in the description of the results (L454-463) and present it in Fig. 9. The uncertainties of the analysis were determined according to Anderson (2019) and are also included in Fig. 9 and in the text. In the supplementary PDF, all numerical values are listed in Table S 2.*

L628. Where to look at on the figure ?

*This is clearly shown in Figure 9 by the surface elevation changes and is further explained and detailed in the revised text (Sect. 5.5)*

L628. "Here". Where ?

*The description of the results and the discussion has been completely restructured and reformulated so that it should now be made clearer.*

L629. What are these active and inactive areas ? I guess the terminology is inappropriate.

*Yes, the therminology is not correct in this case. We mean areas with higher and lower changes in the surace elevation change.*

*We have taken this into account in the restructuring and reformulation of the results and discussion section.*

L629. "eleveation dependency". Absolute or relative to the rock glacier extent?

*We have omitted the anlysis of the altitudinal zones in the revised version, as this should be shortened.*

"Looking" at the figures, it becomes obvious that aggradation has frequently occurred at the front, whereas subsidence is systematic in the rooting zone, no ?

*If one considers all time periods and rock glaciers, we believe that this requires a more differentiated interpretation.*

*We give a detailed interpretation of the surface elevation changes in the discussion in the revised version (Sect 6.5).*

L632. "show very clear activity in subsidence". How much, please ?

*This can now be seen in Fig. 9. Furthermore, we give values in the text wherever necessary. An overview of the values of all rock glaciers and epochs, including their uncertainty, is included in the supplimantary PDF.*

L.635ff. Figure ? Where to see that ? Could it be else (than what has been observed) ? How is a null or a positive balance possible ? There should be a feeding of the rock glacier, which is equaling or exceeding the melt of excess ice at the front. For most rock glaciers, it cannot be reached because the motion rate is too fast (should be only a couple of cm/year maximally for most landforms) or there is no connection with any active feeding mechanism (for all glacier forefield-connected rock glaciers or when a small glacier occupied the rock glacier rooting zone during the Little Ice Age, what should be the case for most rock glaciers of concern by this study).

The methodology to calculate the volume balance must be explained, as well as its limitations and uncertainties.

*In the revised analysis, only in the case of RG05, epoch 1953-1970 is the value above zero when uncertainty is taken into account. This could be explained by the fact that the glacier is surrounded on both sides by steep slopes that deposit avalanches and avalanche debris.*

*The methodology is explained in Sect. 4.4, uncertainties and errors are explained in Sect. 5.2 and limitations due to snow cover are mentions and discussed in Sect. 6.5*

A volume balance cannot be calculated for a rock glacier unit, which is not entirely covered by the data.

*The data cover the entire area of the rock glacier, although snow lies on the surface of the rock glacier in some small areas, as this would lead to bias in the results, the areas in which snow lies in one epoch were excluded for all epochs.*

*We state this in the description of the methodology (Sect. 4.4) and also note it in the discussion of the results (Sect. 6.5).*

L637-8. Provide figure.

*We refer to figure 9. In the revised version, we present the changes in more detail and in a more differentiated way (Sect. 5.5).*

L650. Figure 9. The figure is very interesting, but too complicated, too small, and almost impossible to read

*We have completely revised the illustration, omitted the diagrams and thus improved the readability.*

*Fig. 10*

L657. Provide illustration, map, figure.

*As all referees are in favour of shortening the whole paper and referees 1 & 2 suggested that we delet the chapter on special cases.*

*Therefore, we discuss the rock glaciers that show a distinctive development only briefly in Sect. 6.2 "Atypical development of flow velocities".*

L660. What about local loading (by displaced debris) ?

*We have taken the comment into account in Sect. 6.5 "Interpretation and implications of surface elevation changes".*

L664. Provide values !

*The values can be extracted from Figure 9 in the publication. As described above, the values of all rock glaciers are now included in the supplementary PDF of the paper.*

L665. But most rock glaciers in the European Alps accelerated since the 1990s ! Why to state here specifically that "a strong increase in flow velocity was measured since 1997, which makes a delayed reaction of the rock glacier to the road construction 17 years before very likely". This is tricky and even false (i.e. no specific reaction).

*We have omitted this interpretation in the revised version and only briefly refer to the unusual change in the pattern of flow velocities in the description of the results.*

*L432-433*

L.667. "It is known". Add reference(s)

*This section was deleted during the revision of the results and discussion section.*

L667. "both factors". Which factors ?

*This section was deleted during the revision of the results and discussion section.*

L669. Slope, altitude and ice occurrence are not an internal forcing. They are almost not changing over time. The ice/water content ratio does it.

*This section was deleted during the revision of the results and discussion section.*

L669. "It is evident". ???

*This section was deleted during the revision of the results and discussion section. We address this in the revised discussion.*

*L687-700 and L550-551*

L690. Thermokarst lakes "become a more common feature on rock glaciers due to warming and degradation of permafrost ». But there are not so frequent ! And only where massive (glacier) ice is embedded in the rock glacier.

*We agree with his comment. Since we have omitted the part about the special cases, no statement is made about the possible development of the thermokarst lakes.*

L.693. "shifted its location". Or evolved in size and location consecutively to rapid ice melt at its margins.

*Since we have omitted the chapter on special cases, the development of the thermokarst lake on RG06 is not discussed. A map showing the development of the lake is included in the supplementary PDF.*

L705ff. This section must be heavily synthetized. See also comments on vertical movement in 4.x.x. Has not been reviewed, because the 3D displacement is somehow an obscure concept to me in this paper.

*As we have restructured, rewritten and shortened the results section, this section has also been shortened, restructured and rewritten.*

Figure 10 looks very interesting, whereas it should be adapted to rock glacier sub-areas. The insert in the upper right is not fully necessary.

*We have removed the insert and carried out the analysis on sub-areas. (Fig. 11 and Sect. 5.6)*

Table A1:

Elevation: I've tried to identify the rock glaciers on Google Earth. I don't know how these elevations have been determined, what do they represent. In particular, the max elevation appears to be often exaggerated.

*The supplementary section now contains a .kmz file with the locations of the investigated rock glaciers.*

*The elevations were calculated from our ALS dataset from 2017. This is available in the coordinate system ETRS89 / UTM zone 32N (EPSG:25832). In this coordinate system the ellipsoid GRS 1980 is used, so the elevations are m above GRS 1980. Google Earth uses the WGS 84 / Pseudo-Mercator (EPSG:3857) coordinate system. This uses the ellipsoid WGS 84, which may be the reason for inconsistencies.*

Connection to the upslope unit : Reference and abbreviations ?

*The classification was done according to the IPA Action Group: Rock glacier inventories and kinematics - Baseline Concepts  Inventorying Rock Glaciers as described in the text.*

*L168-170*

*We have defined the abbreviations in the text.*

*L304-305*

RG 01 : I would say GFC for the main unit.

*We agree. However, one lobe is also TC, so kept this but change the order.*

RG 03: GFC. There is no glacier in connection with the rock glacier at present.

*We agree. We changed this.*

RG 04: GFC. There is no glacier in connection with the rock glacier at present.

*We agree. We changed this.*

RG 05: Not sure about the site. The rock glacier I guess is RG-05 is TC, but maybe it is another one.

*No you are right. We changed this.*

RG 06: But for sure with a glacier in the rooting zone during LIA, as attested by the thermokarst lake development in probably glacier ice embedded into the rock glacier.

*We know that there is clear evidence that a glacier must have been in the root zone during the LIA. However, the glacier inventory (Fischer et al. 2015) does not show a glacier here.*

*We added a note that there must have been a small glacier evidenced by the thermokarst lake.*

RG 09: GFC. Why TC ?

*Since in our understanding it is a poly-connected rock glacier.*

Connection to the upslope unit and area covered by 1850 glacier extent : These two characteristics show that the rock glaciers cannot be treated all in the same way. This is extremely important.

*We have taken this comment into account and address it in particular in the discussion of surface elevation changes.*

*Sect. 6.5*

---

## Author Response (AR2)

**Comments to the author**:

Dear Fabian Fleischer and co-authors,

Your manuscript received in the first review round three very detailed and critical reviews that both highlighted the relevance and novelty of the research, in particular the extensive dataset covering 9 rockglaciers over close to 6 decades and they confirm that the manuscripts seems in general to be well within the scope of TC. Besides the generally positive comments, all reviews also raised some important aspects that should be improved before publication and in brief encompass:

-improvement of the methods and its description (issue 3d-displacements/DoD for elevation change)
-improvement of the structure (proper separation of results from discussion)
-shortening of the manuscript (avoid repetitions)
-adjustment and separation of figures to make them easier to read
-minor adjustments in text

As the revisions were rather substantial the revised version went to review again (to one of the previous reviewers), and as the editor, I also read and checked the revised version again.

The re-review clearly stated the substantial improvements (figures, structure, methods) and makes clear that this is a interesting and valuable contribution with regard to longterm dynamical behavior of rock glacier in context of climate forcing. Some of the interpretation remains somewhat speculative but the presented base for interpretation is clear now.

I confirm the referees view and that the comments of the first round of reviews have been well addressed in the revised version (the structure, figures, methodological issues and the interpretation). Thus, this manuscript is now close to publication but there are still quite a few mostly minor editing issues (typos, formulations, captions).

The list of the minor comments below that of the re-review and myself the editor should be addressed by the authors before acceptance of the paper.

I thank the authors for their considerate and detailed response of how to address the points raised by the referees.

Andreas Vieli, editor
30 oct 2021

Dear Andreas Vieli,

We thank you and the reviewer for seeing the relevance of the study and also for acknowledging the improvements we were able to achieve as a result of the review in the revised version. This would not have been possible without the detailed comments and remarks of the reviewers and yourselves. We would also like to thank you and the reviewer for the improvements and comments of the revised mauskript.

Based on the comments, we have made the following minor changes to the mauscript:

- correction of typos
- improvement of formulations and captions
- minor changes in illustrations
- In some places, it has been pointed out that due to the drwbacks of the input data (e.g. snow patches) some results have to be interpreted with caution.

In the following, we respond point-by-point to the comments. Please find our answers in blue in the text.

List of comments by re-review that should be adressed
The title is now fine.

Abstract
L11: Mention the elements of external forcing. Is it a 'shift' or a 'change'?

Another reviewer noted the repetition of change in this sentence.

We have refrained from mentioning the external forcing parameters, as these are mentioned below in the abstarct. We have changed shift back to change.

L13: Please check 'nine or eight', it's not clear to me.

The half-sentence specifies that the study was conducted on eight or nine rock glaciers, depending on the analysis. This is described in more detail in the introduction (L71-74) and throughout the manuskript.

We have changed the sentence to: "…on nine or eight active rock glaciers respectively…"

L120: Table 1, N/A ? The flight altitude is a by-product of georeferencing. Fill in appropriate values.

Thank you for the suggestion.

We have now inserted the values from the respective Agisoft Metashape report files for all processed aerial images. These show the mean flight altitude over the survey area. The old values represented average flight altitudes for the entire flight campaigns which were not always representative for the study area.

4. Methods
L182: Exchange 'orthofoto' by 'orthophoto'. Please stick to the same wording, either orthophoto or orthoimage.

We agree and have now used orthoimage throughout the manuscript.

L215: The resulting raw vector maps can contain erroneous displacement vectors. Vector maps do not contain decorrelation per se. Image decorrelation might be caused by … Please rephrase.

We agree and have reformulated the sentence accordingly.

L225: What is a 'non-machting' DEM? Do you mean non-contemporary?

Here we wanted to express that the DEM might not be from the same year as the aerial photos. Therefore, you are right non-contemprary fits better.

We have replaced non-maching with non-contamporary.

L235: … and therefore DEMs (?) were still a source of error.

We wanted to express that the different quality of the DEMs can also have an impact on the accuracy of orthorectification.

We have reworded the sentence to: "… and therefore might influence the accuracy of the orthoretification."

L249: Formula 1 is a rather strange error propagation.

This formula is used by Fey and Krainer (2020) to determine the LoD for flow velocity measurments derived by image correlation. Kummert er al. (2021) use the mean value of three manual point measurements on stable areas between two image pairs as the error value for flow velocity measuremnets derived by image correlation. Since we have many measurements by using image correlation in stable areas, we do not simply use the mean value, but add two times the standard deviation. In our opinion, this provides a good and relatively conservative estimate of the error.

L266-267: The sentence is semantically correct. However, because of this
(drawback) the subsequent analysis is somehow imperfect.

We are aware of this and address this issue in the discussion of surface elevation changes (L681 – 683, L691-693, L709-711). Since permanent snow patches often occur at altitudes where rock glaciers are found, it will hardly be possible to prevent this.

We have included a reference to Figure 10, which shows the snow masks, and we have also included a sentence at this point to draw attention to the problem:

 "This implies that in some cases the entire landform cannot be considered for the mass balances, so the results of these must be interpreted with caution."

L268: Omit the word 'strictly'. What would have been the other options?

We omited the word strictly.

Alternatives to the approach of Anderson (2019) would be, for example, to simply use the standard deviation estimated from stable area for the characterisation of uncertainty as done in a similar study by Kaufmann et al. (2018). Another alternative would be, for example, the approach of Wheaton et al. (2009) which, however, is based on thresholding. Anderson (2019) clearly shows that his approach provides better and more robust results than approaches based on thresholding. As this is a well-established, peer-reviewed approach for determining the uncertainty of topographic changes, we think it is very suitable.

5. Results
L326, Figure 4: Which kind of error is shown? Flow velocity error? Please annotate
appropriately.

Yes, the flow velocity error is shown here.

We have indicated this accordingly.

Boxplots: The paper includes several boxplots. Maybe, it's good for the understanding to describe in one sentence what is shown.

We agree. We have now added a general description of the boxplots to the caption of the first boxplot in the manuscript (Fig. 4) and we refer here to the following boxplots. (L329-331)

L328: Period is missing: … RG08.

Done.

L399: Figure 6: What are the black dots? Outliers? Do the 'black dots' influence the mean value? Or asked differently: The computation of a mean value is somehow problematic considering masked areas or incomplete areas.

Yes, the black dots are outliers, spatially they are often found at the terminal part of the rock glacier. These are included in the calculation of the mean value. Since the same areas are excluded for each epoch, we believe that a comparison of the mean values is conclusive. In addition, it is only one means of comparison alongside the flow velocity maps, boxplots and violin plots. From another point of view, GPS point measurements on rock glaciers, such as those carried out by PERMOS, also produce mean values, even though the rock glacier was not measured over the entire area.

We have added a sentence in the methods section stating that due to the snow masks the mean values might not be representative for the entire rock glacier, but are only valid for the areas in which measurements were possible in all time periods. (L267-268)

L410, Figure 7: This Figure is rather complex and small. Please add annotation of the vertical axes of the violin plots.

We know that the illustration is somewhat complex, so we will describe it better in the caption. In order to save space and to be able to display the images in a larger format, we have dispensed with a description of the axis and linked it to the images using colour codes (e.g. 2012 - 2017 yellow).

We have now described this in the caption.

Background image is always the older date of the epochs(?). Remark: Timespan is the difference between two epochs.

We have replaced epoch with timespan.

Paragraph 5.5: The reviewer thinks that the computation of a mean surface elevation change and its equivalent volumetric change is questionable. As indicated in the paper analyses were restricted to certain areas only or to incomplete catchment areas. Thus, results obtained have to be analysed critically.

As described before we are aware of this and address this issue in the discussion of surface elevation changes (L681 – 683, L691-693, L709-711). Since permanent snow patches often occur at altitudes where rock glaciers are found, it will hardly be possible to prevent this.

We have now mentioned this again at the beginning of chapter 5.5 and added a reference to the discussion here. (L453-455)

Remark: The reviewer
thinks that in Figure 9 the black error bars of the uncertainty of measurements are too optimistic. It would be good to add at least one critical sentence about the significance level of the volumetric change.

The uncertainties were determined according to Anderson (2019) as this is a well-established, peer-reviewed approach for determining the uncertainty of topographic changes, we think the uncertainties are valid. Anderson (2019) shows that if the systematic error is very small, positive and negative errors balance each other out and thus the total uncertainty, is then largely determined by uncorrelated and correlated random errors. This means that the total error is small if these two error components are small. This is described in chapter 5.2 Errors and uncertainties (L333-341). In the supplement, the different types of error are broken down and specified to three decimal places. Since the uncorrelated and correlated random errors are very small, they are often given here as 0.000.

L504: typo prvious

Done.

Paragraph 5.6: If there is surface-parallel flow/creep mean flow velocity and mean surface elevation change are highly (100%) correlated. Because of differential mass movements (straining) proper interpretation of subset areas is rather difficult.
(comment of editor: I agree with the referee but I think you already discuss this issue pretty well, so not much to do here).

We agree and thank you for your assessment.

We are aware that the analysis is not perfect due to natural conditions (e.g. snow patches) or missing data (e.g. information on the internal structure of rock glaciers and its change), but we clearly address these drawbacks and nevertheless present valid and meaningful results.

L554: typo multidecadal multi-decadal

Done.

6.(?) Discussion
Analysis is still speculative and open for discussion and further research.
(comment of editor: I do not expect you to change too much here, but maybe the tone of the interpretation can be a bit more open).

In our assessment, we do not make a hard statement for any of the considerations and validate our hypotheses with existing literature. Nevertheless, we have chosen a more open tone in some places:

Sentence added: "Although a direct connection cannot be proven by the qualitative analysis."

Rplaced: "…can be seen…" by "…might be seen…"

Rephrased: "This possibly led to…" "This may have portentially led to…"

Figure 12: Figures (c) and (e) are rather small.

We agree that the figures are small. Since they are important and we want to present the seasonal data alongside the annual data, but do not want to create an additional figure, we have decided to present it this way.

We have enlarged the sub-figures c) and e) a little.

7 Conclusion
L704 Use kinematics instead of dynamics.

Done.

List of comments by editor that should be adressed

In general:
- check and better describe the captions again, quite a few figure are not explained that well (box-plot, etc…)

We have revised some of the captions.

- please carefully proofread the manuscript at the end again, preferably by a native english speake, to avoid any issue of editing, language/formulations, etc..

We read the manuscript carefully at the end and corrected some mistakes.

L13: nine or eight rock glaciers respectively

Done.

L16: to variable degrees

Done.

L16/17: Rock glaciers related to glacier forefields showed…

Done.

L26: …rock glaciers in the same catchment…

Done.

L32: 'common layers' awkward formulation, maybe better say 'They suggest rock glacier to be similarly composed by such layers, although…

Done.

L37: …therefore the temperature…

Done.

L84: I would be more specific in referring to the Copernicus data (official reference, weblink where/how downloaded etc…)

We have referred to the copernicus data used according to their guidelines.

We have now added a web link.

Line 97: which (not wich)

Done.

Line 102: I think it should be in plural: …velocities of rock glaciers …

Done.

Line 130/131 and also in table 2: why saying 'chair of physical geography', better say the name of the responsible person or just mention the institute, Physical gepgraphy….

We have used "chair of physical geography" because there is no association with an institute at the university. Chair of physical geography does not refer to a single person but is used as a synonym for the physical geography workgroup.

Line 136: typo, there is a space missing between 'investgated' and 'rock glacier'

Done.

Line 161, caption: Style, repetition of word distance, maybe say: The distance is given to the center of the study area in (km).

Done.

Line 171: delete 'the DFG founded project', this is irrelevant repetation, just say 'of the PROSA project'

Done.

Line 182: '…from the aerial images…'

Done.

Line 366, in caption: there is a full stop missing after 'forcing'. Also the 'mean' at the end of the line should start with a lower case 'm'.

Done.

Line 379/380: this sentence can be deleted or moved to discussion, here you present the results! 'Throughout the Alps, there is only a very small or no clear trend with regard to the annual precipitation development in the 20st century (Beniston, 2006).'

We agree and have removed the sentence.

Line 398ff/caption fig 6: better explain the box plots (what percentiles used, black dots? Medians or means shown,…what are red boxes and dots…. Just be a bit more specific in the caption.
Also what is 'KT 09'? do you mean 'RG09'?

As described earlier, we have included a general description of the boxplot parameters used in the caption of the first boxplot in the manuscript, with reference to the other boxplots (Fig. 4). In Figure 6, we also explain the red boxes, dots and the outliers.

Yes, we mean RG09 and have changed this accordingly.

Line 408: …characterized by relatively…

Done.

Line 411/caption Fig 7: the 'violoin' plot could use some more info and it should be: 'The units of …'

We agree, although these are a helpful illustration of the development of flow velocities, they can be a little confusing if you don't know what is being depicted.

We have included a short definition of a violin plot in the caption and explained its colour coding (L416-419).  Even before the revision, the text briefly explained how to read the violin plots (L445-447).

We have corrected "The units of…".

Line 415: …has to be put into perspective, …

Done.

Line 431: style, repetition: replace 'had zones with higher and lower flow velocities' by 'flow velocities varied spatially'

Done.

Line 440: I think it should be 'give an indication'

Done.

Line 448: …are also shown…

Done.

Line 449, caption: where are these greyed out areas that are stable?????

These are located to the right of the measurements on the respective rock glaciers. For each rock glacier, there is one group of four measurements on the rock glacier and one group of four measurements on the stable areas (greyed out) to the right.

We have added a text to the figure to make this clearer.

Line 459: … are of TC-type…

Done.

Line 461: style, …In contrast for RG06….

Done.

Figure 9: I suggest to label the x-axis properly with the years of the epochs rather than numbers.

We agree. Since the long years (e.g. 1953-1970/71) make it very confusing and narrow to label the x-axe directly, we have inserted the years as a legend in the figure.

Line 495: rather confusing figure references, I think you mean: '…we have plotted their mean values for subareas of the individual rock glaciers in Fig. 11, with the subareas shown in Fig. 10. '

Done.

Lines 496/497: add a comma after each numbered point, e.g.: (1) representative of the rock glacier, (2) both

Done.

Line 504: previous (not prvious)

Done.

Line 507 caption: '…and the four epochs…

Done.

Line 624: an overarching summary of the figure would help: 'Flow velocity variations in context of climatic forcing. (a)…'

Done.

Line 681: …on TC-type rock…

Done. We have also corrected this elsewhere in the manuscript.

Line 760: a space is missing between 'the' and 'study' (thestudy)

Done.

**References**

Anderson, S. W.: Uncertainty in quantitative analyses of topographic change: error propagation and the role of thresholding, Earth Surf. Process. Landforms, 44, 1015–1033, https://doi.org/10.1002/esp.4551, 2019.

Fey, C. and Krainer, K.: Analyses of UAV and GNSS based flow velocity variations of the rock glacier Lazaun (Ötztal Alps, South Tyrol, Italy), Geomorphology, 365, 107261, https://doi.org/10.1016/j.geomorph.2020.107261, 2020.

Kaufmann, V., Seier, G., Sulzer, W., Wecht, M., Liu, Q., Lauk, G., and Maurer, M.: Rock glacier monitoring using aerial photographs: conventional vs. UAV-based Mapping - a comparative study, Int. Arch. Photogramm. Remote Sens. Spatial Inf. Sci., XLII-1, 239–246, https://doi.org/10.5194/isprs-archives-XLII-1-239-2018, 2018.

Kummert, M., Bodin, X., Braillard, L., and Delaloye, R.: Pluri-decadal evolution of rock glaciers surface velocity and its impact on sediment export rates towards high alpine torrents, Earth Surf. Process. Landforms, 105, 113, https://doi.org/10.1002/esp.5231, 2021.

Wheaton, J. M., Brasington, J., Darby, S. E., and Sear, D. A.: Accounting for uncertainty in DEMs from repeat topographic surveys: improved sediment budgets, Earth Surf. Process. Landforms, 25, n/a-n/a, https://doi.org/10.1002/esp.1886, 2009.